# Heading representations in primates are compressed by saccades

Frank Bremmer [1], Jan Churan[1] & Markus Lappe[2]

Perceptual illusions help to understand how sensory signals are decoded in the brain. Here we report that the opposite approach is also applicable, i.e., results from decoding neural activity from monkey extrastriate visual cortex correctly predict a hitherto unknown perceptual illusion in humans. We record neural activity from monkey medial superior temporal (MST) and ventral intraparietal (VIP) area during presentation of self-motion stimuli and concurrent reflexive eye movements. A heading-decoder performs veridically during slow eye movements. During fast eye movements (saccades), however, the decoder erroneously reports compression of heading toward straight ahead. Functional equivalents of macaque areas MST and VIP have been identified in humans, implying a perceptual correlate (illusion) of this perisaccadic decoding error. Indeed, a behavioral experiment in humans shows that perceived heading is perisaccadically compressed toward the direction of gaze. Response properties of primate areas MST and VIP are consistent with being the substrate of the newly described visual illusion.

[1] Department of Neurophysics & Marburg Center for Mind, Brain and Behavior - MCMBB, Philipps-Universität Marburg, Karl-von-Frisch Straße 8a, 35043 Marburg, Germany. [2] Department of Psychology & Otto Creutzfeldt Center for Cognitive and Behavioral Neuroscience, University of Muenster, Fliednerstraße 21, 48149 Münster, Germany. Correspondence and requests for materials should be addressed to F.B. (email: frank.bremmer@physik.uni-marburg.de)

Visual perceptual illusions have been successfully employed to better understand how sensory signals are processed in the brain[1]. Often, such illusions allow to infer at what stage of the processing hierarchy certain computational steps take place, and resultant predictions can be validated experimentally. In the motion domain, examples include the perception of coherent motion[2], the motion aftereffect[3], and implied motion[4] as encoded in cortical areas MT (the middle temporal area) and MST (the medial superior temporal area). In the spatial domain, many perceptual illusions are induced by fast ballistic eye movements, so-called saccades. Behavioral studies in humans have shown that briefly flashed stimuli that are presented in the temporal vicinity of saccades are mislocalized in a characteristic fashion. Dependent on the exact experimental condition, mislocalization can be described as a dynamic, global shift of space or as a compression of perceived space toward the saccade target[5–7]. Neurophysiological studies in macaque monkeys have identified neural correlates of both dynamic error patterns[8–12].

In everyday life, visual motion and eye movements often occur concurrently. As an example, self-motion through an environment induces reflexive, optokinetic like eye movements, consisting of an alternation of smooth tracking phases and saccades[13, 14]. It was shown before that self-motion direction (heading) can be decoded from neural responses during fixed gaze[15, 16]. Accordingly, the question arises, if such decoding is also possible during eye movements. Previous studies have provided evidence for this idea by demonstrating that the neural tuning for heading in areas MST and VIP is invariant with respect to slow eye movements[17–19]. Such an eye movement invariance can be considered a prerequisite for the successful decoding of self-motion direction. Here we test this hypothesis and decode heading during smooth eye movements and during saccades.

We find that a linear heading-decoder performs veridically during slow eye movements. During saccades, however, the decoder incorrectly indicates compression of heading toward straight ahead. In a second experiment, we test whether these

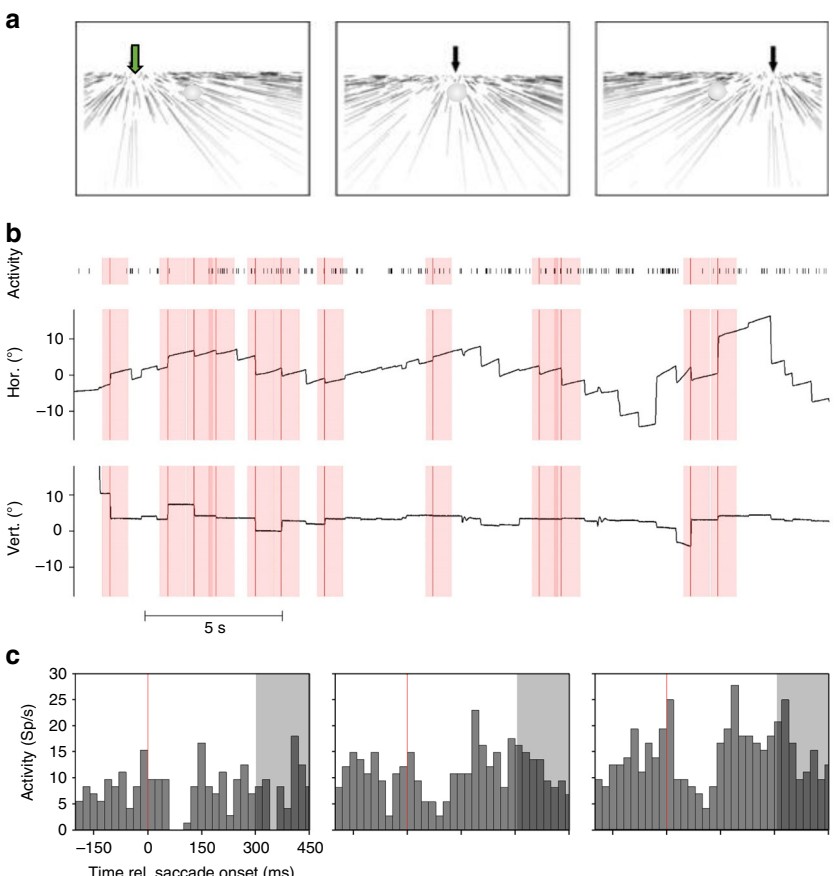

**Fig. 1** Stimuli, eye movements, and neural responses. **a** Panels show schematically the optic-flow stimuli presented to the monkeys. In pseudorandomized order, stimuli simulated self-motion across a ground plane to the left, straight ahead or to the right. Without additional eye movements, heading direction is given by the singularity of the optic flow field[15, 22, 23, 57], as indicated by the tip of the arrow in each panel. **b** The panels show spontaneous horizontal and vertical eye movements as well as activity of an MST neuron for an exemplary trial of self-motion to the left (also indicated by the green arrow in **a**. Reflex-like slow eye movements mainly to the right were accompanied by saccades in the opposite direction. We determined the onset of saccades, as indicated by the vertical red lines in the neural and eye-movement data. In order to determine a neuron's tuning for self-motion direction, we split the long-trial data into 650 ms long snippets, positioned around the onset of each saccade. Our selection criterion for such snippets required that for a given saccade no further saccade must occur from 200 ms before this saccade until 450 ms thereafter. The temporal snippets that met this criterion are marked with red, transparent vertical rectangles in the neural and eye-movement data. **c** The resulting perisaccadic responses of the same neuron, whose discharges are shown in **b**, are shown for all three self-motion directions. This neuron was significantly tuned for rightward self-motion (ANOVA on ranks, 2 df, $P < 0.05$). We used a brief, 150 ms post-saccadic interval (300–450 ms, indicated by the gray shaded areas in the response histograms) of each neuron's response to the three simulated headings to compute the linear regression parameters. For this neuron, the linear regression function $y(t) = 0.108$ (Sp/s)/° * $x + 11.77$ Sp/s; $r^2 = 0.997$, $P < 0.05$, fitted the data significantly

perisaccadic changes in the readout of the decoder have consequences on perception of human subjects. As predicted, perceived heading is perisaccadically compressed. A behavioral control experiment reveals compression to be directed toward the direction of gaze rather than the head- or body-midline. Response properties of primate areas MST and VIP are likely the neural substrate of this newly described visual illusion.

## Results

**Decoding of heading across eye movements.** We recorded activity from 119 neurons in the medial superior temporal area (area MST) and the ventral intraparietal area (area VIP) of three macaque monkeys (Supplementary Table 1 for details). During the recording monkeys freely viewed long sequences of optic flow stimuli simulating self-motion across a ground plane in one of three directions (±30° and straight ahead, Fig. 1a).

Such visual stimuli induce reflexive, optokinetic-like eye movements, i.e., an alternation of slow tracking and fast (saccadic) eye movements. Figure 1b shows representative eye-movement traces in the middle and bottom row along with neural activity (top row). As a first step of our data analysis, we determined the onset times of each saccade during a given trial. The perisaccadic activity for a given self-motion direction was averaged in order to determine the response dynamics of a neuron under study. By definition, each response window represents neural activity during an alternation of slow-tracking eye movements (from −200 to 0 ms), followed by a saccade (from 0 to about 60 ms), followed again by slow-tracking eye movements (from about 60 to 450 ms). Figure 1c shows data from the same neuron from area MST, whose activity during a leftward heading was shown in Fig. 1b. For this neuron, mean discharges were lowest for self-motion to the left (left panel) and highest for self-motion to the right (right panel). For all three self-motion directions, neural discharges were rather constant during the slow eye-movement phases. Shortly after the saccades (indicated by the red vertical line in each panel), however, neural activity was briefly suppressed. Data from all neurons from areas MST and VIP were analyzed accordingly.

In a next step, we implemented an algorithm to decode self-motion direction from neural discharges (see Methods for details). In our decoding approach, firing rate directly translates into heading. We applied the decoder to the continuously recorded discharge of each neuron that showed a statistically significant tuning for heading (ANOVA on ranks, 2 df, $P < 0.05$, see Methods for details), i.e., $71/119 = 60\%$ of the neurons. (Detailed cell numbers are given in Supplementary Table 1).

The decoding approach provided us with a time-resolved estimate of self-motion direction based on the discharges of a population of cells ($n = 71$). We determined heading from 200 ms before until 450 ms after the saccade in steps of 20 ms as the median value of the heading estimates from all neurons (Fig. 2a).

The colored symbols and solid lines indicate the decoded heading. Starting at the earliest time, i.e., 200 ms before the saccade, decoded heading for all three self-motion directions was close to veridical up to 100 ms before saccade onset. Then, however, it began to collapse. One-hundred ten milliseconds after saccade onset, compression toward straight ahead was at maximum, with decoded values being close to zero for all three headings. Thereafter, decoded values rapidly recovered and became close to veridical again. Maximum compression as indicated by the minimum of the standard deviation was observed 110 ms after saccade onset (Fig. 2b).

Given the results shown in Fig. 1c, we next investigated whether the compression of the representation of heading, as observed in the decoding of neural activity from monkey areas

MST and VIP, could merely be a by-product of an overall perisaccadic reduction of neuronal excitability, i.e., saccadic suppression. Such saccadic suppression has been shown previously for areas MST and VIP[20]. Hence, we determined the time course of the average neural activity, aligned to saccade onset (Fig. 3a).

The time courses of saccadic suppression, as shown here, and the time course of compression of decoded heading as shown in Fig. 2b were indeed very similar, although not identical. Maximum suppression was observed 80 ms after saccade onset, i.e., 30 ms earlier than the maximum compression. This might be considered first evidence that a mere saccadic suppression cannot account for the observed compression of the decoded representation of heading.

To further understand the neural basis of the perisaccadic decoding error, we built a rather qualitative model. Here we considered a grand population of neurons from areas MST and

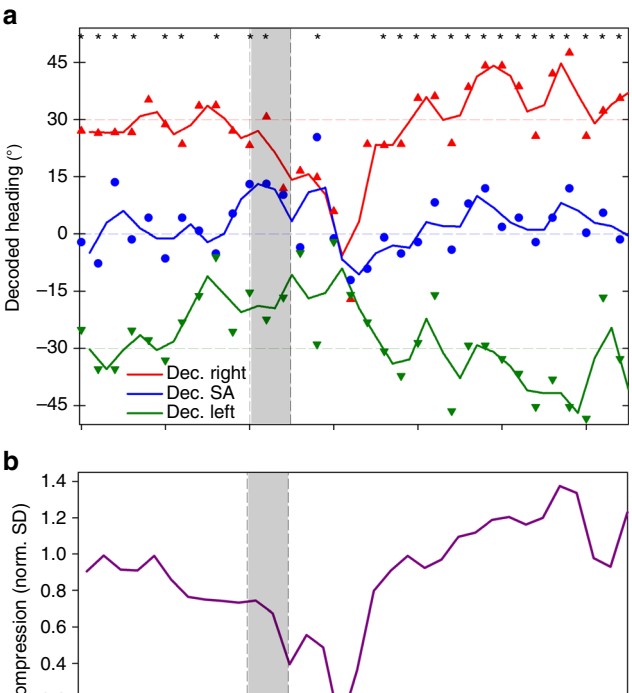

**Fig. 2** Decoded heading. **a** Colored symbols indicate the decoded heading from the neuronal population ($n = 71$) for stimuli simulating movement to the left (−30°, green, downward pointing triangles), straight ahead (0°, blue, circles), and to the right (+30°, red, upward pointing triangles). Data are aligned to saccade onset. The solid lines (colored as the symbols) represent a running mean, based on the average values of decoded heading in two consecutive 20 ms bins, assigned to the time-point centered between the two samples. Asterisks indicate decoded heading values that were significantly different from each other (ANOVA on ranks, 2 df, $P < 0.05$, FDR-corrected). One-hundred ten milliseconds after saccade onset, compression toward straight ahead was at maximum, with decoded values being close to zero for all three headings: leftward heading (−30°) → decoded heading: −9.0°; forward heading (0°)→decoded heading: −6.7°; rightward heading (+30°) → decoded heading: −5.6°. From 160 ms after saccade onset on, decoded headings were again significantly different from each other (ANOVA on ranks, 2 df, $P < 0.05$, FDR-corrected). **b** Normalized standard deviation of the three decoded headings (from the raw data values) in steps of 20 ms as measure of the dynamics of perisaccadic compression

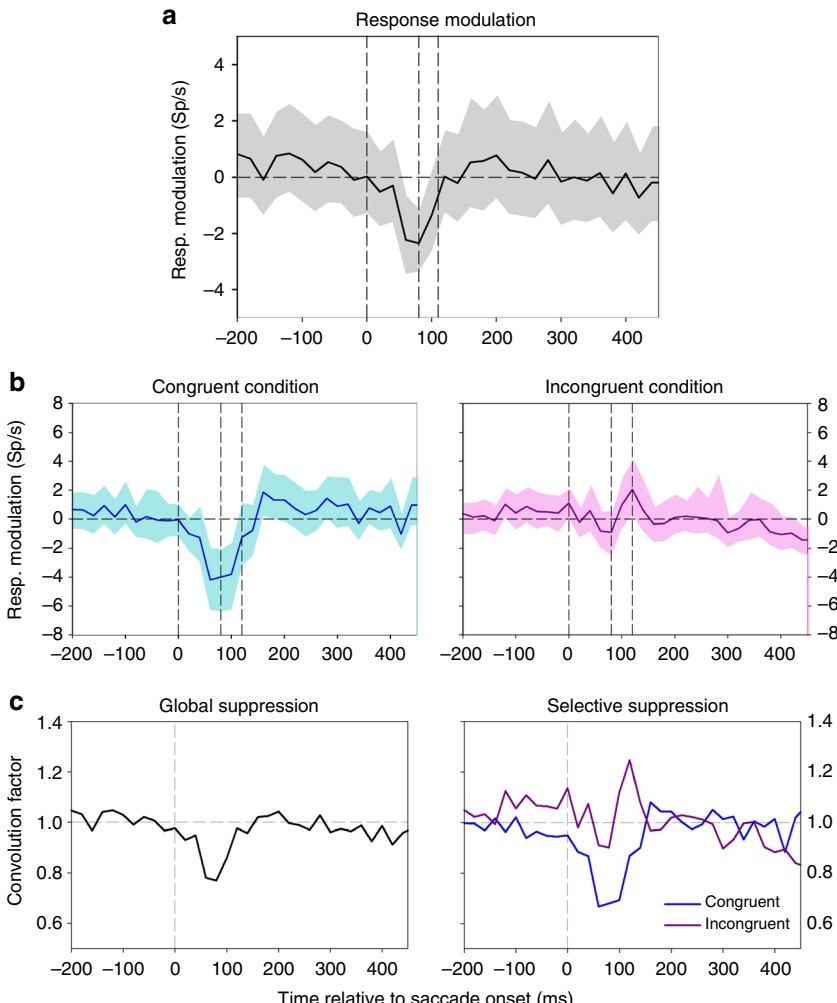

**Fig. 3** Global and selective response modulation at the population level. **a** We first computed for each neuron its time-resolved discharge in the time window [−200 ms 450 ms] and subtracted the mean activity in this window from the time-resolved discharge, providing a neuron's perisaccadic response modulation. We then averaged these response modulations across all 71 neurons. The solid black line depicts this mean response modulation, the gray surround indicates the 95% confidence interval as determined from bootstrapping. The vertical dashed lines indicate (i) saccade onset (0 ms), (ii) the time-point of maximum suppression (80 ms), and (iii) the time-point of maximum compression (110 ms). **b** When neurons were stimulated with their preferred heading direction (congruent condition, left panel, blue curve), saccadic suppression of activity was observed. When the same neurons were stimulated with their non-preferred heading (incongruent condition, right panel, purple curve), no saccadic suppression, but rather a significant response enhancement could be observed. In both panels, the solid lines indicate the time resolved response modulation, while the shaded, colored areas indicate the 95% confidence interval as determined from bootstrapping. Vertical dashed lines as in **a**. **c** For the quantitative model on the role of saccadic suppression on heading decoding we convoluted the time courses of the artificial neurons with the response profiles deduced from global suppression or selective suppression, respectively. These kernels were obtained by normalizing the time courses of (i) global suppression (black line, left panel), (ii) selective suppression occurring during congruent stimulation (blue line, right panel), and (iii) response enhancement occurring during incongruent stimulation (purple line, right panel) such that the values >100 ms before and after the saccade were centered on 1.0

VIP to be split up into two subpopulations: neurons that were tuned for leftward headings constituted "Subpopulation 1". By definition, the slope $m$ of the regression function fitted to the heading responses of these neurons was smaller than zero ($m < 0$). "Subpopulation 2" comprised those neurons that are tuned for rightward headings ($m > 0$). This procedure is graphically illustrated in Fig. 4a. Published data predict that from such a grand population of MST and/or VIP neurons, roughly the same number of neurons should fall into one of the two subpopulations[15, 17, 21–23]. Indeed, in our data set of $n = 71$ neurons, $n_{Subpop1} = 38$ neurons were tuned for leftward self-motion ($m < 0$), while $n_{Subpop2} = 33$ were tuned for rightward self-motion ($m > 0$).

For our decoding approach, we had employed linear regression as approximation for the real neurons' response tunings. We

applied this linear decoder also to our qualitative model. This is illustrated in Fig. 4b. Here we assumed presentation of a rightward self-motion stimulus. Accordingly, the activity of the neurons tuned for rightward heading is high (Subpopulation 2, light blue line), while the activity of neurons tuned for leftward heading is low (Subpopulation 1, pink line). The observed activity distribution of neurons from both subpopulations translates into a veridical heading estimate, i.e., self-motion to the right, as indicated graphically by the purple arrow.

In the next and final step, we presumed perisaccadic modulation of neural activity, i.e., saccadic suppression. As a first hypothesis, we assumed that saccadic suppression acts equally on both subpopulations as global-and-proportional suppression (Global suppression, Fig. 4c, left panel). Accordingly, this suppression would reduce activity in both subpopulations

alike. Given the tight relationship between activity and represented heading, however, this response reduction would have two different, counteracting effects in the two subpopulations. For those neurons that are stimulated with their preferred heading direction (Subpopulation 2, blue line), the decoded heading would shift toward straight ahead, which would be in line with a compression of the represented heading. Yet, for subpopulation 1 (pink line), reduced activity would result in a heading estimate further away from the center. Accordingly, heading as decoded from the two subpopulations (purple arrow) would show a minimal compression, with an increase in variance, i.e., a reduced decoding precision. As an alternative (selective suppression, right subpanel), suppression might act selectively only on those neurons that are driven by the self-motion stimulus, i.e., here: rightward heading for rightward tuned neurons (congruent condition). It is well known that neural activity in motion sensitive areas MST and VIP often gets significantly reduced compared to baseline (i.e., spontaneous activity) when non-preferred stimuli are presented[15, 22] (incongruent condition). It has been suggested that such response modulation results from mutual inhibition of neurons that are tuned to different directions of (self-)motion[24]. This idea suggests that neurons, which receive inhibitory input from those neurons that are driven (close to) optimal by a current stimulus, should reveal (probably with a certain delay) a release from inhibition when the optimally driven neurons are less activated (or suppressed, e.g., by a saccade). Such an increase of activity of the weakly activated cells due to a release from inhibition would cause a decoding shift toward straight ahead. Accordingly, both response modulations would cause the decoded heading to be shifted toward the center, thereby leading to a compression of the representation of heading without modulation of the precision of the decoded heading (purple arrow in Fig. 4c, right panel, compression).

This qualitative model led us to hypothesize the occurrence of selective suppression and release from inhibition in our data set. Accordingly, we analyzed in greater detail the individual heading responses of the neurons in our population. Indeed, we found evidence for a suppression of responses for one movement direction and a slight response enhancement for the opposite direction. Two examples are shown in Fig. 5.

In order to quantify this effect at the population level, we assigned the response of each neuron to leftward and rightward heading either to the congruent or incongruent condition, dependent on the neuron's tuning for heading. Importantly, each neuron contributes to both conditions: leftward tuned neurons contribute to the congruent condition during leftward self-motion and also to the incongruent condition during rightward self-motion (vice versa for rightward tuned neurons). Then, we tested for a significant difference of the perisaccadic response modulations in the congruent and incongruent condition by comparing the neural activity in a fixed, post saccadic window $-=40$–160 ms after saccade onset. In Fig. 5, in each panel, saccade onset is indicated by the vertical, solid line. The time window for analysis [40 ms 160 ms] is indicated by the pair of vertical red dashed lines in each panel.

This window was considered wide enough to capture both, suppressive and dis-inhibitory effects. We compared the neural activity in this time window with activity long after the saccade (300–450 ms after saccade onset) by computing the ratio of the two, defining a modulation index. Based on this definition, an index of 1.0 indicates no perisaccadic response modulation. An index smaller than 1.0 indicates response suppression, while an index >1.0 indicates response enhancement. Our prediction from the hypothesis detailed above was an index below 1.0 for the congruent condition and an index above 1.0 for the incongruent condition. This hypothesis was confirmed in our population data

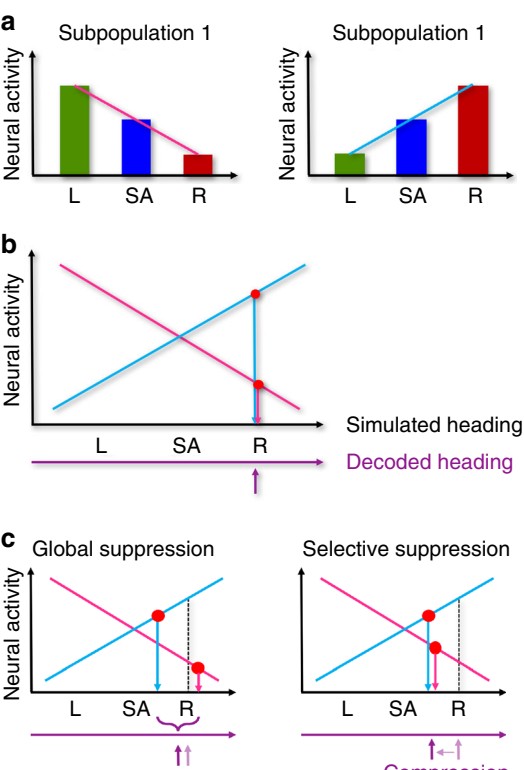

**Fig. 4** Qualitative model on the role of saccadic suppression on heading decoding. **a** We consider a population of MST and VIP neurons to split up into two subpopulations. **b** During smooth eye movements both subpopulations code for the same heading (rightward in this example). **c** Strong perisaccadic compression as found in our neurophysiological data can be explained by selective perisaccadic suppression of stimulus driven responses (right panel), but not by global suppression of activity (left panel). For details see text

($n = 71$ neurons, Fig. 6). The 95% confidence intervals as determined from bootstrapping were below 1.0 in the congruent condition and above 1.0 in the incongruent condition.

While this result was fully in line with our hypothesis of selective suppression and enhancement or release from inhibition, the computation based on a fixed analysis window (40–160 ms after saccade onset) was a rather coarse measure. To overcome this limitation, we assigned the response of each neuron from our recorded sample of MST and VIP neurons to leftward and rightward heading stimuli to the congruent or incongruent condition and determined the full time-course of the population response modulation (Fig. 3b). Indeed, when neurons were stimulated with their preferred heading direction (congruent condition, left panel, blue curve), saccadic suppression of activity was observed. In contrast, when the same neurons were stimulated with their non-preferred heading (incongruent condition, right panel, purple curve), no saccadic suppression could be seen. Instead, a small but significant response enhancement occurred. In both panels, the solid lines indicate this time resolved response modulation, while the shaded, colored areas indicate the 95% confidence interval as determined from bootstrapping. Finally, as hypothesized, the peak of the enhancement (or release from inhibition) in the incongruent condition (120 ms after saccade onset) occurred 40 ms after the peak of suppression in the congruent condition (80 ms after saccade onset). In both panels, both events are indicated together with the time of saccade onset by vertical dashed lines.

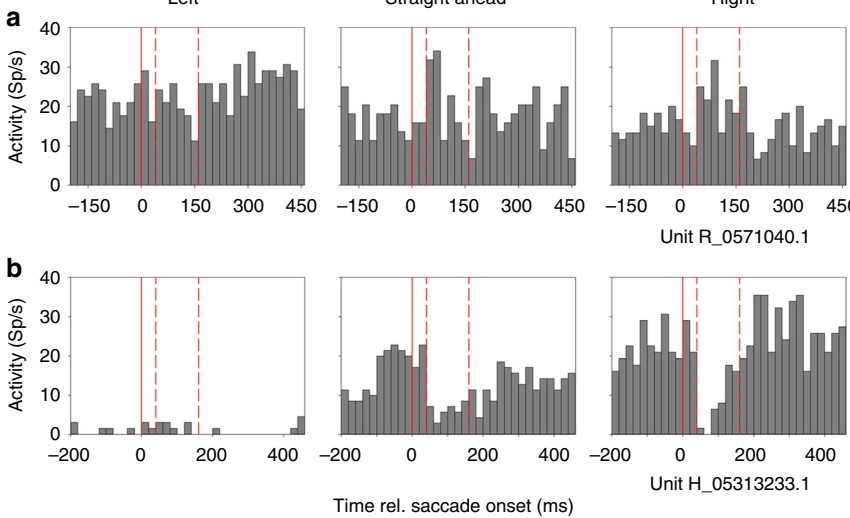

**Fig. 5** Response modulation at the single-cell level. **a** The responses of the first neuron revealed a significant tuning for leftward self-motion, as indicated by the pre- and post-saccadic activities (ANOVA on ranks, 2 df, $P < 0.05$). While there was a slight perisaccadic suppression of responses for leftward self-motion, perisaccadic responses were slightly enhanced for rightward motion. **b** This neuron revealed an opposite tuning, i.e., it was significantly tuned for rightward motion (ANOVA on ranks, 2 df, $P < 0.05$). In line with our hypothesis, this neuron revealed a strong saccadic suppression for rightward heading, but a small release from inhibition for leftward heading

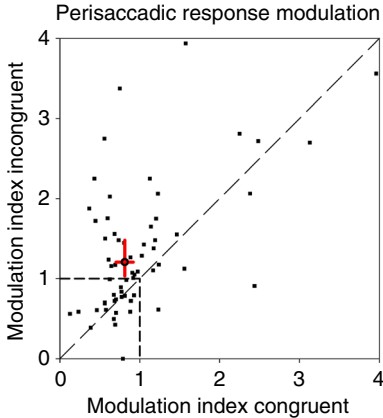

**Fig. 6** Modulation index for congruent and incongruent stimulation. Each data point represents data from a single neuron, with the x-value depicting the index in the congruent condition and the y-value depicting the index of the same neuron in the incongruent condition. The majority of data points lies above the diagonal. For the population of cells, the median of the index was significantly <1.0 (0.812, one-sided sign test, $P < 0.05$) in the congruent condition, and significantly >1.0 in the incongruent condition (1.206, one-sided sign test, $P < 0.05$). The 95% confidence intervals (red lines) as determined from bootstrapping were <1.0 in the congruent condition and >1.0 in the incongruent condition

In a final step, we modeled the effect of saccadic suppression in a quantitative model. The advantage of this approach is that one can apply various forms of perisaccadic response modulation and determine, if and if so which form of suppression reproduces the effect of a compression of decoded heading. For this model, we created a set of $n = 71$ "virtual neurons", i.e., time courses of neural activity. The activity levels were modeled after our real data set, i.e., activity distribution across the population was identical to our real data set, with additional noise. Importantly, model neurons could not have negative firing rates. Instead, all firing rates, which would have been negative, were set to zero. We

simulated presentation of five self-motion directions ($-30°$, $-15°$, $0°$, $15°$, and $30°$). These artificial neurons were linearly tuned to self-motion direction, with the distributions of strengths of their tunings modeled after our real data set. The results are shown in Fig. 7. Obviously, only selective suppression, i.e., suppression of the neurons in the congruent condition combined with a release from inhibition of the neurons in the incongruent condition (top right panel), but not global suppression triggers a compression of decoded heading similar to the one found from our real neurons (Fig. 2), with a maximum of suppression around 100 ms after saccade onset.

In summary, our analysis clearly shows for the first time that heading can be decoded robustly and reliably during smooth eye movements from neural activity in macaque areas MST and VIP. This is remarkable given the rather small sample size of only 71 neurons. Saccades, however, induce a systematic error of decoded heading toward straight ahead (red and green symbols and data curves in Fig. 2a). Since functional equivalents of monkey areas MST and VIP have been identified in humans (MST;[25] VIP[26]), we asked whether the decoding from monkey neurophysiological data would correctly predict human behavior, i.e., whether or not humans perceive compressed heading across saccades. This result would be noteworthy for at least two reasons: firstly, neurophysiological and computational results from the animal model would have correctly predicted a hitherto undescribed visual perceptual illusion. Secondly, it would provide further and strong evidence for the similarity of visual motion processing in humans and non-human primates.

**Heading perception across saccades**. In our psychophysical study, subjects were presented brief (40 ms) sequences of visual stimuli simulating self-motion across a ground plane (Fig. 8). In separate blocks, subjects either kept central fixation or performed a visually guided 10° upward saccade along the vertical meridian.

Data from one subject (VP01) are shown in Fig. 9a (data from all eight subjects are shown in Supplementary Fig. 1). For each subject, perceived heading during fixation trials was slightly biased toward fixation, as indicated by the dashed, colored

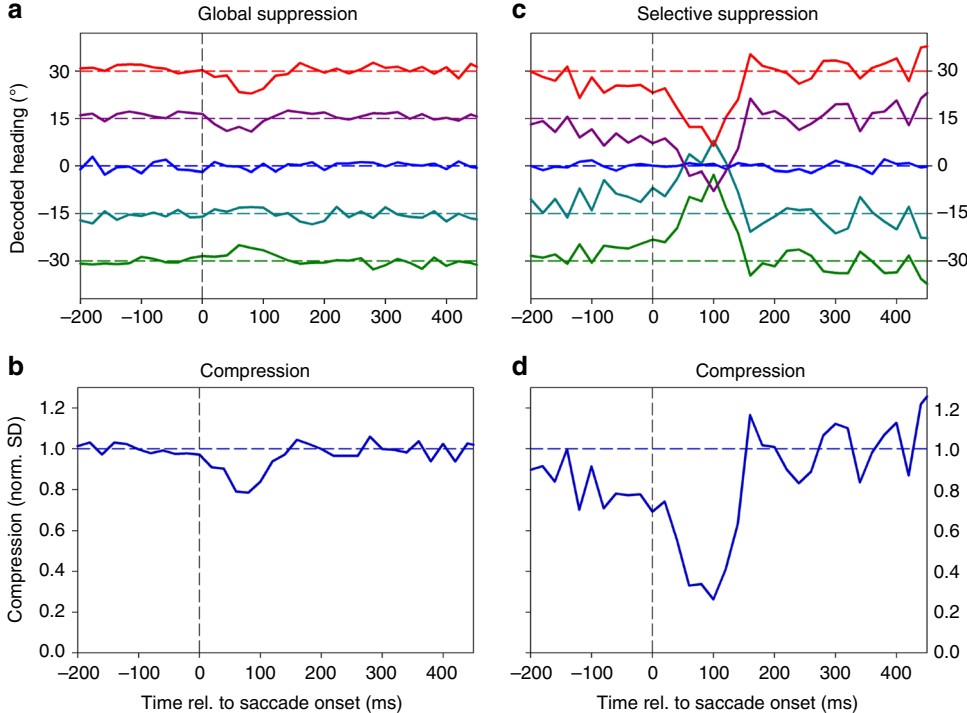

**Fig. 7** Quantitative model on the role of saccadic suppression on heading decoding. We created a set of $n = 71$ virtual neurons, i.e., time courses of neural activity. All response properties were modeled after our real data set (see text for details). We convoluted the time courses with the response profiles deduced from global suppression or selective suppression, respectively (see Methods for details, and Fig. 3). We repeated this procedure 50 times for each of the two forms of saccadic suppression, i.e., global and selective, and computed the average time courses of decoded heading. When applying global suppression (**a**), there was only a marginal modulation of decoded heading shortly after the saccade. This resulted in a minimal compression (**b**). Like for the physiological data, we determined compression as normalized standard deviation of the five time-courses of decoded heading. On the contrary, selective suppression, i.e., suppression of the neurons in the congruent condition combined with a release from inhibition of the neurons in the incongruent condition (**c**), triggers a compression of decoded heading, with a maximum of suppression around 100 ms after saccade onset (**d**)

horizontal lines. Across all subjects, in saccade trials, perceived heading was similar to fixation trials up to 80 ms before saccade onset, and from about 60 ms after saccade onset on. Between those times, i.e., perisaccadically, however, perceived heading became compressed toward straight ahead. Qualitatively, this effect was very consistent across subjects. Like for the physiological data, we determined the time course of compression by computing the normalized standard deviation of the perceived headings. The result for data from the same subject (VP01) is shown in Fig. 9b: maximum compression was observed just prior to saccade onset, i.e., at $t = -1$ ms.

Quantitatively, the temporal dynamics of perceived heading were also very consistent across all eight subjects: peak compression occurred between $-10$ and $-1$ ms relative to saccade onset, with a mean value of $t = -8.25$ ms, as shown in Fig. 9c. Given the similarity of perisaccadic heading compression as determined from our neurophysiological and psychophysical data, we employed temporal cross-correlation to determine the best overlap between the two time-courses. A shift of $\tau = -116$ ms of the neurophysiological with respect to the psychophysical data resulted in the largest correlation coefficient (Fig. 9d). At first glance, it might appear surprising that the behavioral data lead the neurophysiological by 116 ms. Yet, this has to be expected because the value of tau corresponds to a visual response latency and is perfectly in line with values in higher visual cortical areas of the macaque[27].

Previous studies on perisaccadic modulation of visual perception showed that response precision of behavioral responses did not decrease perisaccadically[28]. Accordingly, we were also interested in the precision of the decoded headings as well as of the behavioral responses of our subjects (Supplementary Fig. 2, with behavioral data from each subject shown in Supplementary Fig. 3). Precision did not modulate perisaccadically, neither for the decoded heading nor for the behavioral responses. This finding is in line with the above mentioned behavioral studies.

Motion perception across saccades is impaired due to saccadic suppression[29–31]. Therefore, we aimed to determine whether this reduction of the perisaccadic perception of motion could have had a critical influence on our behavioral data. In a control experiment, we asked five of the eight subjects to discriminate between self-motion stimuli simulating forward and backward self-motion. During steady fixation (>50 ms prior to and after saccade onset), performance of subjects was very high (97.1%), but significantly smaller than perfect (i.e., 100%. One-sided signed rank test, $P < 0.0001$). If saccadic suppression would have strongly compromised self-motion perception, we would have expected a drop in performance close to chance level (50%) in the temporal vicinity of a saccade. Yet, discrimination performance <50 ms prior to and after saccade onset was almost as high as during fixation, i.e., 96.1%. While this value was significantly smaller than 100% (one-sided signed rank test, $P < 0.0001$), it was also significantly larger than chance level (one-sided signed rank test, $P < 0.001$). A paired, one sided Wilcoxon signed rank test found no difference in performance in the perisaccadic interval compared to fixation ($P > 0.5$). Effect size of the perisaccadic perceptual modulation, $d'$, was 0.21, i.e., comparably small. While the nature of the control experiment (discrimination between forward and backward self-motion) was different from our main

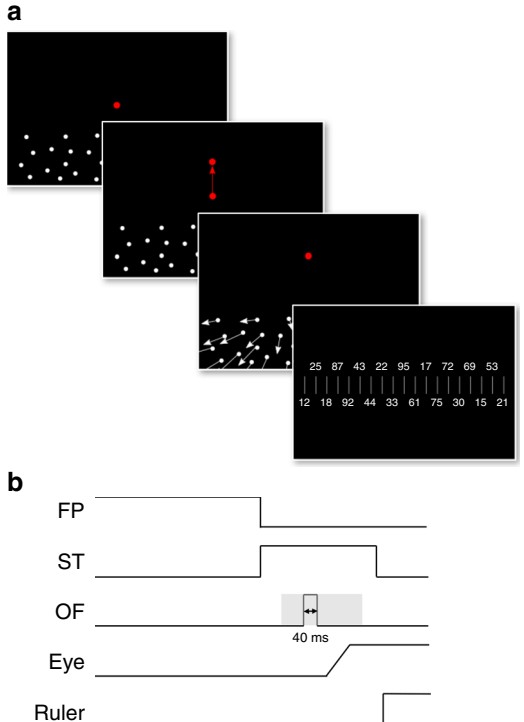

**Fig. 8** The paradigm for testing perceived heading in humans. **a** Self-motion stimuli as well as fixation and saccade targets were generated with the Psychtoolbox[54] and back-projected via a video projector onto a tangent screen covering the central 81 × 65° of the visual field. Across trials, simulated self-motion was pseudo-randomized in one of the five directions: to the left (−30° and −15°), straight ahead (0°), or to the right (15° and 30°). Self-motion stimuli consisted of five consecutive frames of 100% coherent dot motion, i.e., lasting 40 ms during which a total forward displacement of 3.3 m was simulated. Each trial started with presentation of the fixation target and the stationary ground-plane stimulus. After 1600–2000 ms, the fixation target was switched off and the saccade target was switched on (until the end of the trial), inducing a visually guided vertical saccade of 10°. Across trials, the onset of the self-motion stimulus ranged from about 200 ms before to 200 ms after saccade onset. At the end of each trial, the saccade target and the ground plane stimulus were switched off and a ruler with a random sequence of numbers was presented on the screen. Ruler ticks were separated by 1°. Subjects had to indicate via keyboard input the number on the ruler that appeared closest to their perceived heading direction. Pressing the return button, started a new trial. **b** The panel shows schematically the timing of all events

experiment (perception of self-motion direction), it clearly shows that subjects were not virtually blind for brief self-motion stimuli as could have been the case due to saccadic suppression. Accordingly, we conclude that perisaccadic optic flow perception per se was only marginally modulated.

The neurophysiological data had provided evidence for an eye-centered reference frame of the compression effect. This leads to the hypothesis that the center of the compression should move with the eye but not the head, if both were changed independently from each other.

We decided to test this hypothesis in a further control experiment using a heading task instead of a forward/backward discrimination. Different from the main experiment, however, either (i) the subjects' head was directed $\alpha = 15°$ to the left, while gaze was still straight ahead, or (ii) the head was straight ahead, but eyes were directed $\alpha = 15°$ to the left (Fig. 10). Clearly, we observed a perisaccadic shift of perceived heading toward the screen center in the first condition (head to the left, Fig. 10a), but

a shift toward −15°, when the eyes were directed to the left (Fig. 10e). Hence, compression occurred in eye-centered coordinates toward the direction of gaze.

## Discussion

We have investigated the response properties of neurons in macaque extrastriate (area MST) and parietal (area VIP) cortex and developed a model to decode self-motion direction from neural discharges. This decoder performed veridically during fixation and slow eye movements, but it revealed a characteristic error in the temporal vicinity of saccades. This error pattern led us to hypothesize that perceived heading should be compromised by saccadic eye movements. A behavioral study in human observers confirmed this idea.

Visual perception is often far from being veridical. Instead, modulatory influences shape the processing and lead to a percept that can be significantly different from its physical counterpart[32, 33]. Eye movements are often the origin of such modulatory influences[34]. While eye-movement induced visual illusions lead to a (temporary) misperception of the outside world, they allow for a deeper understanding of the neural basis of visual processing in humans and in the animal model, i.e., the macaque monkey[9, 10, 20, 35]. In our current study, however, we took the opposite approach. We started off at the neural level of the macaque and correctly predicted a hitherto unknown visual illusion in humans. Hence, our physiological and behavioral results provide further strong support for the idea that areas MST and VIP are critically involved in the processing of self-motion information in both humans and monkeys. The functional similarities between the species extended from decoding neuronal responses in monkeys to a prediction for human self-motion processing that was confirmed behaviorally.

In the main behavioral experiment, self-motion perception during fixation was biased toward straight ahead, which here was also the direction of gaze. One could assume that in a Bayesian sense, this central bias would reflect a behavioral prior toward the most common heading direction in everyday life. A previous study has indeed investigated priors for heading perception[36]. Surprisingly, also to the authors, the observed prior was not centered on straight ahead. Instead, when testing heading in the full horizontal 2D motion space, two peaks directed 90° to the left and to the right, i.e., perpendicular with respect to straight ahead, evolved. In addition[37], showed that the centrifugal bias was more pronounced when heading could be in any direction within the full 2D horizontal plane as compared to an experimental context, in which heading was restricted within a 90° sector centered on straight ahead. This finding was in line with results from the previous studies[38, 39].

At first glance, our finding of a central bias appears at conflict with the results from refs. [36] and [37]. Yet, given the complete different experimental settings in these previous and our current study, it might not be appropriate to put both results in contrast. Stimuli in our study were extremely short (40 ms), while those employed in the other two studies lasted beyond one second. It is known, e.g., from literature on localization of briefly flashed stimuli that these are perceived closer to the fovea than they actually were[7, 40, 41]. Yet, when targets are presented continuously, over-estimation of perceived location has been reported[42]. In other words: in similar tasks, a bias can strictly depend on the experimental conditions.

References [36, 37] suggested that the observed centrifugal bias might result from a non-uniform distribution of heading preferences in primate areas MST and VIP, as described in the macaque among others, e.g., by Angelaki, DeAngelis and colleagues (MST[16] VIP[43]). In their study on area MST, these authors

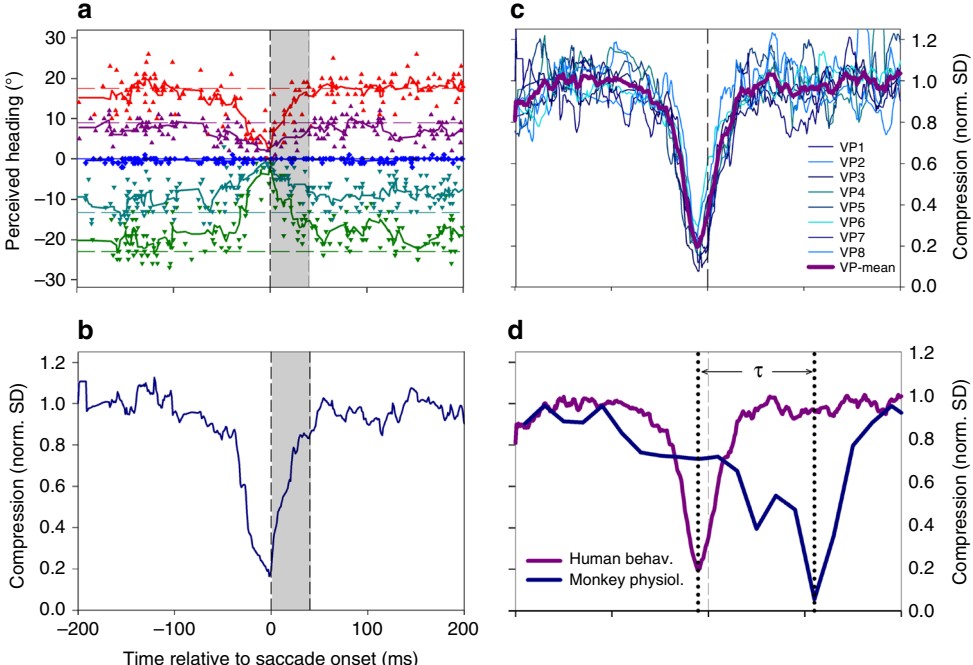

**Fig. 9** Time course of compression of perceived and decoded heading. **a** Responses from one example subject (VP01). Symbols represent data from single trials: upward pointing triangles for heading to the right (magenta: +15°, red: +30°), downward pointing triangles for heading to the left (dark cyan: −15°, green: −30°), and circles for heading straight ahead (blue, 0°). Solid lines represent running means of five consecutive samples each, assigned to the central sample value. Dashed lines show the performance for the same experiment during continuous fixation. **b** Compression, defined as the normalized standard deviation of the five time-courses of perceived heading for the same subject as **a**. Maximum compression as indicated by the minimum value of the normalized standard deviation was observed just prior to saccade onset, i.e., at $t = −1$ ms. **c** Time course of heading compression from the eight subjects (thin blueish lines) and the average compression of perceived heading (thick purple curve). **d** The neurophysiological data (blue data curve) followed the behavioral data (purple data curve) by $\tau = 116$ ms, as determined from cross-correlation of the two time-courses. This value of $\tau$ is consistent with visual response latencies as summarized in ref. [27]

have investigated accuracy and precision of decoded heading from discharges of monkey area MST and found a decoding bias toward the periphery. Indeed, long before and especially long after a saccade, also our decoding from areas MST and VIP revealed a (slight) centrifugal bias (Fig. 2). Accordingly, during slow eye movements, our data are in good agreement with previous results obtained during steady fixation.

In our data analysis, we have employed linear regression analysis for quantifying the neurons' tuning for self-motion direction. This approach is in line with previous studies[22]. Other studies have employed sigmoidal[15] or more complex tuning functions[16] to model heading selectivity. In our study, we have tested a limited range of heading directions (±30°). This range did not allow to determine (i) response saturation values, as required for fitting a sigmoidal function, or (ii) local maxima or minima, as would have been necessary for fitting a rather complex tuning function. Hence, given a set of three data points within the central ±30°, a linear function appeared to be a well-motivated and plausible statistical model, which allowed quantitative modeling of the effects of saccades on the representation of heading information.

In a first behavioral control experiment subjects had to discriminate whether a brief self-motion stimulus simulated forward or backward self-motion. It is known from the literature that saccades impair the perception of visual motion[29]. Given that self-motion sequences in our study were extremely short (40 ms) we had to exclude the possibility that subjects simply were not able to perceive our stimuli due to saccadic suppression of the magnocellular processing of the visual system[44]. Obviously, the nature of this control experiment was different from our main experiment, in which subjects had to indicate their perceived self-

motion direction. Nevertheless, the results of the control experiment clearly showed that subjects were not literally blind for the self-motion stimuli used in our experiment.

Our second behavioral experiment allowed us to determine the reference frame of perisaccadic compression of perceived heading. A previous study had found that, during fixation, perceived direction of visually simulated self-motion was biased toward eye but not head position[45]. Our findings are in line with this previous result. During steady fixation, the mean perceived heading was shifted in the direction of the eccentric eye position by approximately one-third of the gaze amplitude. This value is somewhat smaller than the one from ref. [45] (46%). Yet, like in ref. [45], a peripheral head position with eyes straight ahead did not induce a shift of the mean perceived heading. It was concluded that the perception of visually simulated self-motion is organized in eye-centered coordinates, consistent with neurophysiological data from the animal model, i.e., the macaque monkey (area MST[46]; area VIP[47]). Our perisaccadic data are perfectly in line with this hypothesis, which, in turn, provides further evidence for our idea that the observed compression of perceived heading might be based on response properties of neurons in primate areas MST and VIP.

Our data analyses revealed that the compression in the neurophysiological data was strongest 110 ms after saccade onset. This timing is consistent with visual response latencies as summarized, e.g., by Schmolesky et al.[27]. Visual onset response latencies in the macaque monkey vary from cell to cell and tend to increase from lower toward higher visual cortical areas[27]. Accordingly, a visual event occurring at a specific time induces a response in extrastriate or parietal areas starting roughly 100 ms later and peaking another 10–30 ms later. Taking this latency into

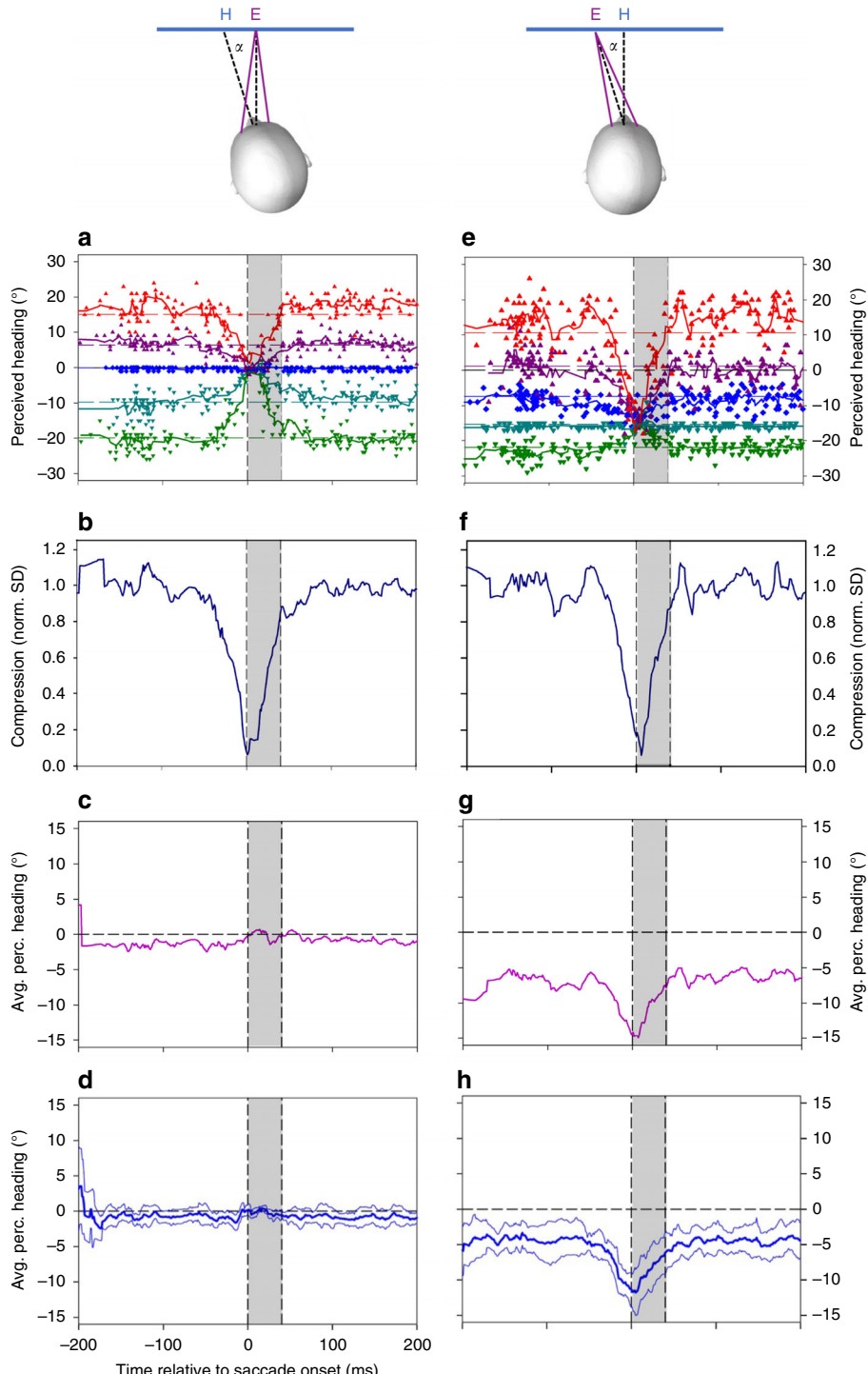

**Fig. 10** Reference frame of the compression of the representation of heading. Single-subject data. The subject performed a 10° upward saccade, and we presented a 40 ms long self-motion sequence. In **a**, **e**, each colored data-point indicates the onset of the self-motion sequence (*x*-value) and the perceived heading (*y*-value). The colored horizontal dashed lines in **a**, **e** indicate the subject's mean heading perception for a given stimulus direction during steady fixation. The subjects' head was either **a** directed $\alpha = 15°$ to the left, while gaze was still straight ahead (i.e., aligned with the body- and screen-midline), or **e** the head was straight ahead, but eyes were directed $\alpha = 15°$ to the left. We observed a perisaccadic shift of perceived heading toward the screen center in the first case (head to the left, **a**), but a shift toward −15°, when the eyes were directed to the left (**e**). The time course and strength of compression were identical in both cases (**b**, **f**). We determined also the time course of the average perceived position, i.e., the time course of the mean of all five heading percepts, which was rather stable over time in the head-turned condition (**c**). In the gaze-turned ead-straight condition, however, also long before and after the saccade, perceived heading was biased away from straight ahead, toward the direction of gaze. In the temporal vicinity of the saccade, the averaged perceived heading was in line with the direction of gaze (**g**). At the time of maximum compression, all headings were perceived as being 15° to the left. The findings for this subject were representative for the group of subjects, as indicated in **d**, **h**. The thick blue line indicates the time-resolved average value of perceived heading across all five subjects. The thin blue lines indicate the time-resolved standard error

account, the compression of perceived heading should be maximal for stimuli presented near saccade onset. This is exactly what we found in the psychophysical data. A quantitative comparison between the time courses of neural responses and those of the behavioral responses of humans yielded a temporal offset of $\tau = 116$ ms, closely matching the average latency. Similar results were obtained in a previous study, where we aimed to determine a neural correlate of saccadic suppression[20]. We found that time courses for visual responses to flashed stimuli in areas MST and VIP of the macaque were similar to the time courses found for saccadic suppression as measured in perceptual experiments in humans. Yet, while the maximum saccadic suppression as measured behaviorally typically is found at saccade onset, the peak of neural saccadic suppression was found roughly 100 ms later, i.e., with the expected latency. Accordingly, we consider the complex time-course of the perisaccadic responses in primate areas MST and VIP (selective suppression) to be the neural basis for the perisaccadic compression of perceived heading. It has to be mentioned, though, that correlation, as employed in our analyses, cannot prove causality. Such a strict proof as based, e.g., on reversible inactivation of both areas simultaneously, however, would have been beyond the scope of our study.

Our finding that the perception of heading from an optic flow field is compressed during a saccade is reminiscent of perisaccadic compression of perceived space, time, and numbers[48]. Like in these other perceptual illusions, the effect of a compressed perception of heading started well before the eyes started to move. This time course argues for an active visual process rather than a passive, purely visually induced effect. Similar conclusions have been drawn concerning other perisaccadic modulations of visual perception[6, 20, 28, 49–52].

While the temporal dynamics of the compression effects are rather similar across various affected measures (among them space, time, and number), our results cannot be explained by these compressions. Instead, they add a further dimension, namely that of self-motion. Space, time, numbers, and motion are all fundamental representations within primate parietal cortex. As deduced from our model, a common mechanism for these perisaccadic illusions may consist of selective modulation of activity in these representations by saccades.

## Methods

**Monkey physiology.** All procedures had been approved by the regional ethics committees (Regierungspräsidium Arnsberg; Regierungspräsidium Gießen) and were in accordance with the published guidelines on the use of animals in research (European Communities Council Directive 86/609/ECC). Experiments had previously been performed in three trained male macaque monkeys (C, H, and R)[18, 19]. We recorded neural activity from the medial superior temporal area (area MST) in two hemispheres of monkeys H and R and from the ventral intraparietal area (area VIP) in two hemispheres of monkeys C and H. Identification of areas was based on MR images, chamber position, electrode depth, preference for flow fields as well as large visual receptive fields. Eye movements were monitored with a search eye-coil system (Skalar Medical BV, Delft, the Netherlands), running at 500 Hz. Further details on recording procedures and animal preparation can be found in ref.[20].

Optic flow stimuli were generated in OpenGL with a frame-rate of 72 Hz. The stimuli consisted of large-field computer-generated sequences that were back projected via a video projector (Electrohome) onto a tangent screen, 48 cm in front of the monkey. The size of the projection covered the central 90° × 90° of the visual field. Optic flow sequences (15–20 s) simulated self-motion of a virtual observer over an extended horizontal plane covered with a texture pattern, located 37 cm below eye level (Fig. 1a). Trials simulated self-motion in one of the three directions: 30° to the left, straight ahead, and 30° to the right.

We recorded activities from 119 neurons in areas MST ($n_1 = 64$; 55 neurons from monkey H and 9 neurons from monkey R) and VIP ($n_2 = 55$; 36 neurons from monkey H and 19 neurons from monkey C) while the monkeys freely viewed long self-motion sequences, lasting between 15 and 20 s. In order to identify those neurons that are significantly tuned to heading, we split each trial in 1000 ms slices and computed the neuron's activity for each temporal slice. Then, we employed an ANOVA on ranks with two degrees of freedom (2 df) on the responses in these temporal slices for the three different self-motion directions.

In primates, self-motion stimuli induce reflexive, optokinetic-like eye movements[13, 14, 53]. For further data analysis, we determined periods of slow tracking and saccades. Saccade onset was determined offline by a speed criterion (80°/s). Tracking phases in which eye position was outside the central 20° of the screen were excluded from further analysis to avoid possible contamination of neural activity by eye-position signals or visually induced screen-border effects. From the remaining data set, all tracking phases were temporally aligned to the onset of the preceding saccade. Average neural activity in 20 ms bins was determined across these tracking phases for each of the three self-motion directions. During fixation, the tuning of neurons in areas MST[15, 16] and VIP[22] for forward-directed self-motion within the central ±45° can be approximated by 2D linear or sigmoidal functions. Given the sparse sampling of simulated horizontal forward self-motion in our experiment (−30°, 0°, and +30°), we used linear regression to approximate the cell's response to heading:

$$y(t) = a \times x(t) + b$$

Here, "$y(t)$" represents the time resolved neural activity, "$a$" depicts the slope of the regression function, $x(t)$ depicts the current heading direction, and $b$ represents the intercept of the regression function, or mean activity of the neuron. We used a brief, 150 ms post-saccadic interval (300–450 ms) of each neuron's response to the three simulated headings to compute the linear regression parameters.

We determined time-resolved decoded heading for each neuron for each of the three self-motion directions by relating activity in 20 ms bins to the linear regression, i.e., neural activity was directly translated into a heading estimate by solving the inverse equation

$$x(t) = (y(t) - b)/a$$

From the population of cells with a statistically significant tuning for heading we determined decoded heading as median of the heading values provided by each neuron. In steps of 20 ms, we employed an ANOVA on ranks (2 df, $P < 0.05$) together with a false discovery rate (FDR) correction to determine decoded headings that were significantly different from each other. A moving average of heading was determined as the average value of two consecutive values assigned to the time-point in the middle between the two samples. Following a procedure introduced in a previous study investigating the perisaccadic compression of perceptual space[7], we determined normalized compression as the standard deviation of the time-courses of the three decoded headings, divided by its mean value in two 100 ms windows. To make sure to not consider perceived heading directions which were under the influence of the saccade, we chose for the 100 ms wide windows prior to (−200 to −100 before saccade onset) and after the saccade (200–300 ms). The time of the maximum compression was determined as the time for which the minimum normalized standard deviation was observed.

We employed bootstrapping (Matlab function bootci, with 1000 repetitions) to determine the time-resolved 95% confidence intervals of neuronal response modulations (Fig. 3a, b) and of the precision of heading-decoding and -perception (Supplementary Fig. 2).

For the quantitative model on the role of saccadic suppression on heading-decoding, we convoluted the time courses of the artificial neurons with the response profiles deduced from global suppression or selective suppression, respectively (Fig. 3). To this end, we normalized the time courses of global or selective suppression, so that the values >100 ms before and after the saccade were centered on 1.0. In the one case, i.e., global suppression, all time courses of neural activity were convoluted with this global suppression function. In the other case, we applied selective suppression, i.e., we convoluted neurons with the suppression function in case of a congruent stimulus condition (blue curve) and with the release-from-inhibition function in the incongruent stimulus condition (magenta curve).

**Human psychophysics.** We recorded eye movements and behavioral data from eight human subjects (three female; 19–28 years). Experiments were approved by the local ethics committee at the Department of Psychology. Participants had normal or corrected to normal vision and gave their written informed consent prior to the experiments, which were in accordance with the Declaration of Helsinki.

Self-motion stimuli as well as fixation and saccade targets were generated with the Psychtoolbox[54] and back-projected via a video projector (Christies, DS + 6K-M, 120 Hz, 1152 × 864 pixels) onto a tangent screen, 70 cm in front of the subjects, covering the central 81 × 65° of the visual field. Stimuli, consisting of a random distribution of white dots (size: 21 arc min; luminance: 105 cd/m²), located 140 cm below eye level, simulated forward self-motion over a horizontal ground plane. Across trials, simulated self-motion was pseudo-randomized in one of five directions: to the left (−30° and -15°), straight ahead (0°), or to the right (15° and 30°). Self-motion stimuli consisted of five consecutive frames of 100% coherent dot motion, i.e., lasting 40 ms (Fig. 10) during which a total forward displacement of 3.3 m was simulated. The fixation target was presented at $[x, y] = [0°, 0°]$, the saccade target at $[x, y] = [0°, 10°]$. Eye position was recorded at 500 Hz by an EyeLink II system (SR Research, Ontario, Canada). Each trial started with a drift correction, followed by presentation of the fixation target and the stationary ground-plane stimulus. After 1600–2000 ms, the fixation target was switched off and the saccade target was switched on (until the end of the trial), inducing a visually

guided saccade of 10°. Across trials, the onset of the self-motion stimulus ranged from about 200 ms before to 200 ms after saccade onset. At the end of each trial, the saccade target and the ground plane stimulus were switched off and a ruler with a random sequence of numbers was presented on the screen. Ruler ticks were separated by 1°. Subjects had to indicate via keyboard input the number on the ruler that appeared closest to their perceived heading direction. Pressing the return button started a new trial. The timing of all events is shown in Fig. 8b. We recorded about 200 trials for each self-motion direction from each subject. Onset times of the self-motion stimuli were uniformly distributed relative to the step of the saccade target. Prior to each experiment, we had determined the average saccade latency of each subject. We corrected for this latency subject-wise, which resulted in an almost uniform distribution of stimulus onset times over the −200 to + 200 ms with respect to saccade onset. This allowed for a sampling of heading perception with approximately 2 ms temporal resolution (average temporal spacing between two samples across all subjects was 2.5 ms). In fixation trials, all visual events were identical, except that subjects had to keep fixation of a central target, which was presented throughout the trial.

In saccade trials, saccade onset was determined offline by a speed criterion (80°/s). All data (stimuli and eye movements) were aligned to saccade onset. For each subject, we first determined the time course of heading perception for each of the five self-motion directions. In line with previous studies[7], compression of heading was defined as the standard deviation of the time courses of the five perceived headings normalized with respect to its average value up to 100 ms before and after the saccade. The time of maximum compression, i.e., the time point for which the standard deviation reached its minimum value, was determined for each subject.

In order to determine the best overlap for the two time-courses of heading compression as obtained from the neurophysiological (monkey) and behavioral (human) data, we employed a temporal cross correlation analysis. The temporal shift between the two time-courses resulting in the highest cross-correlation value was considered the neuronal processing latency.

Motion perception across saccades is impaired (saccadic suppression)[29, 31]. Nevertheless, even brief periods of visual motion can be perceived perisaccadically[30]. Accordingly, in a first control experiment, we aimed to determine whether or not this reduction of the perisaccadic perception of motion had a critical influence on our behavioral data. To this end, we presented to five of the eight subjects the same self-motion stimuli as before, but now simulating straight forward or backward self-motion, again with motion onsets between 200 ms before and 200 ms after saccade onset. In a 2-AFC task, subjects had to indicate whether they had perceived forward or backward self-motion.

In addition, we determined effect sizes of the perisaccadic response modulation, i.e., $d'$. We computed $d'$ as difference of the means of (i) discrimination performance as obtained >50 ms before or after saccade onset $\overline{x}_{\text{pre\&post}}$ and (ii) perisaccadic discrimination performance $\overline{x}_{\text{peri}}$ (i.e., onset of the self-motion stimuli <50 ms before or after saccade onset), divided by the combined standard deviation of these two activity patterns $\text{Std}_{\text{combined}}$[55, 56].

$$d' = \frac{\overline{x}_{\text{pre\&post}} - \overline{x}_{\text{peri}}}{\text{Std}_{\text{Combined}}}, \text{ with } \text{Std}_{\text{combined}} = \sqrt{\frac{(n_1 - 1)\text{std}_1^2 + (n_2 - 1)\text{std}_2^2}{n_1 + n_2 - 2}}$$

Here, $n_1$ and $n_2$ are the number of samples in the pre- and post-saccadic intervals ($n_1$) as well as in the perisaccadic interval ($n_2$). std$_1$ and std$_2$ are the standard deviations of these samples.

In a second control experiment, we aimed to determine the reference frame of the compression of perceived heading. Here, like in the main experiment, subjects performed a 10° upward saccade, and perisaccadically, we presented a 40 ms long self-motion sequence. Different from the main experiment, however, either (i) the subjects' head was directed $\alpha = 15°$ to the left, while gaze was still straight ahead, or (ii) the head was straight ahead, but eyes were directed $\alpha = 15°$ to the left. This experiment tests four possibilities for the center of compression. If compression is directed to the head's midline (variant A), one would expect a shift of the center of compression in the first case (head to the left), but not in the second (eyes to the left). If compression is directed toward the body midline (variant B) or the screen center (variant C) one would expect that the center of compression is always at the center of the screen. If compression is directed toward the center of gaze (variant D), one would expect compression toward the screen center when gaze was straight ahead and toward a position 15° to the left when gaze was at that position.

**Code availability**. MATLAB code used for analysis is available from the corresponding author on reasonable request.

**Data availability**. The data sets generated and analyzed for the current study are available from 10.5281/zenodo.837063.

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

## Acknowledgements

This work was supported by the Deutsche Forschungsgemeinschaft (DFG) Collaborative Research Center CRC/TRR-135 and Research Unit RU-1847. We thank Dirk Hofmann and Philipp Hesse for help with the behavioral experiments. We also thank the three reviewers for their valuable comments on the original version of the manuscript.

## Author contributions

F.B. and M.L. designed the research; Neurophysiological data had been recorded before[18, 19]; J.C. collected part of the behavioral data; F.B., J.C., and M.L. supervised the collection of the remaining behavioral data; F.B. created the decoder and the quantitative model and analyzed the neurophysiological and the behavioral data; and F.B., J.C., and M.L. wrote the paper.

## Additional information

**Competing interests:** The authors declare no competing financial interests.

