## [Peer Review File · Nature Communications]

Reviewers' comments:

Reviewer #1 (Remarks to the Author):

The paper reports a pair of findings, one in monkey physiology and the other in human perception. The physiology finding, boiled down to its essence, is that MST neurons reduce their firing rates during saccades. The human perceptual finding is that reports of self motion tend towards straight ahead during a similar epoch surrounding a saccade. Both appear to be well measured (though the N on the physiological result is very small; see below), but I had difficulty with the interpretation, that there is a compression of space during saccades. There is a much more parsimonious explanation in the well-known phenomenon of saccadic suppression. However, if the simpler explanation can be excluded more forcefully, this is potentially a quite novel and interesting pair of results.

Major points:

1) Interpretation: physiology. Because of the linear regression analysis, any change in rate that brings a neuron toward its baseline will bring the estimated heading closer to zero, so we can interpret the result as a consistent, transient reduction in firing of the neurons in the sample. The timing of this reduction is roughly consistent with saccadic suppression measured in MST and other areas. It would be possible in principle to distinguish a true compression of space from simple suppression, but the decoder in use in this paper will not. One would need to detect the presence of saccades and present controlled probe stimuli during the saccade to measure tuning at this exact moment. Tuning shifts would support a compression hypothesis, but a gain change would not.

2) Interpretation: psychophysics. Here the authors are on considerably stronger ground, since the data can literally be described as a central bias during saccades. However, once again there is an alternate and parsimonious hypothesis that needs to be excluded. In most perceptual measurements, when sensory evidence is weak, subjects rely on their priors. To my knowledge, no one has attempted to estimate priors for heading perception, but one would intuit based on first principles that they would be for straight-ahead motion, since this is how observers spend most of their time in motion.

2a) Control experiment. The crux of their argument lies in the control experiment, where subjects are discriminating forward from backward headings. This is potentially flawed in a couple of important ways, but the description is quite brief. The discrimination is quite different from the main experiment, and can be solved using local motion cues anywhere in the scene, since it boils down to a discrimination between opposite directions. So, potentially very different strategies could be used by the subjects, relying on totally different physiological mechanisms. It is also very easy, so the subjects are very near maximum performance. In this area of a psychometric function, many of the errors are probably "lapses", rather than actual sensory mistakes. But even if they are sensory mistakes, the difference between 94% and 96% performance can be quite large in terms of d' or some similar measure. So, it is a very insensitive control experiment. Lastly, the analysis was unpaired, and a Wilcoxon test should probably have been used.

2b) Analysis in Supplementary Figure 1. This is probably the best argument against the saccadic suppression hypothesis, but I do not think it particularly strong either. It is possible that a proper Bayesian analysis of this data and the data in Figure 4 would be compelling; the argument depends on

the variance of the prior distribution. It can be estimated from the main experiment data, and if it is large, then this case is pretty good. But I would need to see the analysis to be sure. It would need to be performed for each subject, and all the individual subject data presented. The subject in Figure 4 appeared to both have a central and a right bias, which would be estimable with a full analysis.

3) Physiology sample. Finding only 40 neurons from 119 that meet the very lax criterion for tuning that they used is peculiar. But there is no breakdown of the sample or the principal results by monkey or area. This needs to be provided in a supplementary table at the very least. They probably don't have the power to test whether the results are different in VIP or MST, but this is a very interesting question. Assuming the two areas are the same is not a bad first guess, but by no means certain. If at all possible, they should increase this sample size; it would make readers much more confident in the results. I recognize that this is a re-analysis of archived data, but I would hope they could boost the N in some way, either by relaxing the inclusion criteria or even by recording new data.

4) Saccade direction effects. It doesn't make any sense to this reviewer that saccades would compress space isotropically. This is certainly not the case for position reports of flashed stimuli. However, because all saccades were treated the same, it is impossible to know if there was any direction dependence of the effects on the physiology. This is an important question because the saccades in the physiology experiments and the perceptual experiments were very different. Ideally, one would like to see a subdivided analysis between vertical saccades (which were used in the psychophysics), and those with left or right horizontal components. But if the results really are isotropic, then it supports the saccadic suppression hypothesis.

Lesser issues:

5) The timecourse data need to be presented on the same axes, relative to the start of the saccade, without adjustment for latency.

6) No mention is made of the system used to measure eye movements in the human subjects; this needs to be included in the methods. Most video systems include a fixed or variable latency between movement and report. These specs should be included, because they are important to the time-series analysis that is central to this paper.

7) Title. The title makes little sense as written, since heading is a behavior, and the subjects were not heading anywhere. I suggest "heading representations" instead.

Reviewer #2 (Remarks to the Author):

The current paper reports complimentary human psychophysical and macaque neurophysiology experiments showing that heading perception and neural response, respectively, are compressed toward straight ahead around the time of a saccade. The great strengths of the paper are the following: 1) the reported heading compression is a novel and significant finding, 2) this phenomenon is observed in both human psychophysics and monkey neurophysiology.

Presentation and/or discussion could be improved regarding the following points: 1) can this compression be explained as a by-product of previously reported saccadic suppression/modulation of motion processing/perception? 2) The temporal offset between perception and neurophysiological responses needs to be discussed in greater detail; in particular, the claim that MSTd and VIP activity is the neural substrate of the observed perceptual compression would seem to be undermined by the

fact that the perceptual compression PRECEDES the observed compression of neural tuning.

Minor comments:

32: veridically instead of veridical

103: Please explain in greater detail how statistical significance of tuning was determined. What are the samples that are passed to the ANOVA on ranks? Was it mean firing rate during the 300-450 interval on a given trial? How many trials were collected per neuron? Only 40 of 120 neurons showing significant tuning seems rather a low proportion for these areas (1/3). How does this compare with other reports in the literature that tested for significance of tuning?

126: Please make absolutely clear in the figure and/or figure caption and in text that relative compression is the same as normalized standard deviation. When different terms are used in text and axis labels, it gets confusing.

155-156: Clarify language: onset times were uniformly distributed (were they?) over the -200 to +200 ms allowing a sampling of heading perception with 2 ms temporal resolution.

177: Please clarify in the text the direction of the shift. This offset between perceptual and physiological responses is puzzling and deserves to be discussed in greater detail. If I understand correctly, maximum compression for perception occurs around the time of the saccade, while max compression for neural responses is 90 ms later. If the claim is that MSTd and VIP are the neural substrate for the perceptual response, why does the perceptual response precede the neural one?

181: This impairment of motion perception and its possible relation to the current results also deserves to be discussed in greater detail. It seems possible that a modulation of lower-level motion processing, something short of complete motion blindness, could be responsible for the heading compression observed here. The results of the control experiment do not rule this out.

201: "evolve most likely not on purpose" please rephrase to improve English

222: Indeed, this effect looks like regression to the mean. This can happen at a neural level, i.e. around the time of a saccade neural responses tend toward the baseline firing rate. This can also happen at the decision level, i.e. regression toward the mean psychophysical response, namely the middle of the response line, since presented heading were symmetric around 0 (-30, -15, 0 +15, +30). Can your data differentiate between these alternative interpretations?

260-61: So intercept is mean activity of the neuron, and compression is regression to the mean.

275: why mean values over the before/after interval that are not symmetric relative to time of saccade. How was this choice made?

309: please specify/repeat how the time course was reconstructed.

317: please specify direction of temporal shift

320: please explain this impairment in more detail and speculate how it could or could not be responsible for the observed effects

331: Is it fair to say that a more appropriate control would have been to ask subjects to discriminate

between headings of +30 and -30? Again, I do not see how the results of the control rule out the possibility that this is a corollary effect downstream from saccadic modulation of low-level motion processing.

Reviewer #3 (Remarks to the Author):

It is known that many neurons in MST and VIP are tuned to heading direction and the spatial tuning of these neurons can be determined using a linear regression technique. This paper it is noted that neurons in MST and VIP greatly decrease their activity after a saccade in monkeys. We the heading is decoded during this period of decreased activity it was greatly compressed after the saccade with a maximum effect about 80 ms after the saccade onset. The study went on to see if this compressed heading after a saccade was actually perceived in humans. Not only are the findings novel, but the approach is novel as well since usually perceptual illusions are known decades before the underlying neurophysiology is understood. In this instance the authors first looked at the neurophysiology the found the predicted illusion, event though this illusion had not been previously described.

In Fig. 1C what is shown is a decrease in activity during the period after the saccade and this brief suppression is also mentioned in the text. Was it generally the case that activity was suppressed during this period or was that just for this example? If it was generally suppressed then could this be a mechanism by which heading estimation was essentially suppressed during this period to prevent confusion due to motion on the retina due to eye motion during the saccade interfering with heading perception? One could imagine that the general suppression might cause the heading estimates to be compressed around straight ahead because there are many more neurons with best sensitivities occurring near straight ahead (Gu et al., 2010). Is it possible that this is actually what is occurring?

One question I have is why does this occur? Is it a mechanism to suppress potentially confounding visual stimuli that occur during the rapid eye movement of the saccade? I realize this is not something that can be known from the current data but it may be worth some speculation: One interesting phenomenon is that during visual heading estimation a population vector decoder model of neuronal activity in MST suggests (Gu et al., 2010) the horizontal component visual headings will be overestimated. This effect has been minimal when headings choices only cover a limited range that is known to or quickly learned by the subjects as done here and in other prior studies (D'Avossa and Kersten, 1996; Telford and Howard, 1996). However when the full range of visual headings is possible these headings tend to be overestimated (Crane, 2012; Cuturi and Macneilage, 2013) consistent with the predictions made from MST neurophysiology. Additionally visual headings are in retinotopic coordinates in MST and these coordinates also seems to be preserved in perception (Crane, 2015). Of course many naturally occurring situations, people and animals tend to orient their eye position so that it is centered near the focus of expansion (FOE), which for a straight path is the heading direction which would tend to minimize the overestimation of eccentric headings. Although this is not true of the experimental situation, in many situations a saccade would bring the eye away from the FOE or back to it. In either of these situations it may be advantageous to have the eccentricity of the heading minimized: In the away situation because the heading is actually being maintained but the eye is making a saccade to an eccentric target. In the toward situation because the heading is now being minimized. I admit this is kind of a kludge solution which would not be advantageous in every solution.

The numbering of the figures doesn't make sense. Instead of Figs 1-4, then supplemental Fig. 1, why not just have Figs 1-5?

Overall I found this to be a very high quality report and I would endorse publication.

Crane BT (2012) Direction Specific Biases in Human Visual and Vestibular Heading Perception. PLoS One 7:e51383.

Crane BT (2015) Coordinates of Human Visual and Inertial Heading Perception. PLoS One 10:e0135539.

Cuturi LF, Macneilage PR (2013) Systematic biases in human heading estimation. PLoS One 8:e56862.

D'Avossa G, Kersten D (1996) Evidence in human subjects for independent coding of azimuth and elevation for direction of heading from optic flow. Vision research 36:2915-2924.

Gu Y, Fetsch CR, Adeyemo B, Deangelis GC, Angelaki DE (2010) Decoding of MSTd population activity accounts for variations in the precision of heading perception. Neuron 66:596-609.

Telford L, Howard IP (1996) Role of optical flow field asymmetry in the perception of heading during linear motion. Percept Psychophys 58:283-288.

Signed: Benjamin Crane

We thank the reviewers for their extremely valuable comments which critically helped us to significantly improve our manuscript. We respond to the comments point-by-point by using the following fonts/styles to indicate:

Reviewers' comments (Calibri, bold, italics)

`Our response to these comments (Courier)`

New text in the revised, red-lined version of the manuscript. (Calibri, red)

Our revisions were quite substantial. Furthermore, a number of concerns raised by the reviewers were similar or overlapping. Examples are questions (i) about the role of saccadic suppression for the compression of perceived heading, or (ii) about the timing of the peak of compression in the neurophysiological and behavioral data sets. Typically, in a response letter, we not only respond to each concern individually, but we also provide the new text of the revised manuscript with our response to this concern. Yet, given the complete set of new analyses and new text, and given some overlapping questions, this would have made this response letter way to lengthy. Accordingly, we decided to refer to our response in case the same (or at least similar) question had been asked by another Reviewer before. We hope for the understanding of the reviewers. In order to facilitate navigation throughout our reply, we have numbered the questions asked by the reviewers and refer to these numbers in our reply. As an example, Questions 3 asked by Reviewer 2 will be referred to as **R-2,Q-3**.

Reviewer #1 (Remarks to the Author):

The paper reports a pair of findings, one in monkey physiology and the other in human perception. The physiology finding, boiled down to its essence, is that MST neurons reduce their firing rates during saccades. The human perceptual finding is that reports of self motion tend towards straight ahead during a similar epoch surrounding a saccade. Both appear to be well measured (though the N on the physiological result is very small; see below), but I had difficulty with the interpretation, that there is a compression of space during saccades. There is a much more parsimonious explanation in the well-known phenomenon of saccadic suppression. However, if the simpler explanation can be excluded more forcefully, this is potentially a quite novel and interesting pair of results.

Major points:

R-1,Q-1: 1) Interpretation: physiology. Because of the linear regression analysis, any change in rate that brings a neuron toward its baseline will bring the estimated heading closer to zero, so we can interpret the result as a consistent, transient reduction in firing of the neurons in the sample. The timing of this reduction is roughly consistent with saccadic suppression measured in MST and other areas. It would be possible in principle to distinguish a true compression of space from simple suppression, but the decoder in use in this paper will not. One would need to detect the presence of saccades and present controlled probe stimuli during the saccade to measure tuning at this exact moment. Tuning shifts would support a compression hypothesis, but a gain change would not.

We thank Reviewer 1, but also Reviewers 2 and 3, for making this point, which guided us to perform a much more thorough analysis of the perisaccadic modulation of neural activity. Based on this analysis and the results, we come up with a qualitative and a quantitative model of how selective perisaccadic response modulation leads to a compression of the representation of heading.

As pointed out by all three reviewers, at first glance it appears likely that the compression of the representation of heading, as observed in the decoding of neural activity from monkey areas MST and VIP, might simply be due to an overall perisaccadic reduction of neuronal excitability, i.e. saccadic suppression.

In order to test for the reviewers' concerns, we compared, as a first step, the time courses of (i) saccadic suppression of the population activity from areas MST and VIP and of (ii) the compression as determined from our decoding approach. These time courses were indeed very similar, although not identical. As detailed below, the maximum of saccadic suppression as determined from the time-point of the lowest mean perisaccadic neural population activity was observed 80 ms after saccade onset. This is shown in the figure below, which is the new Figure 3 in the revised version of our manuscript. Here, saccade onset and time-point of maximum suppression are indicated by the first and second vertical dashed line. The time-point of maximum compression of decoded heading, however, was observed 110 ms after saccade onset (see Figure 2 in the manuscript). That time-point is indicated by the third dashed vertical line in the figure below. This might be considered first evidence that a global saccadic suppression cannot account for the observed compression of decoded heading.

Figure 3: Average response modulation at the population level.

Figure 4: Qualitative model of the role of saccadic suppression on heading decoding

As a second step, we built a rather qualitative model, in which we considered a grand population of neurons from areas MST and VIP to be split up into two subpopulations: neurons that were tuned for leftward headings constitute "Subpopulation 1". By definition, the slope m of the regression function fitted to the heading responses of these neurons is smaller than zero ($m < 0$). "Subpopulation 2" comprises those neurons that were tuned for rightward headings ($m > 0$). This procedure is graphically illustrated in panel A of the figure above, which has become the new Figure 4 in the revised version of our manuscript. Published data predict that from such a grand population of MST and/or VIP neurons, roughly the same number of neurons should fall into one of the two subpopulations (e.g.: Duffy and Wurtz, 1991; Lappe et al., 1996;

Bremmer et al., 2002; Britten, 2008; Maciokas et al., 2010). Indeed, in our dataset of $n=71$ neurons (see details below), $n_{\text{Subpop1}} = 38$ neurons were tuned for leftward self-motion ($m < 0$), while $n_{\text{Subpop2}} = 33$ were tuned for rightward self-motion ($m > 0$).

As stated also in the original manuscript, we employed linear regression as approximation for the neurons' response tunings. Hence, in our decoding approach, neural activity directly translates into heading estimates. We applied this linear decoder also to our qualitative model. This is illustrated in panel B. Here, we assumed presentation of a rightward self-motion stimulus. Accordingly, the activity of the neurons tuned for rightward heading is high (Subpopulation 2, light blue line), while the activity of neurons tuned for leftward heading is low (Subpopulation 1, pink line). As an important side note: the activity level of the leftward tuned neurons during presentation of a rightward heading stimulus is smaller than during straight-ahead motion. This reduced activity might result, at least in part, from inhibition. In any case, the observed activity distribution of neurons from both subpopulations translates into a veridical heading estimate, i.e. self-motion to the right, as indicated graphically by the purple arrow.

In the next and final step of our qualitative model, we applied perisaccadic modulation of neural activity, i.e. saccadic suppression. As a first hypothesis, we presumed that saccadic suppression acts equally on both subpopulations (Global suppression, Panel C, left sub-panel). Accordingly, this suppression would reduce activity in both subpopulations alike. Given the tight relationship between activity and represented heading, however, this response reduction would have two different, counteracting effects in the two subpopulations. For those neurons being stimulated with their preferred heading direction (Subpopulation 2, blue line), the decoded heading would shift toward straight-ahead, which would be in line with a compression of the represented heading. Yet, for subpopulation 1 (pink line), reduced activity would result in a heading estimate further away from the center, i.e. more towards the periphery. Given that in our decoding approach heading is determined from averaging heading estimates across neurons from both subpopulations, decoded heading (purple arrow) would stay more or less the same, yet, with an increase in variance, i.e. a reduced decoding precision. As an alternative (Selective suppression, right subpanel), suppression might act selectively only on those neurons being driven by the self-motion stimulus, i.e. here: rightward heading for rightward tuned neurons. We call this a congruent condition. As reported in the literature, neural activity in motion sensitive areas MST and VIP often gets significantly reduced compared to baseline (i.e. spontaneous activity) when non-preferred stimuli are presented (e.g.: Lappe et al.,

1996; Bremmer et al., 2002). We call this an incongruent condition. It has been suggested that such response modulation results from mutual inhibition of neurons being tuned to different directions of (self-)motion (e.g. Ross and Dickinson, 2007). This idea suggests, that neurons, which receive inhibitory input from those neurons being driven (close to) optimal by a current stimulus, should reveal (probably with a certain delay) a release from inhibition when the optimally driven neurons are less activated (or suppressed e.g. by a saccade). Importantly, an increase of activity of the weakly activated cells due to a release from inhibition would cause a decoding shift towards straight ahead. Accordingly, both response modulations would cause the decoded heading to be shifted towards straight ahead (i.e. a compression of the representation of heading), without modulation of the precision of the decoded heading (purple arrow in the right panel, Compression).

Figure 5: Response modulation at the single cell level.

We were excited to test this hypothesis of selective suppression and release from inhibition on our data set. As a first step, we had a closer look at the individual response histograms of the population of neurons. Indeed, there was evidence of neurons showing a perisaccadic suppression of responses for one movement direction and (slight) perisaccadic response enhancement for the opposite direction. Two examples are shown in the figure above, which has become the new Figure 5 in the revised manuscript. The responses in the upper row (A) show data from a neuron tuned for leftward self-motion, as indicated by the pre-

and post-saccadic activities. While there was a slight perisaccadic suppression of responses for leftward self-motion (congruent stimulation condition), there was a slight perisaccadic response enhancement for rightward self-motion (incongruent stimulation condition). The bottom row (B) shows data from a neuron tuned for rightward self-motion. In line with our hypothesis, this neuron revealed a strong saccadic suppression during rightward heading (congruent condition), but a small perisaccadic response enhancement (or release from inhibition) during leftward heading (incongruent condition).

In order to quantify this effect at the population level, we assigned the response of each neuron to leftward and rightward heading either to the congruent or incongruent condition, dependent on the neuron's tuning for heading. Importantly, each neuron contributes to both conditions: leftward tuned neurons contribute to the congruent condition during leftward self-motion and to the incongruent condition during rightward self-motion, and *vice versa* for rightward tuned neurons. Then, we tested for a significant difference of the perisaccadic response modulations in the congruent and incongruent condition by comparing the neural activity in a fixed, post saccadic window $t = 40 \text{ ms} - 160 \text{ ms}$ after saccade onset. In Figure 5 above, in each panel, saccade onset is indicated by the vertical, solid line. The time window for analysis [40 ms - 160 ms] is indicated by the pair of vertical red dashed lines in each panel.

Figure 6: Modulation index for congruent and incongruent stimulation

This window was considered wide enough to capture both, suppressive and dis-inhibitory effects. We compared the neural activity in this time window with activity long after the saccade (300 ms to 450 ms after saccade onset) by computing the ratio of the two, defining a modulation index. Based on this definition, an index of 1.0 indicates no perisaccadic change in activity. An index smaller than 1.0 indicates response suppression, while an index larger than 1.0 indicates response enhancement. Our prediction from the hypothesis detailed above was an index below 1.0 for the congruent condition and an index above one for the incongruent condition. The figure above, which is the new Figure 6 of the revised manuscript, gives the result for the population of $n = 71$ neurons. Each data point represents data from a single neuron, with the x-value showing the index in the congruent condition and the y-value showing the index of the same neuron in the incongruent condition. As can be easily seen, the majority of data points lies above the diagonal. For the population of cells, the median of the index was significantly smaller than 1.0 ($0.812 \times 81.2\%$, one-sided signed rank test, $p < 0.05$) in the congruent condition, and significantly larger than 1.0 in the incongruent condition ($1.206 \times 120.6\%$, one-sided signed rank test, $p < 0.05$). 95% confidence intervals as determined from bootstrapping (red lines, Matlab function *bootci* with 1000 repetitions) were below 1.0 in the congruent condition and above 1.0 in the incongruent condition.

Figure 7: Time course of neuronal response modulation during congruent and incongruent stimulation.

While this finding is fully in line with our hypothesis of a selective suppression and enhancement or release from inhibition, the computation

based on a fixed analysis window (40 ms – 160 ms after saccade onset) is a rather coarse measure. Furthermore, a normalization can result in large indices even for minimal response changes in case of overall low activity (as shown in Figure 5B). To overcome this limitation, we assigned the response of each neuron from our recorded sample of MST and VIP neurons to leftward and rightward heading stimuli to the congruent or incongruent condition and determined the full time-course of the population response. The result is shown in the figure above, which is the new Figure 7 of the revised version of the manuscript. Indeed, when neurons were stimulated with their preferred heading direction (*Congruent* condition, left panel, blue curve), a saccadic suppression of activity was observed. In contrast, when the same neurons were stimulated with their non-preferred heading (*Incongruent* condition, right panel, purple curve), no saccadic suppression was observed. Instead, a small but significant response enhancement occurred. Data in both panels were obtained as follows: first we subtracted at the single cell level the mean activity in the response window (from -200 ms to 450 ms relative to saccade onset) from the time course of activity. These response modulations at the single cell level were averaged across the whole population of neurons (n=71) as tested in the congruent and incongruent condition. In both panels, the solid lines indicate this time resolved response modulation, while the colored shaded areas indicate the 95% confidence interval as determined from bootstrapping (Matlab function *bootci*, 1000 repetitions).

Indeed, as hypothesized, the peak of the enhancement (or release from inhibition) in the incongruent condition (120 ms after saccade onset) occurred 40 ms after the peak of suppression in the congruent condition (80 ms after saccade onset). In both panels, both events are indicated together with the time of saccade onset by vertical dashed lines.

Figure 8: Convolution kernel representing global or selective suppression.

In a final step, we modelled the effect of saccadic suppression in a quantitative model. The major advantage of this approach is that one can test various forms of perisaccadic response modulation and determine, if and if so which form of suppression reproduces the effect of a compression of decoded heading. For this model, we created a set of $n=71$ “virtual neurons”, i.e. time courses of neural activity. The activity levels were modelled after our real data set, i.e. activity distribution across the population was identical to our real data set. We then added noise to the activity profiles. It turned out, however, that the exact noise level was not important for the outcome of our analysis. We simulated presentation of five self-motion directions (-30° , -15° , 0° , 15° , and 30°). These artificial neurons were linearly tuned to self-motion direction, with the distributions of strengths of the tunings modelled after our real data set. Based on these overall settings, we generated 650 ms long time courses of stimulus driven activity. The temporal mean of these time courses was used to train the heading decoder. In a final step, we convoluted the time courses with the response profiles deduced from global suppression or selective suppression, respectively. To this end, we normalized the time courses of global or selective suppression as indicated by the figure above, which is the new Figure 8. In the one case, i.e. global suppression, all time courses of neural activity were convoluted with this global suppression function. In the other case, we applied selective suppression, i.e. we convoluted neurons with the suppression function in case of a congruent stimulus condition and with the release-from-inhibition function in the incongruent stimulus condition. We repeated this procedure fifty times for each of the two forms of saccadic suppression, i.e. global and selective, and computed the average time courses of decoded heading.

Figure 9: Quantitative model on the role of saccadic suppression on heading decoding.

The results are shown in the figure above, which is also Figure 9 of the revised version of our manuscript. As can be easily seen, heading could in principal be decoded from the responses of these artificial neurons, as indicated by the solid lines (decoded heading) superimposing the dashed lines, which in turn represent the real heading. Importantly, when applying global suppression (left panel), there was only a marginal modulation of decoded heading shortly after the saccade. Yet, this modulation was clearly different from the decoding pattern obtained from our real neurons (Figure 2). On the contrary, selective suppression, i.e. suppression of the neurons in the congruent condition combined with a release from inhibition of the neurons in the incongruent condition (right panel), triggers a compression of decoded heading, with a maximum of suppression around 100 ms after saccade onset.

We conclude from this set of new findings, which was triggered by the reviewers' comments, that (i) from a theoretical point of view, a compression of heading cannot result from a globally acting saccadic suppression. Instead, (ii) compression results from selective suppression in one set of neurons and enhancement of activity in the complementary set of neurons. It is exactly these activity patterns which we observed in our sample of neurons.

We describe this whole collection of new results in the revised version of the manuscript as follows:

Results: "Given the results shown in Figure 1C, we next investigated whether the compression of the representation of heading, as observed in the decoding of neural activity from monkey areas MST and VIP, could merely be a by-product of an overall perisaccadic reduction of neuronal excitability, i.e. saccadic suppression. Such saccadic suppression has been shown previously for areas MST and VIP 20. Hence, we determined the time course of the average neural activity, aligned to saccade onset. To this end, we first computed for each neuron its time-resolved discharge in the above-mentioned window [-200 ms 450 ms] and subtracted the mean activity in this window. This allowed us to determine a neuron's perisaccadic response modulation, which we then averaged across all 71 neurons. The result is shown in Figure 3.

Figure 3 about here

The solid black line depicts the mean response modulation; the grey surround indicates the 95% confidence interval as determined from bootstrapping. The time courses of saccadic suppression, as shown here, and the time course of compression of decoded heading as shown in Figure 2B were indeed very similar, although not identical. Maximum suppression was observed 80 ms after saccade onset, i.e. 30 ms earlier than the maximum compression. This might be considered first evidence that a mere saccadic suppression cannot account for the observed compression of the decoded representation of heading.

To further understand the neural basis of the perisaccadic decoding error, we built a rather qualitative model. Here, we considered a grand population of neurons from areas MST and VIP to be split up into two subpopulations: neurons that were tuned for leftward headings constituted “Subpopulation 1”. By definition, the slope m of the regression function fitted to the heading responses of these neurons was smaller than zero ($m < 0$). “Subpopulation 2” comprised those neurons that are tuned for rightward headings ($m > 0$). This procedure is graphically illustrated in Figure 4A. Published data predict that from such a grand population of MST and/or VIP neurons, roughly the same number of neurons should fall into one of the two subpopulations (e.g. 15,17,21–23). Indeed, in our dataset of $n=71$ neurons, $n_{\text{Subpop1}} = 38$ neurons were tuned for leftward self-motion ($m < 0$), while $n_{\text{Subpop2}} = 33$ were tuned for right-ward self-motion ($m > 0$).

Figure 4 about here

For our decoding approach, we had employed linear regression as approximation for the real neurons’ response tunings. We applied this linear decoder also to our qualitative model. This is illustrated in Figure 4B. Here, we assumed presentation of a rightward self-motion stimulus. Accordingly, the activity of the neurons tuned for rightward heading is high (Sub-population 2, light blue line), while the activity of neurons tuned for leftward heading is low (Subpopulation 1, pink line). The observed activity distribution of neurons from both subpopulations translates into a veridical heading estimate, i.e. self-motion to the right, as indicated graphically by the purple arrow.

In the next and final step, we presumed perisaccadic modulation of neural activity, i.e. saccadic suppression. As a first hypothesis, we assumed that saccadic suppression acts equally on both subpopulations (Global suppression, Figure 4C, left panel). Accordingly, this suppression would reduce activity in both subpopulations alike. Given the tight relationship between activity and represented heading, however, this response reduction would have two different, counteracting effects in the two subpopulations. For those neurons that are stimulated with their preferred heading direction (Subpopulation 2, blue line), the decoded heading would shift toward straight-ahead, which would be in line with a compression of the represented heading. Yet, for subpopulation 1 (pink line), reduced activity would result in a heading estimate further away from the center. Given that in our decoding approach heading is determined from averaging heading estimates across neurons from both sub-populations, decoded heading (purple arrow) would stay more or less the same, yet, with an increase in variance, i.e. a reduced decoding precision. As an alternative (Selective suppression, right subpanel), suppression might act selectively only on those neurons that are driven by the self-motion stimulus, i.e. here: rightward heading for rightward tuned neurons. We call this a congruent condition. It is well known that neural activity in motion sensitive areas MST and VIP often gets significantly reduced compared to baseline (i.e. spontaneous activity) when non-preferred stimuli are presented (e.g.15,22). We call this an incongruent condition. It has been suggested that such response modulation results from mutual inhibition of neurons that are tuned to different directions of (self-)motion (e.g. 24). This idea suggests, that neurons, which receive inhibitory input from those neurons that are driven (close to) optimal by a current stimulus, should reveal (probably with a certain delay) a release from inhibition when the optimally driven neurons are less activated (or suppressed e.g. by a sac-cade). Importantly, an increase of activity of the weakly activated cells due to a release from inhibition would cause a decoding shift towards straight ahead. Accordingly, both re-

sponse modulations would cause the decoded heading to be shifted towards the center, thereby leading to a compression of the representation of heading without modulation of the precision of the decoded heading (purple arrow in Figure 4C, right panel, Compression).

Figure 5 about here

This qualitative model led us to hypothesize the occurrence of selective suppression and release from inhibition in our data set. In order to test for our hypothesis, we analyzed in greater detail the individual heading responses of the neurons in our population. Indeed, we found evidence for a suppression of responses for one movement direction and a slight response enhancement for the opposite direction. Two examples are shown in Figure 5. The responses of the first neuron (A) revealed a significant tuning for leftward self-motion, as indicated by the pre- and post-saccadic activities. While there was a slight perisaccadic suppression of responses for leftward self-motion, there was a perisaccadic response enhancement for rightward motion. An opposite tuning is shown in the bottom row (B). This neuron was clearly tuned for rightward motion. In line with our hypothesis, this neuron revealed a strong saccadic suppression for rightward heading, but a small release from inhibition for leftward heading.

In order to quantify this effect at the population level, we assigned the response of each neuron to leftward and rightward heading either to the congruent or incongruent condition, dependent on the neuron's tuning for heading. Importantly, each neuron contributes to both conditions: leftward tuned neurons contribute to the congruent condition during leftward self-motion and also to the incongruent condition during rightward self-motion (vice versa for rightward tuned neurons). Then, we tested for a significant difference of the perisaccadic response modulations in the congruent and incongruent condition by comparing the neural activity in a fixed, post saccadic window $t = 40 \text{ ms} - 160 \text{ ms}$ after saccade onset. In Figure 5, in each panel, saccade onset is indicated by the vertical, solid line. The time window for analysis [40 ms 160 ms] is indicated by the pair of vertical red dashed lines in each panel.

Figure 6 about here

This window was considered wide enough to capture both, suppressive and dis-inhibitory effects. We compared the neural activity in this time window with activity long after the saccade (300 ms to 450 ms after saccade onset) by computing the ratio of the two, defining a modulation index. Based on this definition, an index of 1.0 indicates no perisaccadic response modulation. An index smaller than 1.0 indicates response suppression, while an index larger than 1.0 indicates response enhancement. Our prediction from the hypothesis detailed above was an index below 1.0 for the congruent condition and an index above 1.0 for the incongruent condition. Figure 6 gives the result for the population of $n = 71$ neurons. Each data point represents data from a single neuron, with the x-value depicting the index in the congruent condition and the y-value depicting the index of the same neuron in the incongruent condition. As can be easily seen, the majority of data points lies above the diagonal. For the population of cells, the median of the index was significantly smaller than 1.0 ($0.812 = 81.2\%$, one-sided signed rank test, $p < 0.05$) in the congruent condition, and significantly larger than 1.0 in the incongruent condition ($1.206 = 120.6\%$, one-sided signed rank test, $p < 0.05$). The 95% confidence intervals as determined from bootstrapping were below 1.0 in the congruent condition and above 1.0 in the incongruent condition.

Figure 7 about here

While this result was fully in line with our hypothesis of selective suppression and enhancement or release from inhibition, the computation based on a fixed analysis window (40 ms – 160 ms after saccade onset) was a rather coarse measure. To overcome this limitation, we assigned the response of each neuron from our recorded sample of MST and VIP neurons to leftward and rightward heading stimuli to the congruent or incongruent condition and determined the full time-course of the population response modulation. The result is shown in Figure 7. Indeed, when neurons were stimulated with their preferred heading direction (Congruent condition, left panel, blue curve), saccadic suppression of activity was observed. In contrast, when the same neurons were stimulated with their non-preferred heading (Incongruent condition, right panel, purple curve), no saccadic suppression could be seen. Instead, a small but significant response enhancement occurred. In both panels, the solid lines indicate this time resolved response modulation, while the shaded, colored areas indicate the 95% confidence interval as determined from bootstrapping. Finally, as hypothesized, the peak of the enhancement (or release from inhibition) in the incongruent condition (120 ms after saccade onset) occurred 40 ms after the peak of suppression in the congruent condition (80 ms after saccade onset). In both panels, both events are indicated together with the time of saccade onset by vertical dashed lines.

Figure 8 about here

In a final step, we modelled the effect of saccadic suppression in a quantitative model. The advantage of this approach is that one can apply various forms of perisaccadic response modulation and determine, if and if so which form of suppression reproduces the effect of a compression of decoded heading. For this model, we created a set of $n=71$ “virtual neurons”, i.e. time courses of neural activity. The activity levels were modelled after our real data set, i.e. activity distribution across the population was identical to our real data set, with additional noise. We simulated presentation of five self-motion directions (-30° , -15° , 0° , 15° , and 30°). These artificial neurons were linearly tuned to self-motion direction, with the distributions of strengths of their tunings modelled after our real data set. Based on these overall settings, we generated 650 ms long time courses of stimulus driven activity. These time courses were used to train the heading decoder. In a final step, we convoluted the time courses with the response profiles deduced from global suppression or selective suppression, respectively (see Methods for details, and Figure 8). We repeated this procedure fifty times for each of the two forms of saccadic suppression, i.e. global and selective, and computed the average time courses of decoded heading. The results are shown in Figure 9. As can be easily seen, heading could in principal be decoded from the responses of these artificial neurons, as indicated by the solid lines (decoded heading) superimposing the dashed lines, which represent the real heading. Importantly, when applying global suppression (left panel), there was only a marginal modulation of decoded heading shortly after the saccade. Yet, this modulation was clearly different from the de-coding pattern obtained from our real neurons (Figure 2). On the contrary, selective suppression, i.e. suppression of the neurons in the congruent condition combined with a release from inhibition of the neurons in the incongruent condition (right panel), triggers a compression of decoded heading, with a maximum of suppression around 100 ms after saccade onset.”

Discussion: “Our in-depth analysis of the neurophysiological data not only allowed to predict a hitherto unknown visual illusion. It allowed also for determining its neural basis. Our quantita-

tive model suggested that compression results from selective suppression in one set of neurons and a simultaneous enhancement of activity in the complementary set of neurons. Importantly, these different response modulations are induced by the same saccades. It was exactly these activity patterns (selective suppression vs. simultaneous enhancement) which we observed in our neuronal sample.

While the temporal dynamics of the compression effects are rather similar across various affected measures (among them space, time, and number), our results cannot be explained by these compressions. Instead, they add a further dimension, namely that of self-motion. Space, time, numbers, and motion are all fundamental representations within primate parietal cortex. As deduced from our model, a common mechanism for these perisaccadic illusions may consist of selective modulation of activity in these representations by saccades.”

Methods: “We employed bootstrapping (Matlab function *bootci*, with 1000 repetitions) to determine the time-resolved 95% confidence intervals of neuronal response modulations (Figures 3 and 7) and of the precision of heading-decoding and -perception (Figure 13).”

For the quantitative model on the role of saccadic suppression on heading decoding we convoluted the time courses of the artificial neurons with the response profiles deduced from global suppression or selective suppression, respectively (Figure 8). To this end, we normalized the time courses of global or selective suppression, so that the values > 100 ms before and after the saccade were centered on 1.0. In the one case, i.e. global suppression, all time courses of neural activity were convoluted with this global suppression function. In the other case, we applied selective suppression, i.e. we convoluted neurons with the suppression function in case of a congruent stimulus condition (blue curve) and with the release-from-inhibition function in the incongruent stimulus condition (magenta curve).”

R-1, Q-2: 2) Interpretation: psychophysics. *Here the authors are on considerably stronger ground, since the data can literally be described as a central bias during saccades. However, once again there is an alternate and parsimonious hypothesis that needs to be excluded. In most perceptual measurements, when sensory evidence is weak, subjects rely on their priors. To my knowledge, no one has attempted to estimate priors for heading perception, but one would intuit based on first principles that they would be for straight-ahead motion, since this is how observers spend most of their time in motion.*

We fully agree with the reviewer that this is an important and highly relevant point. In fact, a previous study has indeed investigated priors for heading perception (Cuturi and MacNeilage, 2013). Surprisingly, also to the authors, the observed prior was not centered on straight-ahead. Instead, when testing heading in the full horizontal 2-D motion space, two peaks directed 90 degrees to the left and to the right, i.e. perpendicular with respect to straight-ahead, evolved. As stated by the authors: “Lateral biases are inconsistent with standard Bayesian accounts which predict that estimates should be biased toward the most common straight forward heading direction.”.

The authors suggest that this (unexpected) prior was the reason for

finding a peripheral bias in their subjects' heading estimates. Such centrifugal bias for visually simulated, but not vestibular, self-motion had also been reported by Crane (Crane, 2012). This author also showed that the centrifugal bias was more pronounced when heading could be in any direction within the full 2-D horizontal plane as compared to an experimental context, in which heading was restricted within a ninety-degree sector centered on straight-ahead. This finding was in line with results from previous studies (d'Avossa and Kersten, 1996; Telford and Howard, 1996).

Figure 14: Reference frame of the compression of perceived heading. Single subject level.

Cuturi and MacNeilage (2013) and Crane (2012) suggested that the observed centrifugal bias might result from a non-uniform distribution of heading preferences in primate areas MST and VIP, as described in the

macaque among others by Angelaki, DeAngelis and colleagues (MST: Gu et al., 2010; VIP: Chen et al., 2011). In their study on area MST, these authors have developed a model for decoding heading from discharges of monkey area MST and found a decoding bias towards the periphery. Indeed, long before and especially long after a saccade, also our decoding from areas MST and VIP revealed a (slight) centrifugal bias (see our Figure 2). Accordingly, during slow eye-movements, our data are in good agreement with previous results obtained during steady fixation.

Figure 15: Reference frame of the compression of perceived heading at the population level.

Nevertheless, we were keen to test the reviewer’s idea that in all cases, compression of heading representation should be towards the direction typically occurring during everyday life, i.e. towards straight-ahead. Hence, we have designed an additional psychophysical experiment. Here, like in the main experiment, subjects performed a 10° upward saccade, and perisaccadically, we presented a 40 ms long self-motion sequence. Different from the main experiment, however, either (i) the subjects’ head was directed $\alpha = 15^\circ$ degrees to the left, while gaze was still straight-ahead, or (ii) the head was straight-ahead, but eyes were directed $\alpha = 15^\circ$ to the left.

This experiment tests four possibilities for the center of compression. If compression is directed to the head’s midline (variant A), one would expect a shift of the center of compression in the first case (head to the left), but not in the second (eyes to the left). If compression is directed towards the body midline (variant B) or the screen center (variant C), one would expect that the center of compression is always at the center of the screen. If compression is directed towards the

center of gaze (variant D), one would expect compression towards the screen center when gaze was straight ahead and towards a position 15 degrees to the left when gaze was at that position. Clearly, we observed a perisaccadic shift of perceived heading towards the screen center in the first case (head to the left, panel A), but a shift towards -15° , when the eyes were directed to the left (panel D). The time course and strength of compression were identical in both cases (panels B and E). We determined also the time-course of the average perceived position, i.e. the time-course of the mean of all five heading percepts. Despite the strong perisaccadic response modulation, this average perceived heading was rather stable over time in the head-turned condition (Panel C). In the gaze-turned-head-straight-condition, however, also long before and after the saccade, perceived heading was biased towards the gaze direction. This bias got amplified in the temporal vicinity of the saccade (Panel F). At the time of maximum compression, all headings were perceived as being 15° to the left.

This finding of an eye-centered bias was consistent across the group of five subjects tested in this additional psychophysical experiment, as shown in the figure above, which is the new Figure 15. It is also in line with findings from (Crane, 2015). In this study, the author found that visual headings were biased towards eye but not head position. Our finding is in line with this previous result. In our new experiment, during steady fixation, the mean perceived heading was shifted in the direction of the eccentric eye position by approximately one-third of the gaze amplitude. This value is somewhat smaller than the one found in Crane's study (46%). Yet, like in Crane (2015), a peripheral head position with eyes straight-ahead did not induce a shift of the mean perceived heading. Crane (2015) concluded that the perception of visually simulated self-motion is organized in eye-centered coordinates, which would be in line with neurophysiological data from the animal model, i.e. the macaque monkey (area MST: Fetsch et al., 2007; area VIP: Chen et al., 2013). Our data are perfectly in line with this hypothesis, which, in turn, provides further evidence for our idea that the observed compression of perceived heading might be based on response properties of neurons in primate areas MST and VIP.

We report and discuss our findings as follows:

Abstract: "As predicted, perceived heading was perisaccadically compressed. A behavioral control experiment revealed compression to be directed towards the direction of gaze rather than the head- or body-midline."

Results: "The neurophysiological data had provided evidence for an eye-centered reference frame of the compression effect. This leads to the hypothesis that the center of the compression effect is the eye position."

sion should move with the eye but not the head, if both were changed independently from each other.

Figure 14 about here

We decided to test this hypothesis in a further control experiment using a heading task instead of a forward/backward discrimination. Like in the main experiment, subjects performed a 10° up-ward saccade, and perisaccadically, we presented a 40 ms long self-motion sequence. Different from the main experiment, however, either (i) the subjects' head was directed $\alpha = 15^\circ$ degrees to the left, while gaze was still straight-ahead, or (ii) the head was straight-ahead, but eyes were directed $\alpha = 15^\circ$ to the left. This experiment tests four possibilities for the center of compression. If compression is directed to the head's midline (variant A), one would expect a shift of the center of compression in the first case (head to the left), but not in the second (eyes to the left). If compression is directed towards the body midline (variant B) or the screen center (variant C) one would expect that the center of compression is always at the center of the screen. If compression is directed towards the center of gaze (variant D), one would expect compression towards the screen center when gaze was straight ahead and towards a position 15° to the left when gaze was at that position. Figure 14 shows the result for a single observer. Clearly, we observed a perisaccadic shift of perceived heading towards the screen center in the first condition (head to the left, panel A), but a shift towards -15° , when the eyes were directed to the left (panel D). Hence, compression occurred in eye-centered coordinates towards the direction of gaze. The time course and strength of compression were identical in both cases (panels B and E). We determined also the time-course of the average perceived position, i.e. the time-course of the mean of all five heading percepts. Despite the strong perisaccadic response modulation, this average perceived heading was rather stable over time in the head-turned condition (Panel C). In the gaze-turned-head-straight-condition, however, also long before and after the saccade, perceived heading was biased towards the gaze direction. This bias became amplified in the temporal vicinity of the saccade (Panel F). At the time of maximum compression, all headings were perceived as being 15° to the left. The shift of the center of compression towards 15° left in case of shifted gaze, but not towards the direction of the head, was consistent across the group of subjects, as illustrated also in Figure 15."

Discussion: "Our second behavioral experiment allowed us to determine the reference frame of perisaccadic compression of perceived heading. A previous study had found that, during fixation, perceived direction of visually simulated self-motion was biased towards eye but not head position (Crane, 2015). Our findings are in line with this previous result. During steady fixation, the mean perceived heading was shifted in the direction of the eccentric eye position by approximately one-third of the gaze amplitude. This value is somewhat smaller than the one from (Crane, 2015) (46%). Yet, like in (Crane, 2015), a peripheral head position with eyes straight-ahead did not induce a shift of the mean perceived heading. It was concluded that the perception of visually simulated self-motion is organized in eye-centered coordinates, consistent with neurophysiological data from the animal model, i.e. the macaque monkey (Area MST: Fetsch et al., 2007. Area VIP: Chen et al., 2013). Our data are perfectly in line with this hypothesis, which, in turn, provides further evidence for our idea that the observed compression of perceived heading might be based on response properties of neurons in primate areas MST and VIP."

R-1, Q-3: 2a) Control experiment. The crux of their argument lies in the control experiment, where subjects are discriminating forward from backward headings. This is potentially flawed in a couple of important ways, but the description is quite brief. The discrimination is quite different from the main experiment, and can be solved using local motion cues anywhere in the scene, since it boils down to a discrimination between opposite directions. So, potentially very different strategies could be used by the subjects, relying on totally different physiological mechanisms. It is also very easy, so the subjects are very near maximum performance. In this area of a psychometric function, many of the errors are probably “lapses”, rather than actual sensory mistakes. But even if they are sensory mistakes, the difference between 94% and 96% performance can be quite large in terms of d' or some similar measure. So, it is a very insensitive control experiment. Lastly, the analysis was unpaired, and a Wilcoxon test should probably have been used.

This point is well-taken. First of all, we fully agree with the reviewer that our discrimination control experiment (forward vs backward), differs from the perceptual task in the main experiment, i.e. judging heading direction. Nevertheless, we still think that this control was important to perform, because one could otherwise argue that subjects were literally blind for visual stimulation across saccades, due to saccadic suppression. If this had been the case, the performance in this discrimination task would have dropped to chance level, i.e. 50% correct. Yet, this was definitely not the case. Instead, discrimination performance across saccades was above 96%. In the original version of our manuscript, we had already shown that this performance was significantly larger than chance level. Following the reviewer's suggestion, we have replaced the previously employed signed rank test by a paired Wilcoxon signed rank test in order to compare performances during steady fixation and perisaccadically. Yet, also using this test, difference in performance was not statistically significant ($p > 0.5$).

In addition, we followed the reviewer's advice to consider effect sizes, i.e. d' . We computed d' as difference of the means of (i) discrimination performance as obtained >50 ms before or after saccade onset $\bar{x}_{pre \& post}$ and (ii) perisaccadic discrimination performance \bar{x}_{peri} (i.e. onset of the self-motion stimuli <50 ms before or after saccade onset), divided by the combined standard deviation of these two activity patterns $Std_{combined}$ (Cohen, 1988; Hartung et al., 2011).

$$d' = \frac{\bar{x}_{pre \& post} - \bar{x}_{peri}}{Std_{Combined}}, \text{ mit } Std_{combined} = \sqrt{\frac{(n_1-1)std_1^2 + (n_2-1)std_2^2}{n_1 + n_2 - 2}}$$

Here, n_1 and n_2 are the numbers of samples in the pre- and post-saccadic intervals (n_1) as well as in the perisaccadic interval (n_2). std_1 and std_2 are the standard deviations of these samples. d' turned out to be

0.21, i.e. comparably small. Accordingly, we conclude that performance dropped perisaccadically only marginally in the discrimination task and was way above chance level (one-sided signed rank, $p < 0.0001$). We acknowledge this further analysis in the Results and Methods sections, which now reads as follows:

Results: “During steady fixation (>50 ms prior to and after saccade onset), performance of subjects was very high (97.1%), but significantly smaller than perfect (i.e. 100%. one-sided signed rank test, $p < 0.0001$). If saccadic suppression would have strongly compromised self-motion perception, we would have expected a drop in performance close to chance level (50%) in the temporal vicinity of a saccade. Yet, discrimination performance <50 ms prior to and after saccade onset was almost as high as during fixation, i.e. 96.1%. While this value was significantly smaller than 100% (one-sided signed rank test, $p < 0.0001$), it was also significantly larger than chance level (one-sided signed rank test, $p < 0.001$). A paired, one sided Wilcoxon signed rank test found no difference in performance in the perisaccadic interval compared to fixation ($p > 0.5$). Effect size of the perisaccadic perceptual modulation, d' , was 0.21, i.e. comparably small. Accordingly, we conclude that perisaccadic optic flow perception *per se* was only marginally modulated.”

Methods: “In addition, we determined effect sizes of the perisaccadic response modulation, i.e. d' . We computed d' as difference of the means of (i) discrimination performance as obtained >50 ms before or after saccade onset $\bar{x}_{pre \& post}$ and (ii) perisaccadic discrimination performance \bar{x}_{peri} (i.e. onset of the self-motion stimuli <50 ms before or after saccade onset), divided by the combined standard deviation of these two activity patterns $Std_{combined}$ (Cohen, 1988; Hartung et al., 2011).

$$d' = \frac{\bar{x}_{pre \& post} - \bar{x}_{peri}}{Std_{combined}}, \text{ with } Std_{combined} = \sqrt{\frac{(n_1-1)std_1^2 + (n_2-1)std_2^2}{n_1 + n_2 - 2}}$$

Here, n_1 and n_2 are the numbers of samples in the pre- and post-saccadic intervals (n_1) as well as in the perisaccadic interval (n_2). std_1 and std_2 are the standard deviations of these samples.

In a second control experiment, we aimed to determine the reference frame of the compression of perceived heading. Here, like in the main experiment, subjects performed a 10° upward saccade, and perisaccadically, we presented a 40 ms long self-motion sequence. Different from the main experiment, however, either (i) the subjects' head was directed $\alpha = 15^\circ$ degrees to the left, while gaze was still straight-ahead, or (ii) the head was straight-ahead, but eyes were directed $\alpha = 15^\circ$ to the left. If compression had been represented relative to the head, we would have expected a shift of the center of compression in the first case (head to the left), but not in the second (eyes to the left).”

R-1, Q-4: 2b) Analysis in Supplementary Figure 1. This is probably the best argument against the saccadic suppression hypothesis, but I do not think it particularly strong either. It is possible that a proper Bayesian analysis of this data and the data in Figure 4 would be compelling; the argument depends on the variance of the prior distribution. It can be estimated from the main experiment data, and if it is large, then this case is pretty good. But I would need to see the analysis to be sure. It would need to be

performed for each subject, and all the individual subject data presented. The subject in Figure 4 appeared to both have a central and a right bias, which would be estimable with a full analysis.

Figure 11: Perisaccadic perception of heading.

As already mentioned above (see our response to R-1, Q-2), two previous studies by Cuturi and MacNeilage (2013) and Crane (2012) have provided the (unexpected) result that the prior for perceived heading peaks at around 90 degree off from straight-ahead. Furthermore, given our in-depth analysis of the neurophysiological data concerning saccadic suppression, we have concluded that global saccadic suppression cannot account for our results.

Following the reviewer's suggestion, we now show data from all eight subjects in the heading perception task (see figure above, which is now the new Figure 11 in the revised manuscript). The effect of perisaccadic compression of perceived heading was an extremely robust finding, clearly visible in each and every subject. Also, the timing of the peak of compression was very consistent across subjects, in all cases within a few ms before saccade onset. All eight subjects showed an overall centripetal shift of perceived headings also during fixation, as indicated by the colored dashed horizontal lines in each panel. In two subjects (VP2 and VP8), this global centripetal shift was rather strong, but also these two subjects revealed a clear compression of perceived heading. In saccade trials, the same, subject-dependent centripetal shift was observed also long before and long after the saccade.

Such a centripetal shift is different from the centrifugal shift reported by Cuturi and MacNeilage (2013). We can only speculate about the reason of this difference in results. It is most likely that differences are due to the fact that the experimental conditions in the two studies were quite different. In the study by Cuturi and MacNeilage (2013), stimuli lasted well beyond one second, while in our experiment, stimuli lasted only 40 ms. Furthermore, Cuturi and MacNeilage (2013) employed simulated self-motion through a 3-D cloud of random dots, while our random dot stimuli simulated self-motion across a 2-D ground plane. As shown e.g. by (Lappe et al., 1999), the exact spatial layout of a scene can have a strong influences on perceived heading.

Suppl. Figure 1: Precision of heading perception at the single subject level.

In addition to the raw data from all subjects and following the reviewer's suggestion we have performed our analysis concerning the precision of the behavioral performance at the single subject level (see figure above, which is now the Supplementary Figure 1). As can be easily seen, the overall values of the precision were different across subjects. But

in none of the subjects, there was a systematic variation around the time of the saccade.

In addition, we now computed also the precision of heading as decoded from neural activity (see figure below, which is the new Figure 13 of the revised manuscript). Given the rather long temporal intervals of our computation (20 ms bins), we determined the precision of decoded heading by computing the standard deviation of three consecutive decoded values (for comparison: for the behavioral data, we employed nine consecutive samples, with an average inter-sample spacing of 2.5 ms, see below for details). As can be easily seen, also the precision of the decoded heading did not reveal a perisaccadic modulation (see above our response to **R-1,Q-1**).

Figure 13: Precision of heading decoding (left panel) and perception (right panel) relative to saccade onset.

We refer to these different points now across the manuscript as follows:

Results:

“Data from all eight subjects are shown in Figure 11. In each panel, colors represent data for the five different self-motion directions.”

and

“For each subject, perceived heading during fixation trials was slightly biased towards fixation, as indicated by the dashed, colored horizontal lines. For subjects 2 (VP02) and eight (VP08), this bias towards fixation was strongest. Across all subjects, in saccade trials, ...”

and

“Qualitatively, this effect was very consistent across subjects. Like for the physiological data, we determined the time-course of compression by computing the normalized standard deviation of the perceived headings. This is exemplified for data from subject 1 (VP01) in panels A and B of Figure 12.”

and

“Previous studies on perisaccadic modulation of visual perception showed that response precision of behavioral responses did not decrease perisaccadically (e.g. Morrone et al., 2005). Accordingly, we were also interested in the precision of the decoding of headings as well as the

behavioral responses of our subjects. We computed precision as the standard deviation of three (for the decoding) or nine (for the subjects' responses) consecutive samples. The result is shown in Figure 13 (Behavioral data from each subject are shown in Supplementary Figure 1). Data curves show the mean precision and the 95% confidence interval as determined from bootstrapping (See Methods for details). As can be easily seen, precision did not modulate perisaccadically, neither for the decoded heading (panel A) nor for the behavioral responses (panel B). This finding is in line with the above mentioned behavioral studies.”.

R-1, Q-5: 3) Physiology sample. Finding only 40 neurons from 119 that meet the very lax criterion for tuning that they used is peculiar. But there is no breakdown of the sample or the principal results by monkey or area. This needs to be provided in a supplementary table at the very least. They probably don't have the power to test whether the results are different in VIP or MST, but this is a very interesting question. Assuming the two areas are the same is not a bad first guess, but by no means certain. If at all possible, they should increase this sample size; it would make readers much more confident in the results. I recognize that this is a re-analysis of archived data, but I would hope they could boost the N in some way, either by relaxing the inclusion criteria or even by recording new data.

We see the reviewer's point that a sample of $n=40$ neurons is rather small. Yet, different from the reviewer's opinion, we consider the criterion for including neurons in our sample to be very strict. In the original version of our manuscript we considered only those neurons for decoding, whose activity in a very short (only 150 ms wide) time window starting 300 ms after saccade onset was significantly different for the three self-motion directions (as tested by an ANOVA on ranks, 2 df, $p<0.05$). This caused the proportion of neurons ($40/119=33\%$) to be small as compared to published data on neurons in areas MST and VIP being significantly tuned to optic flow stimuli during fixation (see below). As also pointed out by the reviewer, data analysis was performed on a data set which was basis for two other previous publications focusing on the invariance of heading responses during real or simulated eye-movements. In these two studies, we have reported that $61/84$ (72%) of the neurons from area MST and $48/68$ (71%) of the neurons from area VIP revealed a statistically significant optic flow response with respect to baseline. The statistical test (ANOVA on ranks) was performed on rather long response windows (> 1000 ms) with the monkeys fixating throughout the trial (typically 2500 ms).

When comparing values of the proportion of cells with significant responses across studies, it is important to consider the exact measure which has been employed: in some studies, values indicate the proportion of cells being responsive with respect to baseline, i.e. spontaneous activity. But sometimes the values reflect the proportion of cells with a significant tuning, i.e. revealing a significant difference between the responses for different self-motion directions. As an example, Duffy and Wurtz (1991) reported that 86% of the cells were respon-

sive for simulated translational self-motion in 3-D, i.e. with respect to baseline. Yet, as a result from another study (Duffy, 1998), only half of the cells turned out to significantly tuned for visually simulated self-motion in the horizontal plane (as studied in our case). In our own study (Bremmer et al., 2002), we have found that roughly three quarters of the neurons in area VIP (74.4%) respond significantly to visually simulated translational motion in 3-D space, i.e. with respect to baseline. Chen and colleagues reported, that about 50% of the neurons in VIP are significantly tuned for visually simulated self-motion in 3-D space (Chen et al., 2013). Given that in our current study we only presented visually simulated self-motion in the horizontal plane, the expected proportion of tuned VIP neurons was even less than 50%.

	Area MST number / %	Area VIP number / %
Monkey R	5 of 9 / 56%	
Monkey H	44 of 55 / 80%	10 of 36 / 28%
Monkey C		12 of 19 / 63%

Table 1: Number and proportion of neurons contributing to the decoding.

For the revised version of our manuscript, we have followed the reviewer's advice and defined a new approach for testing for significant optic flow responses. As mentioned also in the original manuscript, neural activity and eye-movements were recorded during presentation of long self-motion sequences. Each trial lasted between 15s and 20s. For our new analysis, we split each trial in 1000 ms slices and computed activity for each temporal slice, regardless of whether or not saccades occurred in this slice. In order to determine whether or not neurons were significantly tuned, we employed an ANOVA on ranks on the response levels as determined from the temporal slices for the three different self-motion directions. We find that 71/119 (60%) of the neurons have a significant tuning (ANOVA on ranks, 2 df, $p < 0.05$). From these 71 neurons, $49/64 = 76\%$ are from area MST, while $22/55 = 40\%$ are from area VIP. The table above, which is also the new Table 1 of the revised manuscript, indicates, how these samples were derived from the individual animals. Importantly, the proportion of cells was fully in line with data from the literature as detailed above. In our sample, the proportions of significantly tuned neurons in area VIP were in both cases a bit smaller than those in area MST. This was to be expected, again, as detailed above. The proportion of cells for a given area varied across the animals. This observation might simply be due to the comparably small number of neurons per animal per cortical area. For small sam-

ples, small changes in absolute number induce large changes in proportion. We have revised our Results and Methods sections accordingly:

Results: “We recorded activity from 119 neurons in the medial superior temporal area (area MST) and the ventral intraparietal area (area VIP) of three macaque monkeys. 64 neurons were recorded from area MST (55 neurons from monkey H and 9 neurons from monkey R), and 55 neurons from area VIP (36 neurons from monkey H and 19 neurons from monkey C). During the recording monkeys freely viewed long sequences of optic flow stimuli simulating self-motion across a ground plane in one of three directions (+/- 30 degrees and straight ahead. Figure 1A).”

and

“We applied the decoder to the continuously recorded discharge of each neuron that showed a statistically significant tuning for heading (ANOVA on ranks, 2 df, $p < 0.05$, see Methods for details), i.e. 71/119 = 60% of the neurons. Table 1 details the number and proportion of neurons from each monkey and each area contributing to the decoding.

Table 1 about here

	Area MST. Number / Proportion	Area VIP. Number / Proportion
Monkey R	5 of 9 / 56%	
Monkey H	44 of 55 / 80%	10 of 36 / 28%
Monkey C		12 of 19 / 63%

Table 1: Number and proportion of neurons contributing to the decoding approach.

The decoding approach provided us with a time-resolved estimate of self-motion direction based on the discharges of a population of cells ($n=71$).”

Methods: “We recorded activity from 119 neurons in areas MST ($n_1 = 64$. 55 neurons from monkey H and 9 neurons from monkey R) and VIP ($n_2 = 55$. 36 neurons from monkey H and 19 neurons from monkey C) while the monkeys freely viewed long self-motion sequences, lasting between 15s and 20s. In order to identify those neurons that are significantly tuned to heading, we split each trial in 1000 ms slices and computed the neuron’s activity for each temporal slice. Then, we employed an ANOVA on ranks with two degrees of freedom (2 df) on the responses in these temporal slices for the three different self-motion directions.”

R-1, Q-6: 4) Saccade direction effects. *It doesn’t make any sense to this reviewer that saccades would compress space isotropically. This is certainly not the case for position reports of flashed stimuli. However, because all saccades were treated the same, it is impossible to know if there was any direction dependence of the effects on the physiology. This is an important question because the saccades in the physiology experiments and the perceptual experiments were very different. Ideally, one would like to see a subdivided analysis between vertical saccades (which were used in the psychophysics), and those with left or right horizontal components. But if the results really are isotropic, then it supports the saccadic suppression hypothesis.*

This point is well-taken. Yet, first, we would like to emphasize that the compression of the representation of heading, as found in our present study, is different from a compression of space. While our findings are in line with previous findings concerning the compression of perceptual space, time, and number (for review see Burr et al., 2010), conceptually motion is different from space. As an example, a certain direction of visual motion can occur at any point in space. Hence, what we investigated in our study was not another form of compression of space, but the compression of a representation of heading. We had already referred to this issue in the first version of our manuscript and elaborate on it in the revised version (see below).

Secondly, for position reports of flashed stimuli the compression is typically directed towards the goal of the saccade. Our second control experiment shows a similar focus towards the direction of gaze. For the physiological data, we took only those eye movements in which gaze was within the central 20 deg. Therefore, we believe that the compression effects for those eye movements should be similar enough to be combined in one analysis, even if the saccade directions are diverse, as long as the end positions are homogenous.

Thirdly, given our new results on selective perisaccadic modulation (congruent vs. incongruent visual stimulation), an influence of saccade direction on the neural responses, if existent, must be rather marginal as compared to the influences of the motion responses. As shown above (mainly our response to **R-1,Q-1**), the exact form of a neuron's perisaccadic response modulation was dependent on its directional preference for visual motion: a given saccade induced suppression for one group of neurons (congruent stimulation) and, at the same time, a release-from-inhibition for another group of neurons (incongruent stimulation).

Fourth and finally, for the psychophysical experiment, we chose vertical saccades for two reasons. First, the vertical component of the saccades made is similar for all three FOE positions, as in all cases the stimulus was in the lower half of the screen and resetting saccades hence were upward. Second, for the vertical saccade the saccade direction was orthogonal to the placement of the FOE in the different conditions and to the judgment that the participant had to submit, i.e. heading in the horizontal plane. We think that this choice gives the least interference between eye movement direction and heading task.

We elaborate on this issue now in the Discussion section of the revised manuscript:

Discussion: "Our in-depth analysis of the neurophysiological data not only allowed to predict a hitherto unknown visual illusion. It allowed also for determining its neural basis. Our quantitative model suggests that compression of the representation of heading results from selective

suppression in one set of neurons and a simultaneous enhancement of activity in the complementary set of neurons. Importantly, these different response modulations are induced by the same saccades. It was exactly these activity patterns (selective suppression vs. simultaneous enhancement) which we observed in our neuronal sample.”

Lesser issues:

R-1, Q-7: 5) *The timecourse data need to be presented on the same axes, relative to the start of the saccade, without adjustment for latency.*

Figure 12: Time course of compression of perceived and decoded heading. We followed the reviewer’s suggestion and have replotted the data in the new Figure 12 (see figure above, and also our response to **R-2,Q-2**). We indicate the offset τ between the peak of compression of decoded heading (Monkey Physiology) and the representation of heading in humans (Human Behavior). This offset is 116 ms, which is in very good agreement with the latency of visual responses in primate visual cortex (e.g. Schmolesky et al., 1998). We have adjusted the description of the figure as follows:

Results: “Quantitatively, the temporal dynamics of perceived heading were also very consistent across all eight subjects: peak compression occurred between -10 ms and -1 ms relative to saccade onset, with a mean value of $t = -8.25$ ms, as shown in Figure 12C. This figure shows the time courses of the compression of perceived heading for each subject as well as the mean.

Given the similarity of perisaccadic heading compression as determined from our neurophysiological and psychophysical data, we employed temporal cross correlation to determine the best overlap between the two time-courses. A shift of $\tau = 116$ ms of the neurophysiological with respect to the psychophysical data resulted in the largest correlation coefficient (Figure 12D). At first glance, it might appear surprising that the behavioral data lead the neurophysiological by 116 ms. Yet, this has to be expected because the value of τ corresponds to a visual response latency and is perfectly in line with values in higher visual cortical areas of the macaque (Schmolesky et al., 1998). ”

R-1, Q-8: 6) No mention is made of the system used to measure eye movements in the human subjects; this needs to be included in the methods. Most video systems include a fixed or variable latency between movement and report. These specs should be included, because they are important to the time-series analysis that is central to this paper.

In both, monkeys and humans, eye movements were recorded at 500 Hz, as indicated in the original manuscript (monkeys: line 233; humans: line 294). In case of the monkeys, we used an eye coil system (Scalar Medical, Delft). For the human recordings, we used a video-eye tracker (EyeLink II, SR Research, Ontario, Canada).

R-1, Q-9: 7) Title. The title makes little sense as written, since heading is a behavior, and the subjects were not heading anywhere. I suggest “heading representations” instead.

According to the reviewer's suggestion, we changed the title to:

"Heading representations in primates are compressed by saccades"

Reviewer #2 (Remarks to the Author):

The current paper reports complimentary human psychophysical and macaque neurophysiology experiments showing that heading perception and neural response, respectively, are compressed toward straight ahead around the time of a saccade. The great strengths of the paper are the following: 1) the reported heading compression is a novel and significant finding, 2) this phenomenon is observed in both human psychophysics and monkey neurophysiology.

Reviewer-2, Question-1: Presentation and/or discussion could be improved regarding the following points:

1) can this compression be explained as a by-product of previously reported saccadic suppression/modulation of motion processing/perception?

We thank the reviewer for making this point, which is almost identical to Question 1, asked by Reviewer 1 (R-1,Q-1), and Question 2, asked by Reviewer 3 (R-3,Q-2). For the sake of brevity, we would like to ask Reviewer 2 to consider our very detailed response to R-1,Q-1.

The summary of our new and comprehensive analyses and our response was as follows: we concluded that (i) from a theoretical point of view, a compression of heading cannot result from globally acting saccadic suppression. Instead, (ii) compression results from selective suppression in one set of neurons and enhancement of activity in the complementary set of neurons. It was exactly these activity patterns which we observed in our sample of neurons.

R-2, Q-2: 2) The temporal offset between perception and neurophysiological responses needs to be discussed in greater detail; in particular, the claim that MSTd and VIP activity is the neural substrate of the observed perceptual compression would seem to be undermined by the fact that the perceptual compression PRECEDES the observed compression of neural tuning.

This point is well taken. When recording neural responses to visual stimuli, these responses come naturally with a latency. As summarized e.g. by Schmolesky and colleagues, onset response latencies vary from cell to cell and tend to increase from lower towards higher visual cortical areas (Schmolesky et al., 1998). Accordingly, a visual event occurring at $t = 0$ ms induces a response in extrastriate or parietal areas starting roughly 100 ms later. The response peak typically occurs another 10 ms to 30 ms later. In psychophysics, when human subjects are asked to respond behaviorally to a stimulus, a time axis in a data plot does not indicate the time of the subjects' percept, but the time of the stimulus. The percept is typically formed hundreds of milliseconds later. Nevertheless, we assign this behavioral response to the stimulus occurring e.g. at $t=0$ ms. Accordingly, when comparing the time courses

of neural responses with the behavioral responses of humans concerning visual events, one has to account for the neural response latency. We have taken an analogue approach in a previous study, where we aimed to determine a neural correlate of saccadic suppression (Bremmer et al., 2009). We found that time courses for visual responses to flashed stimuli in areas MST and VIP of the macaque were extremely similar to the time courses found for saccadic suppression as measured in perceptual experiments in humans. Yet, while saccadic suppression as measured behaviorally typically is maximum at saccade onset, the peak of neural saccadic suppression was found roughly 100 ms later (see also our response to **R-1,Q-1**). Accordingly, and as detailed in our response to **R-1,Q-1** and **R-2,Q-1**, we consider the complex time course of the responses in primate areas MST and VIP (selective suppression) to be the neural basis for the compression of heading representation.

We elaborate on this point now in the Results and Discussion section as follows:

Results: "Quantitatively, the temporal dynamics of perceived heading were also very consistent across all eight subjects: peak compression occurred between -10 ms and -1ms relative to saccade onset, with a mean value of $t = -8.25\text{ms}$, as shown in Figure 12C. This figure shows the time courses of the compression of perceived heading for each subject as well as the mean. Given the similarity of perisaccadic heading compression as determined from our neurophysiological and psychophysical data, we employed temporal cross correlation to determine the best overlap between the two time-courses. A shift of $\tau = -116\text{ ms}$ of the neurophysiological with respect to the psychophysical data resulted in the largest correlation coefficient (Figure 12D). At first glance, it might appear surprising that the behavioral data lead the neurophysiological by 116ms. Yet, this has to be expected because the value of τ corresponds to a visual response latency and is perfectly in line with values in higher visual cortical areas of the macaque (Schmolesky et al., 1998)."

Discussion: "Our data analyses revealed that the neurophysiological data followed the behavioral data by $\tau = 116\text{ ms}$. This value of τ is consistent with visual response latencies as summarized e.g. by Schmolesky and colleagues (Schmolesky et al., 1998). Visual onset response latencies in the macaque monkey vary from cell to cell and tend to increase from lower towards higher visual cortical areas (Schmolesky et al., 1998). Accordingly, a visual event occurring e.g. at $t = 0\text{ ms}$ induces a response in extrastriate or parietal areas starting roughly 100 ms later. The response peak typically occurs even another 10 ms to 30 ms later. In psychophysics, when human subjects are asked to respond behaviorally to a stimulus, the time axis in a data plot does not indicate the time of the subjects' percept, but the time of the stimulus. The percept is typically formed hundreds of milliseconds later. Nevertheless, we assign this behavioral response to the stimulus occurring e.g. at $t = 0\text{ ms}$. Accordingly, when comparing the time courses of neural responses with the behavioral responses of humans in response to external events, one has to account for the neural response latency. We have taken an analogue approach in a previous

study, where we aimed to determine a neural correlate of saccadic suppression (Bremmer et al., 2009). We found that time courses for visual responses to flashed stimuli in areas MST and VIP of the macaque were similar to the time courses found for saccadic suppression as measured in perceptual experiments in humans. Yet, while the maximum saccadic suppression as measured behaviorally typically is found at saccade onset, the peak of neural saccadic suppression was found roughly 100 ms later, i.e. with the expected latency. Accordingly, we consider the complex time course of the perisaccadic responses in primate areas MST and VIP (selective suppression) to be the neural basis for the perisaccadic compression of perceived heading.”

Minor comments:

R-2, Q-3: 32: veridically instead of veridical

Done.

R-2, Q-4: 103: Please explain in greater detail how statistical significance of tuning was determined. What are the samples that are passed to the ANOVA on ranks? Was it mean firing rate during the 300-450 interval on a given trial? How many trials were collected per neuron?

The reviewer is right: in the original version of our manuscript, the ANOVA was based on the very short post-saccadic interval 300–450 ms. The response intervals, which were taken from long presentation trials (lasting between 15s and 20s) and which we considered for our analysis, had to be in accordance with our selection criterion: there was no additional saccade from 200 ms before until 450 ms after a chosen saccade. Typically, we recorded three long trials (i.e. approximately 60 s in total) per self-motion direction. Given our strict exclusion criterion of only one saccade within a 650 ms response window, this resulted in a variable number of trials per self-motion direction per neuron, ranging across all recordings from 7 to 62, with a median of 27.

Based on the concerns raised by the reviewer below (**R-2-Q-5**) and also as recommended by Reviewer 1, we have changed our criterion for including neurons in our sample. We now have split the long trials into snippets of 1000 ms, regardless of saccades. We then determined for all snippets for a given self-motion direction the mean neural activity, treated each activity value of a 1000 ms snippet as a sample and computed for these samples an ANOVA across the three self-motion directions (2 df). This procedure increased the number of neurons, on which we applied our decoding approach, from 40 to 71. The proportion of neurons with a significant tuning is in the range as predicted from the literature (see also our response to **R-1, Q-5**). We explain our selection criterion in the Methods section as follows:

Methods: “We recorded activity from 119 neurons in areas MST ($n_1 = 64$. 55 neurons from monkey H and 9 neurons from monkey R) and VIP ($n_2 = 55$. 36 neurons from monkey H and 19

neurons from monkey C) while the monkeys freely viewed long self-motion sequences, lasting between 15s and 20s. In order to identify those neurons being significantly tuned to heading, we split each trial in 1000 ms slices and computed the neuron's activity for each temporal slice. Then, we employed an ANOVA on ranks with two degrees of freedom (2 df) on the responses in these temporal slices for the three different self-motion directions.”

R-2, Q-5: Only 40 of 120 neurons showing significant tuning seems rather a low proportion for these areas (1/3). How does this compare with other reports in the literature that tested for significance of tuning?

This is an important point, which has also been raised by Reviewer-1. As detailed above (see our answers given to **R-2,Q-4** and **R-1,Q-5**), this comparably small proportion of cells was due to our very strict selection criterion. Following Reviewer-1's suggestion, we have adjusted our criterion, which now provides us with 71 neurons for our decoding approach. The proportion of $71/199 = 60\%$ is in full agreement with data from the literature (e.g. Duffy et al., 1998; Chen et al., 2013). For the sake of brevity, we refer to the details as given in the responses to **R-2,Q-4** and **R-1,Q-5**.

R-2, Q-6: 126: Please make absolutely clear in the figure and/or figure caption and in text that relative compression is the same as normalized standard deviation. When different terms are used in text and axis labels, it gets confusing.

Thanks for making this point. Throughout the text in the revised manuscript and also in the axis-labels of all relevant figures, we refer now to “Compression (Normalized standard deviation)”.

R-2, Q-7: 155-156: Clarify language: onset times were uniformly distributed (were they?) over the -200 to +200 ms allowing a sampling of heading perception with 2 ms temporal resolution.

Onset times were uniformly distributed relative to the step of the saccade target. Prior to each experiment, we had determined the average saccade latency of each subject. We corrected for average latency, which resulted in an almost uniform distribution of stimulus onset times over the -200 to +200 ms with respect to saccade onset allowing for a sampling of heading perception with approximately 2 ms temporal resolution. The average temporal spacing between two samples across all subjects was 2.5 ms. We clarify this now in the text:

Results: “We recorded about 200 trials for each self-motion direction from each subject. Onset times of the self-motion stimuli were uniformly distributed relative to the step of the saccade target. Prior to each experiment, we had determined the average saccade latency of each subject. We corrected for this latency subject-wise, which resulted in an almost uniform distribution of stimulus onset times over the -200 to +200 ms with respect to saccade onset. This allowed for a sampling of heading perception with approximately 2 ms temporal resolution (aver-

age temporal spacing between two samples across all subjects was 2.5 ms.”

R-2, Q-8: 177: *Please clarify in the text the direction of the shift. This offset between perceptual and physiological responses is puzzling and deserves to be discussed in greater detail. If I understand correctly, maximum compression for perception occurs around the time of the saccade, while max compression for neural responses is 90 ms later. If the claim is that MSTd and VIP are the neural substrate for the perceptual response, why does the perceptual response precede the neural one?*

We have elaborated on this issue already in our response to **R-2, Q-2**. In brief: a visual stimulus occurring at time $t = 0$ ms induces a neural response in extrastriate and parietals areas roughly 100 ms later. When correlating behavior with neural activity, however, one has to correct for this response latency. Following the reviewer’s advice and also in our answer to **R-1, Q-7**, we elaborate on this issue now in the Discussion section, as detailed in our response to **R-2, Q-2**.

R-2, Q-9: 181: *This impairment of motion perception and its possible relation to the current results also deserves to be discussed in greater detail. It seems possible that a modulation of lower-level motion processing, something short of complete motion blindness, could be responsible for the heading compression observed here. The results of the control experiment do not rule this out.*

This point is well taken. It corresponds also to a major concern of Reviewer 1, **R-1, Q-2**. Following both Reviewers’ advice, we have performed an additional psychophysical experiment. Here, subjects had either to rotate the head or their gaze towards $\alpha = -15^\circ$, while gaze (in case of a turned head) or head (in case of shifted gaze) was straight ahead. We found a compression towards -15° in case of shifted gaze, but not head. This finding of an eye-centered bias was consistent across the group of five subjects. It is also in line with findings from (Crane, 2015). In this study, the author found that visual headings were biased towards eye but not head position. Crane (2015) concluded that the perception of visually simulated self-motion is organized in eye-centered coordinates, which would be in line with neurophysiological data from the animal model, i.e. the macaque monkey (area MST: Fetsch et al., 2007; area VIP: Chen et al., 2013). Our data are perfectly in line with this hypothesis, which, in turn, provides further evidence for our idea that the observed compression of perceived heading might be based on response properties of neurons in primate areas MST and VIP. For the sake of brevity, we would like to ask the Reviewer to check for our very detailed response to **R-1, Q-1**, including two new data figures.

R-2, Q-10: 201: *“evolve most likely not on purpose” please rephrase to improve English*

Done.

R-2, Q-11: 222: *Indeed, this effect looks like regression to the mean. This can happen at a neural level, i.e. around the time of a saccade neural responses tend toward the baseline firing rate. This can also happen at the decision level, i.e. regression toward the mean psychophysical response, namely the middle of the response line, since presented heading were symmetric around 0 (-30, -15, 0 +15, +30). Can your data differentiate between these alternative interpretations?*

This point is well taken and overlaps with the point raised by Reviewer-1, Question-2 (R-1,Q-2). As detailed above, a shift of gaze direction leads to a shift of the center of compression, regardless of the spatial layout of the self-motion directions. Accordingly, we exclude a regression to the mean as basis for the observed perceptual illusion. Instead, we consider eye-centered responses in primate areas MST and VIP (Fetsch et al., 2007; Chen et al., 2013) as neural correlate of this behavioral effect. For the sake of brevity, we would like to refer the reviewer to our detailed response to R-1,Q-2.

R-2, Q-12: 260-61: *So intercept is mean activity of the neuron, and compression is regression to the mean.*

As mentioned above (our response to R-1,Q-1), a global peri-saccadic suppression of neural activity cannot account for the observed compression of decoded heading. As another strong indication and as discussed above (our response to R-2,Q-11), a shift of gaze direction leads to a shift of the center of compression, regardless of the spatial layout of the self-motion directions. In summary, we consider regression-to-the-mean not appropriate to account for the observed results.

R-2, Q-13: 275: *why mean values over the before/after interval that are not symmetric relative to time of saccade. How was this choice made?*

In our analysis, we followed the approach of Lappe and colleagues (2000). These authors have investigated the perceptual compression of visual space across saccades. They have introduced a quantitative measure of compression: they computed the standard deviation of perceptual measures, in their case the perceived spatial location of a flashed, visual stimulus: They "... normalized to the[ir] respective average values 100 ms before and after the saccade." (Quote from Lappe et al., 2000). In our current study, saccade duration was variable. To make sure to not consider perceived heading directions which were under the influence of the saccade, we chose for the 100 ms wide windows prior to (-200 ms to -100 ms before saccade onset) and after the saccade (200 ms – 300 ms). We explain this choice now in more detail:

Methods: "From the population of cells with a statistically significant tuning for heading we determined decoded heading as median of the heading values provided by each neuron. In steps of 20 ms, we employed an ANOVA on ranks (2 df, $p < 0.05$) together with a false discovery rate

(FDR) correction to determine decoded headings which were significantly different from each other. A moving average of heading was determined as the average value of two consecutive values assigned to the time-point in the middle between the two samples. Following a procedure introduced in a previous study investigating the perisaccadic compression of perceptual space (Lappe et al., 2000), we determined normalized compression as the standard deviation of the time-courses of the three decoded headings, divided by its mean value in two 100 ms wide windows. To make sure to not consider perceived heading directions which were under the influence of the saccade, we chose for the 100 ms wide windows prior to (-200 ms to -100 ms before saccade onset) and after the saccade (200 ms – 300 ms). The time of the maximum compression was determined as the time for which the minimum normalized standard deviation was observed.”

R-2, Q-14: 309: please specify/repeat how the time course was reconstructed.

For each subject, we obtained in pseudorandomized order approximately 200 heading responses per self-motion direction. Self-motion onsets were uniformly distributed with respect to stimulus onset, corrected for the average latency of each subject (see also our response to **R-2, Q-7**). Across subjects, the average temporal distance between two samples for a given heading direction was 2.5 ms. In order to be able to compute a time-resolved estimate of the compression, we computed the running mean, with a constant number of samples (9). Values were assigned to the central time-point of such a group of samples. In a final step, we interpolated between the samples to achieve a temporally continuous running mean. We explain this now in more detail in the Methods section, as detailed in our response to **R-2, Q-7**.

R-2, Q-15: 317: please specify direction of temporal shift

The temporal shift was the result of the temporal cross-correlation between the behavioral and neurophysiological data. We had to shift the neurophysiological data curve “to the left”, i.e. towards negative values, to achieve the best overlap with the behavioral data. This can easily be explained by an over-simplified example. Let’s assume, a visual flash was presented at $t = 0$ ms. A group of neurons in visual cortex will show a burst of activity at a latency of 100 ms. The visual percept of an observer will be based on this burst of activity. To align both event traces, i.e. light-on and neural activity, the activity trace has to be shifted by 100 ms “to the left”, so that both, the external event and the burst occur at $t = 0$ ms. Further details are given in our response to **R-2, Q-8**.

R-2, Q-16: 320: please explain this impairment in more detail and speculate how it could or could not be responsible for the observed effects

This concern is in line with **R-1,Q-1**, for which we have given a very comprehensive response, and which we would like to refer to as our response. In addition, we have elaborated a bit on this statement, which now reads as:

Methods: “Motion perception across saccades is impaired (saccadic suppression) (Ilg and Hoffmann, 1993; Frost and Niemeier, 2015). Nevertheless, even brief periods of visual motion can be perceived perisaccadically (Castet and Masson, 2000). Accordingly, in a first control experiment, ... ”

R-2, Q-17: 331: *Is it fair to say that a more appropriate control would have been to ask subjects to discriminate between headings of +30 and -30? Again, I do not see how the results of the control rule out the possibility that this is a corollary effect downstream from saccadic modulation of low-level motion processing.*

In our main experiment, we have asked subjects to determine heading directions. The subjects were not aware, that headings could take one of five values, -30, -15, 0, 15, and 30 degrees. As indicated in Figure 11 of the revised manuscript, observers were clearly able to dissociate the heading directions long before and after a saccade, but not during the saccade. We consider this a first part of our answer to the point raised here.

Secondly, and as also pointed out in our responses to **R-1,Q-3** and **R-3,Q-2**, our control experiment was intended in the first place to rule out that, perisaccadically, subjects were literally blind for the type of stimulus used in our study. This was clearly not the case, as indicated by the high proportion of correct perisaccadic trials in the control experiment.

Instead, and as detailed in our response to **R-1,Q-1**, it is the balanced interaction of selective suppression and response enhancement (most likely a release from inhibition) which induced the compression of decoded heading and the compression of perceived heading.

We elaborate on the control experiment in the Results as follows:

Results: “Motion perception across saccades is impaired due to saccadic suppression (Ilg and Hoffmann, 1993; Castet and Masson, 2000; Frost and Niemeier, 2015). Therefore, we aimed to determine if this reduction of the perisaccadic perception of motion could have had a critical influence on our behavioral data. In a control experiment, we asked five of the eight subjects to discriminate between self-motion stimuli simulating forward and backward self-motion. During steady fixation (>50 ms prior to and after saccade onset), performance of subjects was very high (97.1%), but significantly smaller than perfect (i.e. 100%. one-sided signed rank test, $p < 0.0001$). If saccadic suppression would have strongly compromised self-motion perception, we would have expected a drop in performance close to chance level (50%) in the temporal vicinity of a saccade. Yet, discrimination performance <50 ms prior to and after saccade onset was almost as high as during fixation, i.e. 96.1%. While this value was significantly smaller than 100% (one-

sided signed rank test, $p < 0.0001$), it was also significantly larger than chance level (one-sided signed rank test, $p < 0.001$). A paired, one sided Wilcoxon signed rank test found no difference in performance in the perisaccadic interval compared to fixation ($p > 0.5$). Effect size of the perisaccadic perceptual modulation, d' , was 0.21, i.e. comparably small. Accordingly, we conclude that perisaccadic optic flow perception *per se* was only marginally modulated”.

Reviewer #3 (Remarks to the Author):

Reviewer-3, Question-1: *It is known that many neurons in MST and VIP are tuned to heading direction and the spatial tuning of these neurons can be determined using a linear regression technique. This paper it is noted that neurons in MST and VIP greatly decrease their activity after a saccade in monkeys. We the heading is decoded during this period of decreased activity it was greatly compressed after the saccade with a maximum effect about 80 ms after the saccade onset. The study went on to see if this compressed heading after a saccade was actually perceived in humans. Not only are the findings novel, but the approach is novel as well since usually perceptual illusions are known decades before the underlying neurophysiology is understood. In this instance the authors first looked at the neurophysiology the found the predicted illusion, event though this illusion had not been previously described.*

We thank the reviewer for this positive view.

R-3, Q-2: *In Fig. 1C what is shown is a decrease in activity during the period after the saccade and this brief suppression is also mentioned in the text. Was it generally the case that activity was suppressed during this period or was that just for this example? If it was generally suppressed then could this be a mechanism by which heading estimation was essentially suppressed during this period to prevent confusion due to motion on the retina due to eye motion during the saccade interfering with heading perception? One could imagine that the general suppression might cause the heading estimates to be compressed around straight ahead because there are many more neurons with best sensitivities occurring near straight ahead (Gu et al., 2010). Is it possible that this is actually what is occurring?*

We thank the reviewer for making this important point. It had also been made by the other two reviewers and we have answered in detail in our response to reviewer 1's question-1 (please see above, **R-1, Q-1**). In order to shorten this response letter, we would like to refer to the details given there. In brief: thanks to the reviewers' comments, we have analyzed the perisaccadic response behavior in much greater detail. It turned out that saccadic suppression was only observed when neurons were stimulated with their preferred heading (we termed this a congruent condition). When neurons were stimulated with the non-preferred heading (we termed this an incongruent condition), perisaccadic activity was not suppressed but rather revealed a brief response enhancement, most likely as a consequence of a release from inhibition. As deduced from our quantitative model, it is only this simultaneous interplay of saccadic suppression of neurons in the congruent condition and response

enhancement of neurons in the incongruent condition which leads to a compression of the decoded heading. Importantly, we find exactly this predicted response behavior in our data set. The more detailed explanation can be found in response to the first question of reviewer 1 (**R-1,Q-1**) and in the Results and Discussion section of the revised manuscript.

In addition, our control experiment in humans, although a discrimination task, clearly showed that subjects were not literally blind for self-motion stimuli presented across saccades. Instead, they could discriminate between forward and backward self-motion in more than 96% of the cases. We also elaborate on this point in our response to **R-1,Q-3** and in the Results and Discussion sections of the revised manuscript.

R-3, Q-3: *One question I have is why does this occur? Is it a mechanism to suppress potentially confounding visual stimuli that occur during the rapid eye movement of the saccade? I realize this is not something that can be known from the current data but it may be worth some speculation: One interesting phenomenon is that during visual heading estimation a population vector decoder model of neuronal activity in MST suggests (Gu et al., 2010) the horizontal component visual headings will be over estimated. This effect has been minimal when headings choices only cover a limited range that is known to or quickly learned by the subjects as done here and in other prior studies (D'Avossa and Kersten, 1996; Telford and Howard, 1996). However when the full range of visual headings is possible these headings tend to be overestimated (Crane, 2012; Cuturi and Macneilage, 2013) consistent with the predictions made from MST neurophysiology. Additionally visual headings are in retinotopic coordinates in MST and these coordinates also seems to be preserved in perception (Crane, 2015). Of course many naturally occurring situations, people and animals tend to orient their eye position so that it is centered near the focus of expansion (FOE), which for a straight path is the heading direction which would tend to minimize the overestimation of eccentric headings. Although this is not true of the experimental situation, in many situations a saccade would bring the eye away from the FOE or back to it. In either of these situations it may be advantageous to have the eccentricity of the heading minimized: In the away situation because the heading is actually being maintained but the eye is making a saccade to an eccentric target. In the toward situation because the heading is now being minimized. I admit this is kind of a kludge solution which would not be advantageous in every solution.*

We thank the reviewer for making this point and for further considerations on this issue. This comment, together with the remarks of Reviewers 1 and 2, has guided us to design an additional psychophysical experiment, which serves also as a control. Again, in order to shorten this response letter, we would like to refer to our detailed response to **R-1,Q-2**. In brief: in this additional and new set of experiments, we compared the perceived heading in the original setting, i.e. with gaze and head directed straight-ahead, with two new conditions in which either (i) gaze was shifted $\alpha = 15$ degrees to the left, with the head directed straight-ahead, or (ii) with gaze directed straight-ahead and the head rotated $\alpha = 15^\circ$ to the left (Results shown in Figure 14 of the

revised manuscript). In a previous study, Crane had used a somewhat similar approach when comparing heading perception in human subjects when providing pure visual, pure vestibular or combined visual-vestibular stimulation (Crane, 2015). This study found that visual headings were biased towards eye but not head position. Our finding is in line with this previous result. During steady fixation, the mean perceived heading was shifted in the direction of the eccentric eye position by approximately one-third of the gaze amplitude. This value is somewhat smaller than the one from Crane's study (46%). Yet, like in Crane (2015), a peripheral head position with eyes straight-ahead did not induce a shift of the mean perceived heading. Crane (2015) concluded that the perception of visually simulated self-motion is organized in eye-centered coordinates, which would be in line with neurophysiological data from the animal model, i.e. the macaque monkey (area MST: Fetsch et al., 2007; area VIP: Chen et al., 2013). Our data are perfectly in line with this hypothesis, which, in turn, provides further evidence for our idea that the observed compression of perceived heading might be based on response properties of neurons in primate areas MST and VIP. We elaborate on this in the Discussion as follows:

Discussion: "Our second behavioral experiment allowed us to determine the reference frame of perisaccadic compression of perceived heading. A previous study had found that, during fixation, perceived direction of visually simulated self-motion was biased towards eye but not head position (Crane, 2015). Our findings are in line with this previous result. During steady fixation, the mean perceived heading was shifted in the direction of the eccentric eye position by approximately one-third of the gaze amplitude. This value is somewhat smaller than the one from (Crane, 2015) (46%). Yet, like in (Crane, 2015), a peripheral head position with eyes straight-ahead did not induce a shift of the mean perceived heading. It was concluded that the perception of visually simulated self-motion is organized in eye-centered coordinates, consistent with neurophysiological data from the animal model, i.e. the macaque monkey (Area MST: Fetsch et al., 2007. Area VIP: Chen et al., 2013). Our data are perfectly in line with this hypothesis, which, in turn, provides further evidence for our idea that the observed compression of perceived heading might be based on response properties of neurons in primate areas MST and VIP."

R-3, Q-4: *The numbering of the figures doesn't make sense. Instead of Figs 1-4, then supplemental Fig. 1, why not just have Figs 1-5?*

Based on all three reviewers' comments we have significantly revised our manuscript. We now present 15 figures in the main text and one additional figure as supplement. We have renumbered all figures accordingly.

R-3, Q-5: *Overall I found this to be a very high quality report and I would endorse publication.*

Thanks!

Crane BT (2012) *Direction Specific Biases in Human Visual and Vestibular Heading Perception*. PLoS One 7:e51383.

Crane BT (2015) *Coordinates of Human Visual and Inertial Heading Perception*. PLoS One 10:e0135539.

Cuturi LF, Macneilage PR (2013) *Systematic biases in human heading estimation*. PLoS One 8:e56862.

D'Avossa G, Kersten D (1996) *Evidence in human subjects for independent coding of azimuth and elevation for direction of heading from optic flow*. Vision research 36:2915-2924.

Gu Y, Fetsch CR, Adeyemo B, Deangelis GC, Angelaki DE (2010) *Decoding of MSTd population activity accounts for variations in the precision of heading perception*. Neuron 66:596-609.

Telford L, Howard IP (1996) *Role of optical flow field asymmetry in the perception of heading during linear motion*. Percept Psychophys 58:283-288.

Cited literature:

Bremmer F, Duhamel J, Ben Hamed S, Graf W (2002) Heading encoding in the macaque ventral intraparietal area (VIP). Eur J Neurosci 16:1554–1568.

Bremmer F, Kubischik M, Hoffmann K-P, Krekelberg B (2009) Neural Dynamics of Saccadic Suppression. JNeurosci 29:12374–12383.

Britten KH (2008) Mechanisms of self-motion perception. AnnuRevNeurosci 31:389–410.

Burr DC, Ross J, Binda P, Morrone MC (2010) Saccades compress space, time and number. Trends Cogn Sci 14:528–533.

Castet E, Masson GS (2000) Motion perception during saccadic eye movements. NatNeurosci 3:177–183.

Chen A, DeAngelis GC, Angelaki DE (2011) Representation of vestibular and visual cues to self-motion in ventral intraparietal cortex. JNeurosci 31:12036–12052.

Chen X, DeAngelis GC, Angelaki DE (2013) Eye-centered representation of optic flow tuning in the ventral intraparietal area. JNeurosci 33:18574–18582.

Cohen J (1988) Statistical power analysis for the behavioral sciences. L. Erlbaum Associates.

Crane BT (2012) Direction specific biases in human visual and vestibular heading perception. PLoS ONE 7:e51383.

Crane BT (2015) Coordinates of Human Visual and Inertial Heading Perception. PLoS One 10:e0135539.

Cuturi LF, MacNeilage PR (2013) Systematic biases in human heading estimation Lappe M, ed. PLoS ONE 8:e56862.

d'Avossa G, Kersten D (1996) Evidence in human subjects for independent coding of azimuth and elevation for direction of heading from optic flow. Vis Res 36:2915–2924.

Duffy CJ (1998) MST neurons respond to optic flow and translational movement. JNeurophysiol 80:1816–1827.

Duffy CJ, Wurtz RH (1991) Sensitivity of MST neurons to optic flow stimuli. I. A continuum of response selectivity to large-field stimuli. JNeurophysiol 65:1329–1345.

Fetsch CR, Wang S, Gu Y, DeAngelis GC, Angelaki DE (2007) Spatial reference frames of visual,

- vestibular, and multimodal heading signals in the dorsal subdivision of the medial superior temporal area. *JNeurosci* 27:700–712.
- Frost A, Niemeier M (2015) Suppression and reversal of motion perception around the time of the saccade. *Front Syst Neurosci* 9:143.
- Gu Y, Fetsch CR, Adeyemo B, DeAngelis GC, Angelaki DE (2010) Decoding of MSTd population activity accounts for variations in the precision of heading perception. *Neuron* 66:596–609.
- Hartung J, Knapp G, Sinha BK (2011) *Statistical Meta-Analysis with Applications*. New York, NY: Wiley.
- Ilg UJ, Hoffmann K-P (1993) Motion perception during saccades. *Vision Res* 33:211–220.
- Lappe M, Awater H, Krekelberg B (2000) Postsaccadic visual references generate presaccadic compression of space. *Nature* 403:892–895.
- Lappe M, Bremmer F, Pekel M, Thiele A, Hoffmann K-P (1996) Optic flow processing in monkey STS: A theoretical and experimental approach. *J Neurosci* 16:6265–6285.
- Lappe M, Bremmer F, Van Den Berg AV V (1999) Perception of self motion from visual flow. *Trends Cogn Sci* 3:329–336.
- Maciokas JB, Britten KH (2010) Extrastriate area MST and parietal area VIP similarly represent forward headings. *J Neurophysiol* 104:239–247.
- Morrone MC, Ross J, Burr DC (2005) Saccadic eye movements cause compression of time as well as space. *NatNeurosci* 8:950–954.
- Ross J, Edwin Dickinson J (2007) Effects of adaptation to Glass pattern structure and to path of optic flow. *Vision Res* 47:2150–2155.
- Schmolesky MT, Wang Y, Hanes DP, Thompson KG, Leutgeb S, Schall JD, Leventhal AG (1998) Signal timing across the macaque visual system. *JNeurophysiol* 79:3272–3278.
- Telford L, Howard IP (1996) Role of optical flow field asymmetry in the perception of heading during linear motion. *PerceptPsychophys* 58:283–288.

Reviewers' comments:

Reviewer #1 (Remarks to the Author):

I would like to thank the authors for their detailed rebuttal and the tremendous effort they put into the revision of this paper. But I would also like to point out that reviewer time is also valuable, and many of their points could have been made much more compactly.

My general take on the paper is that while each of the two phenomena is interesting, the case that the two phenomena are causally linked is not strong. Their rebuttal to this question as raised by referee #2 (R2, Q2) is not at all convincing. The negative latency of the perceptual effects strongly implies an extraretinal origin, while the positive latency of the physiological effect would more likely be consistent with a retinal origin. The idea that we perceive events before neurons can respond to them, because of neural latencies, makes little sense. The paper would need to be re-written to remove all claims of causality.

Also, given the paucity of neurons from monkey R, it is not clear to me that a paper based on this sample really meets the spirit of the convention that major new results need to be replicated in two animals. (See point 5 below).

The rest of my review will follow the numbering of the original review. I also have a new concern that I denote with the notation 1n, at the end of the review.

1) It is an interesting finding that neurons tend to increase their responses to non-preferred stimuli while decreasing their responses to preferred stimuli. However, the strong claim that global suppression does not produce compression is not borne out by their own analysis (Figure 9); the difference is one of degree. Such differences of degree could be dependent on the details of the readout model used to link neurons to perception; the linear model is nice for its simplicity but is unlikely to be what is used by the brain. After all, it would require a separate lookup table or inverse transform to be maintained for each neuron in the population. And while neurons are approximately monotonic across a limited range of central headings, they certainly aren't linear. Since the quantitative model assumed a linear tuning function, my confidence in the results is somewhat limited.

2) In their response to this question, the authors rest on a paper from the literature that used considerably different methods from theirs (Cuturi and MacNeilage, 2013). However, their own results differ qualitatively from those, in that the subjects in this work show an overall centripetal bias, which would be consistent with a prior of forward headings. This seems very important, and will need much more extensive discussion (and possibly evidence) than they currently give it. But in any case, the data from their own work do not support the main point they brought up in rebuttal, since in this paper, the biases show a central prior.

3) The authors misunderstood my suggestion; I apologize since my wording was ambiguous. Their statistics were fine; I was talking about how easy the task was. The d' I was talking about was the traditional Green and Swets version, not its use in calculation of effect sizes. The main point was that the control task was extremely easy, so it was insensitive to moderate changes in performance (unlike the main task). I don't think there is an analytic solution to this problem; the control data are simply weak to exclude the hypothesis of nonspecific effects. They would need either to collect fresh control data or extensively qualify the interpretation of this experiment.

4) See above (point 2). Inclusion of the single subject data is very helpful. Thanks.

5) As mentioned above, this is an editorial judgment call. The sample of neurons from the two monkeys is so uneven that one could not meaningfully test to see if the results were similar or different between the monkeys. As I understand the editorial conventions in the field, primary results need to come from two monkeys for this reason, to allow a test of the consistency of results. However, there might be enough data to meaningfully compare the results between MST and VIP, and this might be interesting. They certainly should not be pooled until they are shown to be similar according to the metrics used in this work. I leave this question for the editors.

6) Nicely addressed.

1n) This would fall under the category of significant concerns, and contains two related issues. On reflection, the linear decoder seems less satisfactory than ever. The authors are aware of the excellent 2010 paper by Gu et al, using state-of-the art decoding methods to ask a similar question. So it seems strange that they would employ a simpler method which certainly is only a crude approximation to what the population is doing. The errors that are introduced might be substantial since it would be at extreme headings where the linear model would overestimate performance, because of floor and ceiling effects. These estimates would probably have undue leverage on the population estimate. At the very least, they should try a more realistic method to convince the reviewers that their results are robust against the underlying assumptions. I would suggest the Fisher information approach as being the most unbiased. Relatedly, I was unable to find out whether the linear decoder allowed modeled neuronal firing rates to become negative; as written it seems possible. This could easily lead to substantial errors.

Reviewer #2 (Remarks to the Author):

The authors have addressed my concerns. The additional analyses and modeling examining the relationship between suppression and compression has greatly strengthened the paper. The additional psychophysical study demonstrating that compression operates in retinal coordinates is also valuable. My primary suggestion is to reduce the number of figures; 15 seems excessive for a communication paper. This can be done by combining figures and moving less vital information to the supplement.

I suggest the following:

Combine Figs. 3, 7 and 8 into one figure

Combine Figs. 12, 14 and 15 into one figure, perhaps moving some elements to the supplement

Move Figs. 11 and 13 to the supplement.

Remove table 1 and put this information into the text.

This would reduce the total figure count to 9 which seems more appropriate.

Otherwise, I would suggest adding another suppression model, or even replacing the current global suppression model with a global-and-proportional suppression, i.e. suppression in proportion to overall activity rather than a uniform decrease. Referring to figure 4C left panel, a smaller suppression of subpopulation 1 (i.e. in proportion to the reduced activity) would lead to a compression. This seems a more reasonable model for suppressive or inhibitory processes. Such a model would, nevertheless, seem to be refuted by the release from suppression revealed by later analyses. So this additional analysis would not detract from the impact of the current results.

Regarding the release from suppression shown for the population (Fig 7, right panel), it appears this may be driven by a few very extreme neurons with a high incongruent modulation index (see Fig. 6). It would be helpful if the authors could comment on this.

Minor comments:

Line 414: veridically

Line 427: remove comma after "both"

Reviewer #3 (Remarks to the Author):

The authors did an excellent job of addressing my previous criticisms. The paper is ready for publication in my opinion.

We thank the reviewers for their very valuable and insightful comments which helped us to further improve our manuscript. Like for the first revision, we respond to the comments point-by-point by using the following fonts/styles to indicate:

Reviewers' comments (Calibri, bold, italics)

Our response to these comments (Courier)

New text in the revised, red-lined version of the manuscript. (Calibri, red)

Our revisions of the original manuscript had been quite substantial. Accordingly, also the response-to-the-reviewers was very extensive. Based on the reviewers' feedback, in this response to their remarks, we aim to be more compact, but at the same time as precise as possible.

We would like to begin with a rather general comment:

- The neurophysiological part of our manuscript is based on data from **two cortical areas** (MST and VIP) recorded in **three monkeys**. Throughout the manuscript, we consider data from both areas together. This is, because we relate our findings to human behavior. Based on previous imaging work (e.g. Bremmer et al., 2001; Huk et al., 2002; Wall and Smith, 2008), it is most likely, that the functional equivalents of macaque areas MST and VIP are involved in the heading perception task, as described in the human behavioral part of our manuscript. The subjects' behavior is based on the brain's processing of self-motion information as a whole. We hence do not aim to relate the observed human perception to processing only in area MST or area VIP. This rather holistic approach is supported by neurophysiological studies in the macaque which have documented strong functional similarities and overlapping response properties concerning optic flow processing between areas MST and VIP (e.g. Lappe et al., 1996; Bremmer et al., 2002, 2010; Britten, 2008; Angelaki et al., 2011; DeAngelis and Angelaki, 2012; Kaminiarz et al., 2014). We hence consider it a strength of our study rather than a weakness to consider data from both areas together as neural correlate of human perception.

Like in the first revision and in order to facilitate navigation throughout our reply, we have numbered the questions asked by the reviewers and refer to these numbers in our reply.

Finally, according to Nature Communications' *Manuscript Checklist*, the previous version of our manuscript has lacked a summarizing paragraph, written in present tense, at the end of the Introduction and a statement on data availability at the end of the Methods. We have now added these sections and have re-written the Abstract in present tense.

Reviewers' comments:

Reviewer #1 (Remarks to the Author):

R1, Q1: I would like to thank the authors for their detailed rebuttal and the tremendous effort they put into the revision of this paper. But I would also like to point out that reviewer time is also valuable, and many of their points could have been made much more compactly.

My general take on the paper is that while each of the two phenomena is interesting, the case that the two phenomena are causally linked is not strong. Their rebuttal to this question as raised by referee #2 (R2, Q2) is not at all convincing. The negative latency of the perceptual effects strongly implies an extraretinal origin, while the positive latency of the physiological effect would more likely be consistent with a retinal origin. The idea that we perceive events before neurons can re-

spond to them, because of neural latencies, makes little sense. The paper would need to be re-written to remove all claims of causality.

We thank the reviewer for appreciating our effort to clarify open questions and apologize for being perhaps a bit too detailed.

Extraretinal effects: We fully agree with the reviewer that the observed effects are most likely due to extraretinal effects. This, however, is by no means in contradiction to our neurophysiological data. There is clear experimental evidence that extraretinal effects contribute to saccadic suppression. As an example, Diamond and colleagues have tested this explicitly in human observers and found an increase in contrast-sensitivity only for real but not simulated saccades (Diamond et al., 2000). The peak of saccadic suppression as measured behaviorally is typically found around the time of saccade onset. In a previous work we have identified a neural correlate of saccadic suppression by recording impulse response functions in extrastriate and parietal areas (MT, MST, LIP, and VIP) of the macaque monkey (Bremmer et al., 2009). The time course of saccadic suppression as found in motion sensitive areas MT, MST, and VIP was very similar to the behavioral time course found in humans. Yet, physiological data were "delayed" with respect to the behavioral data by the average response latency of the neurons in these areas, i.e. approximately 100ms. Nevertheless, also in the neurophysiological data-set, saccadic suppression started to build up earlier and hence, must have been caused by an (yet not identified) extraretinal signal.

In our current study, the peak of saccadic suppression occurred 80ms after saccade onset, i.e. perfectly in line with data from Bremmer et al., 2009. Suppression, however, started to build up much earlier (see Figure 3). This strongly argues for an extraretinal effect, too, and, hence, is fully in line with previous neurophysiological and our current behavioral data.

We had discussed this point already explicitly in the previous version of our manuscript. In the new version, it can be found in the paragraph starting at line 516. Given that the other two reviewers did not ask for a further elaboration, we suggest to keep the text as is.

Timing: As to the question of timing of neurophysiological and behavioral effects. We see the point. Yet, we do not follow the reviewer's view that the peak of compression in the neurophysiological data has to precede the peak in the behavioral data. This is for the following reason: In psychophysics, when human subjects are asked to respond behaviorally to a stimulus, the time axis in the resulting data plot does not indicate the time of the subjects' percept, but the time of the stimulus. To be more specific, let's consider a virtual experiment, in which a visual stimulus was presented at $t=0$ ms. In this experiment, the subjects' task was to judge, when the stimulus has occurred. In such case, they will respond, on average, that the stimulus occurred at $t=0$ ms. If one would record neural activity of the subjects' visual system (e.g. by means of EEG), the first neural signature of the response to this visual stimulus, however, would occur probably at some point between $t=40-120$ ms, i.e., in all cases a considerable amount of time after the stimulus itself. This "delay" of the neural response is simply due to processing latencies in the retina and the following stages of visual processing. Why is

“delayed” neural response not in conflict with behavior, though? The simple answer is that the response of the subjects, which is based on the “delayed” neural response, occurs much later; in case of a verbal response in the range of seconds after the stimulus.

Accordingly, when comparing neural responses and perception, neurophysiological data have to be corrected for the neural processing latency. Based on data by Schmolesky and colleagues (Schmolesky et al., 1998), response latencies in extrastriate and parietal cortex are in the order of 100ms. In our cross-correlation analysis, we find a delay of the peak-of-compression in the neurophysiological data with respect to the peak-of-compression in the behavioral data of $\tau = 116\text{ms}$ (Figure 9), i.e. fully in line with the data by Schmolesky and colleagues.

Again, we had discussed this point already explicitly in the previous version of our manuscript. In the new version, it can be found in the paragraph starting at line 492. Hence, we suggest to keep the text as is.

Causality: We agree with the reviewer that our data cannot prove causality. We have carefully checked the previous version of our manuscript. There, we have at no point stated that the observed neurophysiological effects caused the observed behavioral effects. What we showed and mentioned in the text was a correlation between neurophysiological and behavioral data. We mention this now even more explicitly in the discussion as follows (line 512):

“It has to be mentioned, though, that correlation, as employed in our analyses, cannot prove causality. Such a strict prove as based e.g. on reversible inactivation of both areas simultaneously, however, would have been beyond the scope of our study.”

R1, Q2: Also, given the paucity of neurons from monkey R, it is not clear to me that a paper based on this sample really meets the spirit of the convention that major new results need to be replicated in two animals. (See point 5 below).

We have answered to this comment in a more general way already prior to our individual responses to the three reviewers.

Neurophysiological data-set: At this point, we would like to add a few more details. As discussed previously, neurophysiological data had been recorded before (Area MST: Bremmer et al., 2010; Area VIP: Kaminiarz et al., 2014). In these studies, we had investigated self-motion tuning of neurons from monkey areas MST and VIP during simulated and real eye movements. In the MST study, we had recorded $n_{R,MST} = 23$ neurons from monkey R and $n_{H,MST} = 61$ neurons from monkey H (i.e.: $n_{MST} = 84$). In the VIP study, we had recorded $n_{H,VIP} = 45$ neurons from monkey H and $n_{C,VIP} = 23$ neurons from monkey C (i.e.: $n_{VIP} = 68$). These values are in the standard range used in neurophysiological studies on macaque monkeys.

From this pool of $n_{SimEyeMove} = 152$ neurons ($n_{MST} = 84 + n_{VIP} = 68$ neurons), $n_{RealEyeMove} = 119$ neurons could be kept long enough for stable recordings and were tested during real eye-movements: $n_{R,MST} = 9/23$ neurons from monkey R and $n_{H,MST} = 55/61$ neurons from monkey H from recordings in area MST; $n_{H,VIP} = 36/45$ neurons from monkey H and $n_{C,VIP} = 19/23$ neurons from monkey C from recordings in area VIP. It is data from this sample which is reported in the current manuscript. We

employed a rather strict criterion to include neurons from this data pool in our decoding approach. 71/119 neurons fulfilled this strict criterion.

As mentioned above, we consider it a strength of our study to consider data from areas MST and VIP together as neural correlate of human perception. Nevertheless, in order to answer the reviewer's comment, we determined for each animal and each area the activity in the congruent and incongruent stimulus conditions (modulation index, Figure 6). The differential response behavior in these two conditions was one of the key findings of our analysis and was the basis for setting up our model (Figure 7). A modulation index smaller than 1.0 indicates response suppression, while a value larger than 1.0 indicates response enhancement. For the full population (two areas, three monkeys), we have reported in the manuscript a median value of **0.812** for the congruent condition and **1.206** for the incongruent condition. Importantly, we find this response behavior of suppression in the congruent condition and enhancement in the incongruent condition in each animal and each area, independent of the number of neurons, as detailed in the table below.

Area / Monkey	Congruent	Incongruent
MST / R	0.735	1.089
MST / H	0.916	1.212
VIP / H	0.845	1.378
VIP / C	0.650	1.262
Full population	0.812	1.206

In summary: data come from three animals and should be considered together (as we do in our manuscript), since we compare neural responses to human behavior, which most likely is based on the computation performed in both areas, i.e. the functional equivalents of monkey areas MST and VIP. Nevertheless, even when breaking up our population into the individual parts, also for comparably small numbers of neurons, the critical response feature (suppression vs. enhancement) is conserved.

R1, Q3: The rest of my review will follow the numbering of the original review. I also have a new concern that I denote with the notation 1n, at the end of the review.

1) It is an interesting finding that neurons tend to increase their responses to non-preferred stimuli while decreasing their responses to preferred stimuli. However, the strong claim that global suppression does not produce compression is not borne out by their own analysis (Figure 9); the difference is one of degree. Such differences of degree could be dependent on the details of the readout model used to link neurons to perception; the linear model is nice for its simplicity but is unlikely to be what is used by the brain. After all, it would require a separate lookup table or inverse transform to be maintained for each neuron in the population. And while neurons are approximately monotonic across a limited range of central headings, they certainly aren't linear. Since the quantitative model assumed a linear tuning function, my confidence in the results is somewhat limited.

We thank the reviewer for appreciating our key finding of differential responses in the congruent vs. the incongruent stimulus condition. The further comment has two major points: (i) linearity and (ii) the effect of global suppression.

Linearity: We agree with the reviewer that the linear model might not fully catch the brain's way to represent heading information. Yet, this was by no means the goal of our approach. Instead, the linear function was an approximation which allows to model the effects of response suppression on heading perception quantitatively.

In his/her further comments (see below, R1-Q9), the reviewer refers to an excellent study by Gu, Angelaki, DeAngelis and colleagues. In this study, the authors presented self-motion stimuli in the horizontal plane, with heading directions separated by 45° . Figure 3A of their paper, depicted above in panel A, shows the tuning of a neuron from area MST as black solid line. Sample points are indicated by black circles, including standard error bars (The red curve indicates Fisher information, which we refer to below in response to R1-Q9). In panel B, we have graphically superimposed two alternative tuning functions: a sigmoidal (green, dashed line, shown separately in panel C) and a linear function (blue, solid line, shown separately in panel D). Importantly, within the central $\pm 45^\circ$, all three functions (black, green and blue line) fit the data well. Differences become apparent only beyond $\pm 45^\circ$. In our study, we have tested a comparably limited range of heading directions ($\pm 30^\circ$, indicated by the greenish and blueish shaded areas in panels C and D)). As illustrated in panel B, this range did not allow to determine (i) response saturation values, which would have been required for fitting a sigmoidal function, or (ii) local maxima or minima, as would have been necessary for fitting a rather complex tuning function, i.e. the black line in panels A and B.

In summary, given a set of three data points within the central ± 30 degrees, a linear function is a well-motivated and plausible statistical model, which allows quantitative modeling of the effects of saccades on the representation of heading information in macaque areas MST and VIP.

Effect of global suppression: Based on this comment, we have now quantified the compression resulting from global vs. selective suppression in our model (see panels in the bottom row in the figure below). Like for the neurophysiological and the behavioral data, compression was computed as the normalized standard deviation of the five data curves. As can be seen from this new part of the figure, global saccadic suppression results in a rather small compression, whereas selective saccadic suppression results in a strong compression, similar to our neurophysiological and behavioral findings. In the previous version of our manuscript we had concluded that in case of global suppression "there was only a marginal modulation of decoded heading shortly after the saccade." We still consider this an

appropriate description of our results. For the sake of completeness, we replaced former Figure 9 (showing only the time courses of decoded heading) by the figure shown below (new Figure 7), i.e. including the time-course of compression.

We consider both points, linearity and global suppression, in the Results and Discussion sections as follows:

Effect of global suppression, Results, line 281: "Importantly, when applying global suppression (top left panel), there was only a marginal modulation of decoded heading shortly after the saccade, resulting in a minimal compression as indicated the bottom left panel. Like for the physiological data, we determined compression as normalized standard deviation of the five time-courses of decoded heading."

Figure legend of Figure 7: "This resulted in a minimal compression as indicated in the bottom left panel. Like for the physiological data, we determined compression as normalized standard deviation of the five time-courses of decoded heading."

Linearity, Discussion, line 460: "In our data analysis, we have employed linear regression analysis for quantifying the neurons' tuning for self-motion direction. This approach is in line with previous studies (e.g. Bremmer et al., 2002). Other studies have employed sigmoidal (e.g. Lappe et al., 1996) or more complex tuning functions (e.g. Gu et al., 2010) to model heading selectivity. In our study, we have tested a limited range of heading directions ($\pm 30^\circ$). This range did not allow to determine (i) response saturation values, which would have been required for fitting a sigmoidal function, or (ii) local maxima or minima, as would have been necessary for fitting a rather complex tuning function. Hence, given a set of three data points within the central ± 30 degrees, a linear function appeared to be a well-motivated and plausible statistical model, which allowed quantitative modeling of the effects of saccades on the representation of heading information."

R1, Q4: 2) In their response to this question, the authors rest on a paper from the literature that used considerably different methods from theirs (Cuturi and MacNeilage, 2013). However, their own results differ qualitatively from those, in that the subjects in this work show an overall cen-

tripetal bias, which would be consistent with a prior of forward headings. This seems very important, and will need much more extensive discussion (and possibly evidence) than they currently give it. But in any case, the data from their own work do not support the main point they brought up in rebuttal, since in this paper, the biases show a central prior.

At first glance, our finding of a central bias might appear at conflict with the results from (Crane, 2012; Cuturi and MacNeilage, 2013). Yet, given the completely different experimental settings in these previous and our current study, it might not be appropriate to put both results in contrast. Stimuli in our study were extremely short (40ms), while those employed in the other two studies lasted beyond 1s. It is known e.g. from literature on localization of briefly flashed stimuli that these are perceived closer to the fovea than they actually were (e.g. Müsseler et al., 1999; Lappe et al., 2000; Kaminiarz et al., 2007). Yet, when targets are presented continuously, overestimation of their eccentricity has been reported (e.g. Bock, 1993). In other words: in similar tasks, a bias can strictly depend on the experimental conditions.

As detailed in the previous version of our manuscript, the new behavioral dataset (peripheral gaze vs. peripheral head direction) had shown a shift of the bias towards the periphery during peripheral fixation. This shift is similar to the one shown by Crane (2015). Like in this previous work, the shift towards gaze direction was not complete. While it was 46% in Crane's study, it was about one third in our experiment (i.e. 5° vs. 15°). Perisaccadically, the compression was directed fully towards the center of gaze. This could imply that the perisaccadic compression of perceived heading is different from any perceptual bias during fixation, be it centrifugal as in the study by Cuturi and MacNeilage, 2013, and Crane, 2012, or centripetal, as in our study. To fully answer this question, a completely new study focusing on this issue would be necessary.

Following the reviewer's advice, we detail the report and the discussion of our findings in the revised manuscript as follows:

Results, line 405: "In the gaze-turned-head-straight-condition, however, also long before and after the saccade, perceived heading was biased away from straight-ahead. Perisaccadically, this biased perception again was strongly modulated (Panel G)."

Figure 10: "In the gaze-turned-head-straight-condition, however, also long before and after the saccade, perceived heading was biased away from straight-ahead, towards the direction of gaze. In the temporal vicinity of the saccade, the averaged perceived heading was in line with the direction of gaze (Panel G). At the time of maximum compression, all headings were perceived as being 15° to the left. The findings for this subject were representative for the group of subjects, as indicated in panels D and H. The thick blue line indicates the time-resolved average value across all five subjects. The thin blue lines indicate the time-resolved standard error."

Discussion, line 432: "In the main behavioral experiment, self-motion perception during fixation was biased towards straight-ahead, which here was also the direction of gaze. One could assume that in a Bayesian sense, this central bias would reflect a behavioral prior towards the most common heading direction in everyday life. A previous study has indeed investigated priors for heading perception (Cuturi and MacNeilage, 2013). Surprisingly, also to the authors, the observed prior was not centered on straight-ahead. Instead, when testing heading in the full horizontal 2-D motion space, two peaks directed 90 degrees to the left and to the right, i.e. perpendicular with respect to straight-ahead, evolved. In addition, Crane (Crane, 2012) showed that the centrifugal bias was more pronounced when heading could be in any direction within the full 2-D horizontal plane as compared to an experimental context, in which heading was restricted within a ninety-degree sector centered

on straight-ahead. This finding was in line with results from previous studies (d'Avossa and Kersten, 1996; Telford and Howard, 1996).

At first glance, our finding of a central bias appears at conflict with the results from (Cuturi and MacNeilage, 2013) and (Crane, 2012). Yet, given the complete different experimental settings in these previous and our current study, it might not be appropriate to put both results in contrast. Stimuli in our study were extremely short (40ms), while those employed in the other two studies lasted beyond one second. It is known e.g. from literature on localization of briefly flashed stimuli that these are perceived closer to the fovea than they actually were (e.g. Müsseler et al., 1999; Lappe et al., 2000; Kaminiarz et al., 2007). Yet, when targets are presented continuously, overestimation of perceived location has been reported (Bock, 1993). In other words: in similar tasks, a bias can strictly depend on the experimental conditions.

Cuturi and MacNeilage (2013) and Crane (2012) suggested that the observed centrifugal bias might result from a non-uniform distribution of heading preferences in primate areas MST and VIP, as described in the macaque among others by Angelaki, DeAngelis and colleagues (MST: Gu et al., 2010; VIP: Chen et al., 2011). In their study on area MST, these authors have investigated accuracy and precision of decoded heading from discharges of monkey area MST and found a decoding bias towards the periphery. Indeed, long before and especially long after a saccade, also our decoding from areas MST and VIP revealed a (slight) centrifugal bias (see our Figure 2). Accordingly, during slow eye-movements, our data are in good agreement with previous results obtained during steady fixation."

R1, Q5: 3) The authors misunderstood my suggestion; I apologize since my wording was ambiguous. Their statistics were fine; I was talking about how easy the task was. The d' I was talking about was the traditional Green and Swets version, not its use in calculation of effect sizes. The main point was that the control task was extremely easy, so it was insensitive to moderate changes in performance (unlike the main task). I don't think there is an analytic solution to this problem; the control data are simply weak to exclude the hypothesis of nonspecific effects. They would need either to collect fresh control data or extensively qualify the interpretation of this experiment.

Based on the reviewer's comment, we have now elaborated our description and discussion of the control experiment as follows:

Results, line 374: "While the nature of the control experiment (discrimination between forward and backward self-motion) was different from our main experiment (perception of self-motion direction), it clearly shows that subjects were not virtually blind for brief self-motion stimuli as could have been the case due to saccadic suppression."

Discussion, line 470: "In a first behavioral control experiment subjects had to discriminate if a brief self-motion stimulus simulated forward or backward self-motion. It is known from the literature that saccades impair the perception of visual motion (Illg and Hoffmann, 1993). Given that self-motion sequences in our study were extremely short (40ms) we had to exclude the possibility that subjects simply were not able to perceive our stimuli due to saccadic suppression of the magnocellular processing of the visual system (Burr et al., 1994). Obviously, the nature of this control experiment was different from our main experiment, in which subjects had to indicate their perceived self-motion direction. Nevertheless, the results clearly showed that subjects were not literally blind for the self-motion stimuli used in our experiment."

R1, Q6: 4) See above (point 2). Inclusion of the single subject data is very helpful. Thanks.

Thanks.

R1, Q7: 5) As mentioned above, this is an editorial judgment call. The sample of neurons from the two monkeys is so uneven that one could not meaningfully test to see if the results were similar or

different between the monkeys. As I understand the editorial conventions in the field, primary results need to come from two monkeys for this reason, to allow a test of the consistency of results. However, there might be enough data to meaningfully compare the results between MST and VIP, and this might be interesting. They certainly should not be pooled until they are shown to be similar according to the metrics used in this work. I leave this question for the editors.

We have commented on this question already at the very beginning of our response, and, in a more detail, in response to R1,Q2.

We are deeply convinced that our sample of neural data is fully in line with general standards concerning results from awake, behaving monkeys. And, as mentioned before, we consider it a strength of our study rather than a weakness to consider data from both areas (MST and VIP) together as neural correlate of human perception.

R1, Q8: 6) Nicely addressed.

Thanks.

R1, Q9: 1n) This would fall under the category of significant concerns, and contains two related issues. On reflection, the linear decoder seems less satisfactory than ever. The authors are aware of the excellent 2010 paper by Gu et al, using state-of-the art decoding methods to ask a similar question. So it seems strange that they would employ a simpler method which certainly is only a crude approximation to what the population is doing. The errors that are introduced might be substantial since it would be at extreme headings where the linear model would overestimate performance, because of floor and ceiling effects. These estimates would probably have undue leverage on the population estimate. At the very least, they should try a more realistic method to convince the reviewers that their results are robust against the underlying assumptions. I would suggest the Fisher information approach as being the most unbiased. Relatedly, I was unable to find out whether the linear decoder allowed modeled neuronal firing rates to become negative; as written it seems possible. This could easily lead to substantial errors.

Firing rates: Thanks for making this point. As assumed by the reviewer and given our goal to create model neurons with response properties very similar to those of real neurons, model neurons could not have negative firing rates. All firing rates, which would have been negative, were set to zero. We add this information in the Results section as follows:

Results, line 269: **“Importantly, model neurons could not have negative firing rates. Instead, all firing rates, which would have been negative, were set to zero.”**

Linear decoder: We respectfully disagree with the reviewer’s opinion about the role of the linear decoder. We have previously shown that linear regression (in horizontal and vertical direction) can be used to decode heading from a population of VIP neurons (Bremmer et al., 2002). Similarly, we had shown before that sigmoidal functions can be employed to decode heading from a population of MST neurons (Lappe et al., 1996). In both studies, we had simulated horizontal and vertical forward self-motion while animals had to fixate a central target. As shown graphically in our response to R1-Q3, in the range of self-motion directions as used in our current study (i.e. +/- 30° in the horizontal plane), linear, sigmoidal and more complex functions can fit the responses of MST and VIP neurons equally well.

We agree with the reviewer that Gu and colleagues did an excellent job in their 2010 paper. Yet, the key goal of their study was very different to ours. They used Fisher information to show that population activity in area MST predicts the dependence of heading thresholds on eccentricity. In other words: this paper dealt with the precision of heading estimates, as documented by:

- the title: "Decoding of MSTd Population Activity Accounts for Variations in the Precision of Heading Perception"
- the first sentence of their Results section, their page 597: "To quantify the precision with which subjects discriminate heading, seven human subjects and two macaques were tested in a two-interval task in which each trial consisted of two sequential translations, a 'reference' and a 'comparison'".

The paper very convincingly shows that Fisher information is indeed an appropriate and unbiased way of determining thresholds via the sensitivity analysis of the tuning curve for self-motion in full 2-D horizontal space. It does not address the issue of accuracy, though.

Importantly, the key goal of our study was to decode heading from the population data, i.e. to determine heading accuracy. This difference in goals is also explicitly mentioned in the Gu et al., 2010, paper:

- Results, last sentence, their page 603: "Because humans and monkeys performed a relative judgment in our two-interval heading task, our data do not address the accuracy of heading estimation"

Furthermore, the Gu-paper explicitly acknowledges that decoding of heading (i.e. accuracy) can be done in various ways:

- Results, page 601: "Recall, however, that Fisher information provides an upper bound on sensitivity but does not specify a type of decoding. There are multiple ways that MSTd responses could be decoded, and it is beyond the scope of this paper to consider these broadly."

Our linear decoder is one of these "multiple ways". Hence, and in accordance with the paper by Gu and colleagues, we believe that our linear decoder is a simple, statistically plausible, and well-suited way to quantitatively model the effects of saccades on the representation of heading information.

Reviewer #2 (Remarks to the Author):

R2, Q1: The authors have addressed my concerns. The additional analyses and modeling examining the relationship between suppression and compression has greatly strengthened the paper. The additional psychophysical study demonstrating that compression operates in retinal coordinates is also valuable. My primary suggestion is to reduce the number of figures; 15 seems excessive for a communication paper. This can be done by combining figures and moving less vital information to the supplement.

I suggest the following:

Combine Figs. 3, 7 and 8 into one figure

Combine Figs. 12, 14 and 15 into one figure, perhaps moving some elements to the supplement

Move Figs. 11 and 13 to the supplement.

Remove table 1 and put this information into the text.

This would reduce the total figure count to 9 which seems more appropriate.

We thank the reviewer for his/her very positive view of our revised manuscript.

We have followed the reviewer's suggestion to reduce the overall number of figures for all but one. We suggest to keep former Figure 12 (new Figure 9) as stand-alone figure as it is a key figure depicting (i) the behavioral data and (ii) the comparison of the time-courses of the neurophysiological and behavioral data.

Figure legends of the merged figures and references to the respective figures in the main text have been merged and adjusted accordingly.

The information from the table is now given in the Results section (line 109):

"In detail, for area MST it were 5/9 (56%) neurons from monkey R and 44/55 (80%) neurons from monkey H. For area VIP, it were 10/36 (28%) neurons from monkey H and 12/19 (63%) neurons from monkey C."

R2, Q2: Otherwise, I would suggest adding another suppression model, or even replacing the current global suppression model with a global-and-proportional suppression, i.e. suppression in proportion to overall activity rather than a uniform decrease. Referring to figure 4C left panel, a smaller suppression of subpopulation 1 (i.e. in proportion to the reduced activity) would lead to a compression. This seems a more reasonable model for suppressive or inhibitory processes. Such a model would, nevertheless, seem to be refuted by the release from suppression revealed by later analyses. So this additional analysis would not detract from the impact of the current results.

We thank the reviewer for making this point, which slightly overlaps with Question #-3 of Reviewer 1 (i.e. R1,Q3). The graphical depiction in Figure4C was actually meant to reflect the quantitative model, i.e. a global-and-proportional suppression. We fully agree with the reviewer that in its previous form, however, it suggests that global saccadic suppression would induce a reduction of activity by the same absolute amount of activity, i.e. spikes/s. This, however, is not what our quantitative model does. Instead, as put by the reviewer, it acts as "global-and-proportional suppression", as reflected by our term "convolution factor". We have corrected Figure 4C accordingly. This brings also the qualitative model (Figure 4C) in line with the quantitative model (Figure 7), which shows a marginal modulation of decoded heading in case of global suppression and a strong compression of decoded heading in case of selective suppression.

We have adjusted the main text and the description of Figure 4 accordingly:

Main text, line 174: "As a first hypothesis, we assumed that saccadic suppression acts equally on both subpopulations as *global-and-proportional* suppression (Global suppression, Figure 4C, left panel). The mathematical equivalent of this modulation is a multiplication of the current activity with a value smaller than 1.0."

Main text, line 184: "Given that in our decoding approach heading is determined from averaging heading estimates across neurons from both subpopulations, decoded heading (purple arrow) would show a minimal compression, with an increase in variance, i.e. a reduced decoding precision."

Main text, line 197: “Such an increase of activity of the weakly activated cells due to a release from inhibition (i.e. multiplication with a value larger than 1.0) would cause a decoding shift towards straight ahead.

Main text, line 281: “Importantly, when applying global suppression (top left panel), there was only a marginal modulation of decoded heading shortly after the saccade, resulting in a minimal compression as indicated in the bottom left panel. Like for the physiological data, we determined compression as normalized standard deviation of the five time-courses of decoded heading.”

Figure 4: “Strong compression as found in our neurophysiological data can be explained by selective perisaccadic suppression of stimulus driven responses (C, right panel), but not by global suppression of activity (C, left panel). For details see text.”

R2, Q3: Regarding the release from suppression shown for the population (Fig 7, right panel), it appears this may be driven by a few very extreme neurons with a high incongruent modulation index (see Fig. 6). It would be helpful if the authors could comment on this.

Figure 7 (which has become the new Figure 3B) shows the time course of the mean activity, corrected for the mean, i.e. response modulation (magenta line). The purple shaded area depicts the 95% percent confidence interval as determined from bootstrapping (Matlab function *bootci*, with 1000 repeats). Given this statistical treatment, we consider the effect significant. This view is supported by the fact that the width of the confidence interval appears rather stable across the whole time course. If the data curve had been driven by a few outliers, the width of the confidence interval should have clearly enlarged at this point in time, which was not the case.

More importantly, for computing the time courses of response modulation in said Figure, we first subtracted at the single neuron level a neuron’s mean activity from its response time course and then averaged these response modulations across all 71 neurons (mean). The modulation indices (Figure 6), on the other hand, result from normalized data. Here, we computed at the single neuron level the ratio of average neural activity from 40 ms to 160 ms after saccade onset and average activity long after the saccade (300 ms to 450 ms after saccade onset). Normalization per se can result in large differences of the resulting values. Hence, for the indices we determined the median (rather than the mean) value.

Taken together, we consider the time course data of the response modulation as shown in Figure 3B a robust result.

R2, Q4: Minor comments:

Line 414: veridically

Line 427: remove comma after “both”

Done, thanks.

Reviewer #3 (Remarks to the Author):

R3, Q1: The authors did an excellent job of addressing my previous criticisms. The paper is ready for publication in my opinion.

Thanks.

References:

- Angelaki DE, Gu Y, DeAngelis GC (2011) Visual and vestibular cue integration for heading perception in extrastriate visual cortex. *ProcPhysiolSociety(JPhysiol)* 589:825–833.
- Bock O (1993) Localization of objects in the peripheral visual field. *Behav Brain Res* 56:77–84.
- Bremmer F, Duhamel J, Ben Hamed S, Graf W (2002) Heading encoding in the macaque ventral intraparietal area (VIP). *Eur J Neurosci* 16:1554–1568.
- Bremmer F, Kubischik M, Hoffmann K-P, Krekelberg B (2009) Neural Dynamics of Saccadic Suppression. *JNeurosci* 29:12374–12383.
- Bremmer F, Kubischik M, Pekel M, Hoffmann K-P, Lappe M (2010) Visual selectivity for heading in monkey area MST. *Exp Brain Res* 200:51–60.
- Bremmer F, Schlack A, Shah NJ, Zafiris O, Kubischik M, Hoffmann K-P, Zilles K, Fink GR (2001) Polymodal motion processing in posterior parietal and premotor cortex: a human fMRI study strongly implies equivalencies between humans and monkeys. *Neuron* 29:287–296.
- Britten KH (2008) Mechanisms of self-motion perception. *AnnuRevNeurosci* 31:389–410.
- Burr DC, Morrone MC, Ross J (1994) Selective suppression of the magnocellular visual pathway during saccadic eye movements. *Nature* 371:511–513.
- Chen A, DeAngelis GC, Angelaki DE (2011) Representation of vestibular and visual cues to self-motion in ventral intraparietal cortex. *JNeurosci* 31:12036–12052.
- Crane BT (2012) Direction specific biases in human visual and vestibular heading perception. *PLoS ONE* 7:e51383.
- Cuturi LF, MacNeilage PR (2013) Systematic biases in human heading estimation Lappe M, ed. *PLoS ONE* 8:e56862.
- d’Avossa G, Kersten D (1996) Evidence in human subjects for independent coding of azimuth and elevation for direction of heading from optic flow. *Vis Res* 36:2915–2924.
- DeAngelis GC, Angelaki DE (2012) Visual-Vestibular Integration for Self-Motion Perception.
- Diamond MR, Ross J, Morrone MC (2000) Extraretinal control of saccadic suppression. *JNeurosci* 20:3449–3455.
- Gu Y, Fetsch CR, Adeyemo B, DeAngelis GC, Angelaki DE (2010) Decoding of MSTd population activity accounts for variations in the precision of heading perception. *Neuron* 66:596–609.
- Huk AC, Dougherty RF, Heeger DJ (2002) Retinotopy and functional subdivision of human areas MT and MST. *JNeurosci* 22:7195–7205.
- Ilg UJ, Hoffmann K-P (1993) Motion perception during saccades. *Vision Res* 33:211–220.
- Kaminiarz A, Krekelberg B, Bremmer F (2007) Localization of visual targets during optokinetic eye movements. *Vis Res* 47:869–878.
- Kaminiarz A, Schlack A, Hoffmann K-P, Lappe M, Bremmer F (2014) Visual selectivity for heading in the macaque ventral intraparietal area. *J Neurophysiol* 112:2470–2480.
- Lappe M, Awater H, Krekelberg B (2000) Postsaccadic visual references generate presaccadic compression of space. *Nature* 403:892–895.
- Lappe M, Bremmer F, Pekel M, Thiele A, Hoffmann K-P (1996) Optic flow processing in monkey STS: A theoretical and experimental approach. *J Neurosci* 16:6265–6285.
- Müsseler J, van der Heijden AHC, Mahmud SH, Deubel H, Ertsey S, Musseler J, van der Heijden AHC, Mahmud SH, Deubel H, Ertsey S (1999) Relative mislocalization of briefly presented stimuli in the retinal periphery. *PerceptPsychophys* 61:1646–1661.
- Schmolesky MT, Wang Y, Hanes DP, Thompson KG, Leutgeb S, Schall JD, Leventhal AG (1998) Signal timing across the macaque visual system. *JNeurophysiol* 79:3272–3278.
- Telford L, Howard IP (1996) Role of optical flow field asymmetry in the perception of heading during linear motion. *PerceptPsychophys* 58:283–288.
- Wall MB, Smith AT (2008) The representation of egomotion in the human brain. *CurrBiol* 18:191–194.

Reviewers' comments:

Reviewer #1 (Remarks to the Author):

Once again, I thank the authors for their thoroughness and hard work. And, indeed, for the better copy-editing of this rebuttal. My opinion of the paper remains that it describes two interesting observations, but there are significant weaknesses in the modeling that links the two. I now agree that the data are probably too sparse to do better, however. I think there are only a couple of residual substantial concerns that must be addressed. Because the numbering has gotten complex, I will identify these by content instead.

Important concerns:

Causality: In the main body of the paper, the authors are sufficiently careful in their wording. However, in the abstract, the last sentence is still too strong. I strongly suggest replacing "... strongly suggest response properties of primate areas MST and VIP are the neural substrate ..." with "... suggest the response properties of the two areas are consistent with being the substrate of the newly described visual illusion." Also, on line 522, "prove" should be "proof".

Control experiment: The authors did a very good job in matching their wording to the strength of the experiment, since "not virtually blind" is indeed well supported by the data. However, virtual blindness is not to be expected from saccadic suppression, so the control experiment is so weak as to be essentially useless. The paper would be greatly strengthened if they could do a more sensitive control experiment. For instance, if they had subjects discriminating nearby (peri-threshold) headings on the pro-saccade and anti-saccade sides of the current gaze, they would strongly bolster their arguments for selective suppression. These data would not be difficult to obtain or analyze.

Suggestions:

Area and monkey breakdown: The table in the rebuttal that breaks down the physiological results by monkey and area was very reassuring, and I would suggest they consider including it either in the main text or in the supplemental material.

Timing: I think my confusion about timing was maintained longer than it had to be because of the complex and arcane description in that section of the text, starting about line 500 in the present version. If I was confused by this relatively simple point, I would hazard that many other readers will be as well. The authors should re-write this section from the ground up simplifying the argument as much as possible.

Reviewer #2 (Remarks to the Author):

The authors have done a thorough job of addressing my previous comments, and I believe the work is now ready for publication. I have nothing more to add.

We thank reviewer 1 for his/her very valuable and insightful comments. These helped us to further improve our manuscript. Like for the previous revision, we respond to the comments point-by-point by using the following fonts/styles to indicate:

Reviewers' comments (Calibri, bold, italics)

Our response to these comments (Courier)

New text in the revised, red-lined version of the manuscript. (Calibri, red)

Reviewers' comments:

Reviewer #1 (Remarks to the Author):

Important concerns:

Causality: In the main body of the paper, the authors are sufficiently careful in their wording. However, in the abstract, the last sentence is still too strong. I strongly suggest replacing "... strongly suggest response properties of primate areas MST and VIP are the neural substrate ..." with "... suggest the response properties of the two areas are consistent with being the substrate of the newly described visual illusion." Also, on line 522, "prove" should be "proof".

We have followed the reviewer's suggestion and replaced the last sentence by the sentence suggested by him/her. Thanks also for pointing towards the typo. In the new line 534, we have replaced "prove" by "proof".

Control experiment: The authors did a very good job in matching their wording to the strength of the experiment, since "not virtually blind" is indeed well supported by the data. However, virtual blindness is not to be expected from saccadic suppression, so the control experiment is so weak as to be essentially useless. The paper would be greatly strengthened if they could do a more sensitive control experiment. For instance, if they had subjects discriminating nearby (peri-threshold) headings on the pro-saccade and anti-saccade sides of the current gaze, they would strongly bolster their arguments for selective suppression.

We agree with the reviewer that virtual blindness is not to be expected from saccadic suppression but we nonetheless felt it important to show that stimuli such as our complex flow field can be appropriately discriminated even during the saccade and for the very brief presentation durations we used. While there is a large body of literature on optic flow perception, no prior work has used stimuli with such brief presentation, or looked at perception during saccades with optic flow. We therefore think our first control experiment is necessary to demonstrate that optic flow can be perceived in these situations. If this were not the case, our main experiment would not have been possible.

Based on reviewer #-1's previous feedback, we had already added a further control experiment in which we used a heading task rather than a forward vs. backward discrimination task. Here, subjects performed saccades with either the head or the gaze rotated to one side. This second control experiment had allowed us to determine the reference frame of the behavioral effects. Importantly, the result of this second control experiment, i.e. compression of perceived heading to occur in eye centered coordinates, was perfectly in line with results from neurophysiological recordings in the monkey, showing eye centered encoding of optic flow in areas MST and VIP (area MST: Fetsch et al., 2007; area VIP: Chen et al., 2013).

With regard to the reviewer's current suggestion, i.e. to discriminate **"nearby (peri-threshold) headings on the pro-saccade and anti-saccade sides of the current gaze"**, we would ask to observe that there are no anti-saccades involved in the present experiment and that the *selective suppression* is not related to the direction of the saccade but to the preference of the individual neurons in the populations encoding leftward and rightward heading. Any psychophysical experiment would involve both populations and hence is unlikely to yield a discrimination of the selective suppression. Moreover, due to the compression of heading that we have shown in the main experiment, a fine discrimination between nearby (peri-threshold) headings appears impossible as these headings will all be compressed peri-saccadically and hence be difficult to discriminate. Thus, any such experiment could only reiterate our finding of peri-saccadic compression of heading, but not act as a more sensitive control.

[REDACTED]

[REDACTED]

[REDACTED]

Suggestions:

Area and monkey breakdown: The table in the rebuttal that breaks down the physiological results by monkey and area was very reassuring, and I would suggest they consider including it either in the main text or in the supplemental material.

In the previous version we had followed the recommendation of reviewer #-2 to remove the table from the manuscript and refer to the

number of neurons in plain text in the manuscript. Following reviewer #-1's new feedback, we suggest, as a compromise, to have the table with the number of neurons in the supplements, which we have done in the revised version.

Timing: I think my confusion about timing was maintained longer than it had to be because of the complex and arcane description in that section of the text, starting about line 500 in the present version. If I was confused by this relatively simple point, I would hazard that many other readers will be as well. The authors should re-write this section from the ground up simplifying the argument as much as possible.

We appreciate the reviewers suggestion. We have completely rewritten the first half of this paragraph, which now reads as:

“Our data analyses revealed that the compression in the neurophysiological data was strongest 110ms after saccade onset. This timing is consistent with visual response latencies as summarized e.g. by Schmolesky and colleagues (Schmolesky et al., 1998). Visual onset response latencies in the macaque monkey vary from cell to cell and tend to increase from lower towards higher visual cortical areas (Schmolesky et al., 1998). Accordingly, a visual event occurring at a specific time induces a response in extrastriate or parietal areas starting roughly 100 ms later and peaking another 10 ms to 30 ms later. Taking this latency into account, the compression of perceived heading should be maximal for stimuli presented near saccade onset. This is exactly what we found in the psychophysical data. A quantitative comparison between the time courses of neural responses and those of the behavioral responses of humans yielded a temporal offset of $\tau = 116\text{ms}$, closely matching the average latency. Similar results were obtained in a previous study, where we aimed to determine a neural correlate of saccadic suppression (Bremmer et al., 2009). “

Reviewer #2 (Remarks to the Author):

The authors have done a thorough job of addressing my previous comments, and I believe the work is now ready for publication. I have nothing more to add.

Thanks .

References

- Bremmer F, Kubischik M, Hoffmann K-P, Krekelberg B (2009) Neural Dynamics of Saccadic Suppression. *JNeurosci* 29:12374–12383.
- Chen X, DeAngelis GC, Angelaki DE (2013) Eye-centered representation of optic flow tuning in the ventral intraparietal area. *JNeurosci* 33:18574–18582.
- Fetsch CR, Wang S, Gu Y, DeAngelis GC, Angelaki DE (2007) Spatial reference frames of visual, vestibular, and multimodal heading signals in the dorsal subdivision of the medial superior temporal area. *JNeurosci* 27:700–712.
- Greenlee MW, Frank SM, Kaliuzhna M, Blanke O, Bremmer F, Churan J, Cuturi LF, MacNeilage PR, Smith AT (2016) Multisensory Integration in Self Motion Perception. *Multisens Res* 29:1–32.
- Schmolesky MT, Wang Y, Hanes DP, Thompson KG, Leutgeb S, Schall JD, Leventhal AG (1998) Signal timing across the macaque visual system. *JNeurophysiol* 79:3272–3278.

Reviewers' Comments:

Reviewer #1 (Remarks to the Author):

I am content with the authors' response to the last round of review. Only one minor thing caught my eye, which I am sure was intentional. The former version of the paragraph describing the timing remains in the manuscript, lines 500-512. This should be removed before it goes to copy-editing.

Reviewer #-1 had correctly pointed out that while editing the previous version of our manuscript, we had doubled a paragraph (lines 501 – 512) in the Discussion section, starting with: “Our data analyses revealed that the compression in the neurophysiological data was strongest 110ms after saccade onset. This timing is consistent with ...”.

We now have removed lines 501 – 512 of the previous version of our manuscript.